# All Circuits Lead to Rome:
# Rethinking Functional Anisotropy in Circuit and Sheaf Discovery for LLMs

**Xi Chen** [* 1 2]  **Mingyu Jin** [* 3]  **Jingcheng Niu** [* † 4]  **Yutong Yin** [5]  **Jinman Zhao** [1]  **Bangwei Guo** [3]
**Dimitris N. Metaxas** [3]  **Zhaoran Wang** [5]  **Yutao Yue** [2 6]  **Gerald Penn** [1]

## Abstract

In this paper, we present empirical and theoretical evidence against a central but largely implicit assumption in circuit and sheaf discovery (CSD), which we term the *Functional Anisotropy Hypothesis*: the idea that functions in large language models (LLMs) are localised to a unique or near-unique internal mechanism. We show that a single LLM task can instead be supported by multiple, structurally distinct circuits or sheaves that are simultaneously faithful, sparse, and complete. To systematically uncover such competing mechanisms, we introduce Overlap-Aware Sheaf Repulsion, a method that augments the CSD objective with an explicit penalty on structural overlap across multiple discovery runs, enabling the discovery of circuits or sheaves with strong task performance but minimal shared structure across a plethora of common CSD benchmarks. We find that this phenomenon becomes increasingly pronounced as the number of discovered sheaves grows and persists robustly across major CSD methods. We further identify an ultra-sparse three-edge sheaf and show that none of its edges is individually indispensable, undermining even weakened notions of canonical or essential components. To explain these findings, we propose a *Distributive Dense Circuit Hypothesis* and provide a theoretical analysis demonstrating that non-unique, low-overlap circuit explanations arise naturally from high-dimensional superposition under mild assumptions. Together, our results suggest that

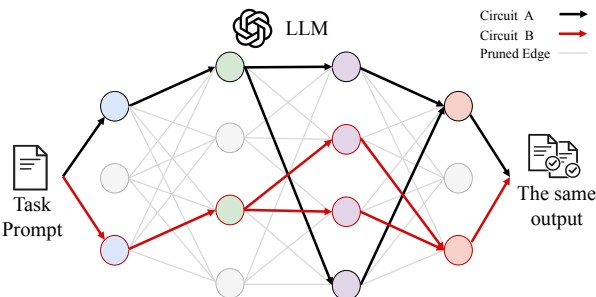

*Figure 1.* **Illustration of our findings.** We can identify multiple distinct circuits or sheaves that perform the same LLM task.

mechanistic explanations in LLMs are inherently non-canonical and call for a rethinking of how CSD results should be interpreted and evaluated.[1]

## 1. Introduction

Nowadays, circuit and sheaf discovery (CSD; Wang et al., 2022a; Conmy et al., 2023; Syed et al., 2024; Yu et al., 2025, *inter alia*) have emerged as promising directions in mechanistic interpretability, which aim to uncover the internal mechanisms and computational processes that underlie how language models (LMs) perform different tasks and exhibit various capabilities. These approaches aim to pry open the "black box" of large language models (LLMs) by grounding model behaviour in explicit and interpretable computational mechanisms, thereby providing principled explanations of their strong performance and offering insight into how they can be improved when failures arise.

A central but largely implicit assumption underlying prior CSD work is that LLM capabilities are supported by a single, well-defined internal mechanism. We refer to this assumption as the *Functional Anisotropy Hypothesis*. Under this view, model functions are assumed to be anisotropically distributed across the architecture, such that each capability corresponds to a unique circuit or sheaf.[2] This assumption underlies the dominant evaluation paradigms in CSD. Ex-

---

[*]Equal contribution [†]Work done while at University of Toronto [1]University of Toronto [2]The Hong Kong University of Science and Technology (Guangzhou) [3]Rutgers University [4]TU Darmstadt [5]Northwestern University [6]Institute of Deep Perception Technology, JITRI. Correspondence to: Gerald Penn <gpenn@cs.toronto.edu>, Yutao Yue <yutaoyue@hkust-gz.edu.cn>.

*Proceedings of the $43^{rd}$ International Conference on Machine Learning*, Seoul, South Korea. PMLR 306, 2026. Copyright 2026 by the author(s).

---

[1]Codes are available at: https://github.com/TonyXiChen/OASR.
[2]Circuits capture causally relevant subgraphs within the full

isting approaches either attempt to recover a hand-crafted ground-truth circuit, for example using Tracr (Lindner et al., 2023) and related benchmarks (Gupta et al., 2024), or aim to extract the smallest possible circuit that preserves task performance (Mueller et al., 2025). Both paradigms implicitly presume that a single minimal mechanism provides a sufficient explanation of how LLMs perform the task.

In this paper, we make a surprising observation: **multiple almost non-overlapping sheaves (or circuits) can each faithfully perform the same task independently** (illustrated in Figure 1). We introduce a novel *overlap-aware sheaf repulsion* process to systematically uncover such cases across tasks and settings. Using the classic indirect object identification (IOI) task as a case study, we show that multiple structurally distinct sheaves achieve equally strong performance. This non-uniqueness becomes more pronounced at scale: as we identify 20 high-quality sheaves, their union grows while their intersection steadily shrinks, revealing increasing structural divergence despite comparable performance. Strikingly, this phenomenon yields an ultra-sparse three-edge sheaf for IOI. Even in this extreme setting, no single component is indispensable: removing any of the three edges still allows the discovery of high-quality sheaves. Together, these results rule out even a weakened form of the Functional Anisotropy Hypothesis, in which mechanisms decompose into essential and auxiliary components.

Prior work has reported phenomena that appear related to our findings, most notably so-called *backup* mechanisms and the *hydra effect*; however, there are crucial differences, and these prior accounts cannot explain the phenomenon we observe. Notably, Wang et al. (2022a) identified Backup Name-Mover Heads in GPT-2 that remain largely inactive during normal inference but activate to recover task performance when the primary Name-Mover Heads are ablated. Similarly, McGrath et al. (2023) described the hydra effect, where ablating a crucial attention head or layer prompts an alternative component to take over its function as a compensatory backup. By contrast, our findings suggest a different regime: for a single task, multiple structurally distinct circuits or sheaves can coexist within the same model, each independently supporting the task and remaining causally relevant under intervention. This calls for a rethinking of the Functional Anisotropy Hypothesis: backup-style explanations treat redundancy as an ablation-triggered exception, but in our setting, non-uniqueness is a feature of normal model operation, making such accounts an unsatisfactory patch rather than a mechanism-level explanation.

Finally, we prove a *Distributive Dense Circuit Hypothesis*: for a given LLM task, there exist multiple structurally dis-

tinct, low-overlap mechanisms (circuits or sheaves) that are simultaneously faithful to the same task behaviour. Intuitively, this non-uniqueness arises because the space of valid computation subgraphs grows combinatorially with model depth and width, making it increasingly unlikely that a single sparse mechanism uniquely accounts for task execution.

This hypothesis has direct implications for how circuit and sheaf discovery results should be interpreted. Our findings do not invalidate CSD: the mechanisms it uncovers are clearly meaningful and causally relevant. However, they should no longer be interpreted as identifying *the* mechanism underlying a task. Rather, each discovered circuit or sheaf represents one valid realisation among many within a larger space of functionally equivalent mechanisms. As a result, a naively reductionist view—where task behaviour is attributed to a single sparse and indispensable subgraph—is insufficient to explain the observed empirical non-uniqueness. Instead, our results point toward a more distributed view of computation in LLMs, in which task behaviour emerges from a dense and compositional regime of competing, partially redundant mechanisms.

Overall, we raise a fundamental question and advance a rethinking of CSD through the following contributions:

- We formalise the problem of circuit and sheaf discovery and explicitly surface the *Functional Anisotropy Hypothesis*, a previously implicit assumption underlying much of prior work on mechanistic interpretability (§2).
- We provide empirical evidence that a single task or capability can be supported by multiple, structurally distinct mechanisms, and develop a principled framework for discovering such alternatives (§3).
- We show that even extremely compact explanations, including an ultra-sparse three-edge sheaf, do not admit uniquely essential components, challenging canonical notions of minimal or necessary mechanisms (§4).
- We propose the *Distributive Dense Circuit Hypothesis*, which offers a theoretical explanation for why multiple faithful, low-overlap circuits naturally arise in large language models (§5).

## 2. Preliminaries: Sheaf, Circuit and Functional Anisotropy

Before presenting our findings, we briefly review the notions of sheaves and circuit discovery, as well as the concept of *functional anisotropy*, which underlies the central motivation of the mechanistic interpretability literature.

### 2.1. Circuit and Sheaf Discovery

As discussed above, the core of CSD consists of three steps. ① We formalise the full computation of an LLM, using the residual stream formulation, as a DAG, which we refer to

---

model, whereas sheaves additionally require functional sufficiency in isolation; see §2.1 for a detailed discussion. We therefore focus on sheaves, though our findings apply to circuits as well.

as the *computation graph*; ② We then define a target task or capability by curating an appropriate dataset; ③ Finally, using a specific discovery method, we identify a subgraph whose activity corresponds to the model's performance on the task. Below, we provide a brief review of these concepts; a more detailed explanation is given in Appendix A.

**Residual Stream and the Computation Graph** Elhage et al. (2021) introduced the concept of the *residual stream*. Instead of describing a Transformer in terms of procedural pseudocode that specifies how the computation proceeds step by step, this perspective views the model as incrementally adding a series of updates to a central hidden state:

$$\mathbf{x}_{i+1} = \overbrace{\mathbf{x}_i + \sum_{h \in \mathcal{H}^{(i)}} \Delta_h^{(i)}(\mathbf{x}_i)}^{\mathbf{x}_i^{\mathrm{mid}}} + \Delta_{\mathrm{mlp}}^{(i)}(\mathbf{x}_i^{\mathrm{mid}}), \quad (1)$$

where $\mathbf{x}_i$ denotes the LM's output at layer $i$ (block $i$), $\mathcal{H}^{(i)}$ is the set of attention heads in block $i$, $\Delta_h^{(i)}(\cdot) \in \mathbb{R}^{n \times d}$ denotes the residual contribution produced by head $h$, and $\Delta_{\mathrm{mlp}}^{(i)}(\cdot) \in \mathbb{R}^{n \times d}$ denotes the residual contribution produced by the MLP. In Elhage et al.'s (2021) formulation, biases and layer normalisation are abstracted away for clarity, a convention we also follow throughout this paper.

Note that the residual stream defines a recursive structure and can be unrolled. The final output of the model $\mathbf{x}_{-1}$ is:

$$\mathbf{x}_{-1} = \mathbf{x}_0 + \sum_i \sum_{h \in \mathcal{H}^{(i)}} \Delta_h^{(i)}(\mathbf{x}_i) + \sum_i \Delta_{\mathrm{mlp}}^{(i)}(\mathbf{x}_i^{\mathrm{mid}}). \quad (2)$$

Each intermediate state $\mathbf{x}_i$ can itself be further unrolled; for readability, we omit this expansion here. This decomposition provides an explicit map of how information flows through the model. In particular, the output of any component is influenced by all components (attention heads and MLPs) in preceding layers. This structure naturally induces a DAG representation of computation. Particularly, for a function in the residual stream of the form $f(a+b)$, we introduce directed edges $a \to f$ and $b \to f$. Thus, in the computation graph $G = (V, E)$, $V$ consists of model components (attention heads, MLPs, and abstract input/output nodes), and $E$ contains directed edges representing the aforementioned information flow between components. In a standard Transformer language model (e.g., GPT-2 small with 12 layers and 12 attention heads), the output node receives edges from all preceding components; each MLP node receives edges from all components in earlier layers as well as the attention heads in its own layer; and each attention head receives edges from all components in preceding layers.

**Sheaf and Circuit Discovery** Interestingly, work that adopts this computation graph formulation has shown that a task can often be supported by only a sparse subgraph.

Wang et al. (2022a) are among the first to successfully demonstrate such results.[3] They introduce the *indirect object identification* (IOI) task, which has since become one of the most commonly used benchmark tasks in the field, and show that a sparse subnetwork contributes causally to IOI task performance, while the remaining components can be largely ablated with minimal effect. Inspired by this finding, ACDC (Conmy et al., 2023) automated circuit discovery and formalised an explicit residual-stream-based computation graph formulation that has since become the standard representation used in subsequent circuit discovery work.

CSD work can be broadly characterised along two axes: the discovery method (*heuristic* vs. *gradient*-based) and the underlying formulation (*circuits* vs. *sheaves*):

☆ ACDC (Conmy et al., 2023) adopts a heuristic, intervention-based approach, iteratively removing edges from the residual-stream computation graph, measuring the resulting causal effect on task performance, and pruning or retaining edges based on a chosen threshold.

☆ EAP (Syed et al., 2024) replaces iterative ablations with a gradient-attribution-based formulation, estimating the importance of edges via attribution scores and selecting task-relevant subgraphs through thresholding.

☆ Edge Pruning (EP; Bhaskar et al., 2024) represents a paradigm shift in circuit discovery by reframing it as a gradient-based optimisation problem. Edges whose masks are turned off are treated as pruned and replaced with patched baseline activations, obtained either by activation interchange from corrupted inputs or by mean aggregation, following the same intervention scheme as ACDC and EAP. Under this patched execution, circuit discovery is formulated as an optimisation problem whose objective is to find a sparse binary mask assignment that preserves task fidelity.

☆ DiscoGP (Yu et al., 2025) spotted a common limitation shared by circuit-based formulation. The discovered circuits are typically not *standalone*, meaning that they typically fail to reproduce the model's behaviour when executed in isolation (not replacing with patched activations where a lot of the original information retains, but with zero). To address this, they introduce the notion of a *sheaf*, defined as a subgraph that both causally contributes to the computation and can independently sustain task performance. Methodologically, DiscoGP formulates sheaf discovery as a joint optimisation problem over edge selections[4] using a Gumbel–Sigmoid relaxation with a straight-

---

[3]Note that Wang et al. (2022a) use a different computation graph formulation, treating the model as a graph over forward-pass terms with interactions defined by attention and value paths.

[4]DiscoGP also considers joint pruning of edges and weight parameters. In this paper, we focus exclusively on edges, as this yields a theoretically more manageable computation-graph formu-

through estimator, enabling gradient-based learning of discrete subgraphs while enforcing standalone fidelity.

Overall, optimisation-based methods tend to outperform heuristic approaches by jointly optimising edge selections, with DiscoGP reporting the state-of-the-art performance.

**Evaluation Protocol** Throughout the paper, we evaluate a discovered circuit or sheaf by executing the model on the corresponding masked computation graph, without modifying any model weights. Unless otherwise stated, inactive edges are zero-ablated: information is allowed to flow only through selected edges, and pruned edges contribute zero to downstream components. We report task accuracy as the fraction of examples for which the masked model preserves the correct task-level prediction criterion; for IOI, this corresponds to assigning higher probability to the correct indirect object than to the distractor name. We use *edge density* to denote the fraction of selected edges among all candidate edges, and measure structural similarity between two discovered mechanisms by edge-level intersection-over-union,

$$\text{IoU}(A, B) = \frac{|E_A \cap E_B|}{|E_A \cup E_B|}.$$

When reporting complement accuracy, we evaluate the complementary masked graph obtained by retaining the edges not selected by the discovered mechanism.

### 2.2. Functional Anisotropy Hypothesis

Much of the CSD literature implicitly assumes what we call the *Functional Anisotropy Hypothesis*: that a given function in a neural network is implemented by a unique, or near-unique, internal mechanism that is structurally privileged over alternatives. In practice, this assumption is reflected in how circuit discovery is framed as the identification of the subgraph responsible for a behaviour. For example, Wang et al. (2022a) describe their goal as "identifying an induced subgraph of the model's computational graph ... responsible for completing the task." Similarly, Conmy et al. (2023) characterise circuits as "subgraphs with distinct functionality," reinforcing the view that task-relevant computation can be localised to a specific mechanism. More recent optimisation-based approaches adopt the same framing: Bhaskar et al. (2024) define circuit discovery as "identifying the subset of a model's computational graph that is most relevant to a particular model behaviour."

More recently, Méloux et al. (2025) presents the most relevant formulation regarding the uniqueness and identifiability of mechanistic explanations. They formally examine whether a fixed model behavior admits a unique mechanistic explanation. Our work presents two key differences.

*Table 1.* **Evaluation of two sheaves discovered for IOI using OASR.** Both sheaves achieve perfect IOI accuracy and satisfy standard sheaf quality criteria, despite having only a 4.1% IoU overlap.

| Sheaf | IOI Acc. | Comp. Acc. | Edge Density | Edge # |
|-------|----------|------------|--------------|--------|
| $A$ | 100% | 46.20% | 3.56% | 1158 |
| $B$ | 100% | 45.80% | 3.97% | 1289 |

$$\|A \cap B\| = 96; \ \|A \cup B\| = 2351; \ \text{IoU}(A, B) = 4.1\%$$

First, Méloux et al. (2025) study simple models and tasks, whereas we evaluate pretrained LMs (e.g., GPT-2, Pythia) on realistic linguistic tasks. Second, their notion of "circuit" differs from our residual-based circuits or sheaves; their work is better viewed as providing foundational evidence for superposition in neural networks, rather than directly characterising circuits in transformer LMs. Nevertheless, despite these differences, their formulation is the closest in spirit to ours, and our empirical findings can be seen as carrying their identifiability question from toy settings into realistic LLMs and circuit-level mechanisms.

Importantly, this assumption is also baked into how CSD methods are evaluated. Tracr-based benchmarks and related semi-synthetic tasks assess success by how closely a discovered circuit matches a predefined ground-truth mechanism (Lindner et al., 2023; Gupta et al., 2024), implicitly treating the existence of a single correct explanation as given. Similarly, minimality-based evaluations such as MIB reward circuits that achieve high task performance with as few components as possible (Mueller et al., 2025), presuming that further compression converges toward a unique, essential core. In both cases, evaluation criteria implicitly encode the Functional Anisotropy Hypothesis by favouring explanations that are singular, minimal, and canonical.

## 3. Functional Plethorae of Mechanisms

In this section, we present our main empirical results, showing that a single LLM task can be faithfully realised by many distinct circuits and sheaves. We introduce *Overlap-Aware Sheaf Repulsion* (OASR; §3.1) to systematically discover low-overlap mechanisms, demonstrate that non-uniqueness intensifies as the number of discovered sheaves increases (§3.2), and show that this effect persists across multiple circuit and sheaf discovery methods beyond DiscoGP (§3.3).[5]

### 3.1. Discovering Functionally Equivalent Sheaves with *Overlap-Aware Sheaf Repulsion*

**Overlap-Aware Sheaf Repulsion** Recall that DiscoGP formulates sheaf discovery as a differentiable edge-selection

---

lation and aligns more closely with prior methods such as ACDC, EAP, and EP, which likewise operate solely at the level of edges.

[5]All results are reported on GPT-2, as it is the only model commonly supported by all prior CSD methods. See Appendix H for more results on other models.

problem by associating each edge $e \in E$ with a learnable logit $l_e$ and optimising a binary pruning mask $m_e \in \{0, 1\}$ under sparsity, fidelity, and completeness objectives. To make this discrete optimisation tractable a Gumbel–Sigmoid relaxation is used to compute a continuous edge score:

$$s_e = \sigma \left( \frac{l_e - \log \frac{\log \mathcal{U}_1}{\log \mathcal{U}_2}}{\tau} \right), \qquad (3)$$

where $\mathcal{U}_1, \mathcal{U}_2 \sim \mathrm{Uniform}(0, 1)$, and using a straight-through estimator (Bengio et al., 2013) to obtain a hard mask while preserving gradients. Sparsity is induced through a differentiable penalty on the expected edge activation:

$$\mathcal{L}_{\mathrm{sparsity}} = \frac{1}{|E|} \sum_{e \in E} \sigma(l_e), \qquad (4)$$

which softly discourages selecting unnecessary edges without imposing hard constraints.

Inspired by this, we introduce an *overlap-aware sheaf repulsion* mechanism to discover competing sheaves. After obtaining an initial sheaf $R$, we add an overlap loss that penalises the expected activation of edges already selected:

$$\mathcal{L}_{\mathrm{overlap}}(R) = \frac{1}{|E|} \sum_{e \in R} \sigma(l_e), \qquad (5)$$

and optimise the combined objective:

$$\mathcal{L} = \mathcal{L}_{\mathrm{fidelity}} + \lambda_s \mathcal{L}_{\mathrm{sparsity}} + \lambda_c \mathcal{L}_{\mathrm{complete}} + \lambda_o \mathcal{L}_{\mathrm{overlap}}(R). \qquad (6)$$

This additional term induces a repulsive interaction between successive discovery runs, encouraging the optimiser to identify alternative sheaves that satisfy fidelity and completeness while minimising structural overlap with previously discovered mechanisms. Repeating this process yields multiple competing, low-overlap sheaves that faithfully implement the same task.

**Two Sheaves for IOI**  We first present results on the IOI task, and later show that our findings generalise to other major benchmarks. IOI, introduced by Wang et al. (2022a), has since become the most widely used task for CSD.

Using the overlap-penalised procedure, we discover two sheaves, $A$ and $B$, that both perform IOI perfectly. As shown in Table 1, the two sheaves achieve identical IOI accuracy and are comparable across standard evaluation metrics, including completeness accuracy and edge density. By existing criteria, both $A$ and $B$ would be considered good sheaves and offer meaningful explanatory power. Notably, their overlap is extremely small: the intersection-over-union, $\mathrm{IoU}(A, B)$, is only 4.1%, which is close to the overlap expected by chance. This indicates that two largely distinct sheaves can support the same task, a finding that directly runs counter to the Functional Anisotropy Hypothesis.

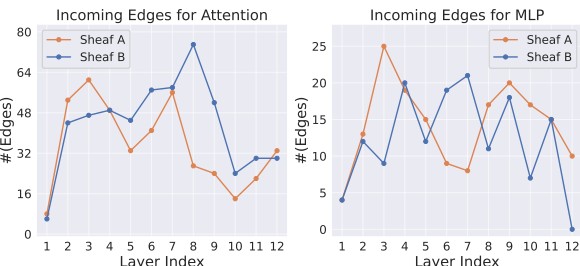

*(a)* The pair of IOI sheaves found using OASR with the least overall IoU. The distributions of incoming edges for MLPs differ substantially in the mid layers.

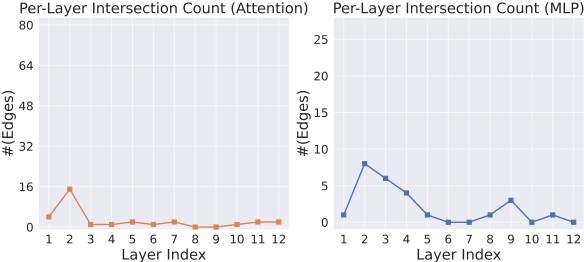

*(b)* The distribution of the number of incoming edges in the intersection for all layers in the above IOI sheaf pair for attention heads and MLPs. They agree more in the early layers, with very few edges in the intersection for the mid and later layers.

*Figure 2.* **Analysis of structural differences between two IOI sheaves.** The two sheaves exhibit markedly different layer-wise edge distributions despite identical task performance, indicating that the observed low overlap is not simply due to a trivial reparameterisation or rotation of model components.

This phenomenon is qualitatively different from previously discussed notions of redundancy, such as backup components or the hydra effect. Prior work has shown that certain edges or components may remain largely inactive during normal inference and only become engaged when the primary mechanism is disrupted. For example, Wang et al. (2022a) identified backup Name-Mover Heads that activate only after the main Name-Mover Heads are ablated, while McGrath et al. (2023) described the hydra effect, in which alternate components compensate for the removal of a critical head or layer. In contrast, our results reveal a fundamentally different picture: multiple circuits or sheaves that support the same task are already present and active during standard model operation. Rather than serving merely as latent backups, these sheaves represent alternative task-supporting mechanisms within the same model, suggesting that task-relevant computation in LLMs is not necessarily concentrated in a single privileged subgraph, but can be realised by multiple, structurally distinct and functionally viable subgraphs.

*Table 2.* **Evaluation of two sheaves discovered for common benchmark tasks.** For each task, we report the performance of the two discovered sheaves and their IoU overlap.

| Task | Sheaf $A$ Acc. | Sheaf $B$ Acc. | IoU($A, B$) |
|---|---|---|---|
| IOI | 100% | 100% | 4.1% |
| BLiMP | 96.8% | 92.6% | 5.1% |
| AGA | 96.0% | 95.3% | 6.2% |
| ANA | 98.0% | 91.3% | 5.3% |
| DNA | 100% | 96.2% | 5.8% |
| DNA i | 100% | 99.0% | 6.2% |
| DNA a | 98.5% | 97.0% | 7.5% |
| DNA ia | 100% | 99.0% | 6.4% |
| Docstring | 98.9% | 100% | 11.0% |

**A Closer Look at Structural Differences** To characterise how competing sheaves differ structurally, we analyse their layer-wise composition. Figure 2a shows the pair of IOI sheaves with minimal overall IoU, revealing sharply different distributions of edges across layers despite identical task performance. Complementarily, Figure 2b examines the pair with the least similar layer-wise density profiles, which likewise exhibit markedly different structural patterns even without explicit overlap minimisation. Together, these results show that competing sheaves differ not only at the level of individual edges but also in how task-relevant computation is distributed across layers, indicating genuinely distinct structural organisations rather than superficial reshufflings.

**All Common Tasks** This phenomenon is not specific to IOI. Applying the same overlap-penalised discovery procedure across a range of commonly used benchmark tasks, we consistently identify two sheaves per task that both achieve strong task performance. As summarised in Table 2, the two sheaves for each task are comparable under standard evaluation metrics, yet exhibit very low IoU overlap. Across tasks, the observed overlap is close to what would be expected by chance, indicating that the existence of multiple, largely distinct task-supporting sheaves is a general pattern rather than an artefact of a particular benchmark.

### 3.2. Mutual Intersection Further Diminishes Across Many Sheaves

Building on the two-sheaf results, we next examine what happens when the number of discovered sheaves is increased for a fixed task. For each task, we repeat the search 20 times. We use OASR throughout, and for comparison we also include results obtained without applying OASR, where diversity arises solely from different random initialisations. Table 3 summarises the results.

Across all tasks, we find that the mutual intersection across the 20 discovered sheaves is extremely small. In many cases, the global intersection contains only a few dozen edges, corresponding to a mutual IoU well below 1%. This holds even

though the individual sheaves consistently achieve strong task performance and satisfy standard quality criteria. Importantly, introducing the OASR further reduces the shared intersection across runs, while leaving sparsity and performance largely unchanged. Taken together, these results indicate that increasing the number of discovered sheaves does not lead to convergence onto a common core. Instead, task-relevant computation can be realised through many distinct sheaves with negligible shared structure.

The vanishing mutual intersection observed here cannot be explained as noise from random initialisation or instability in the discovery procedure. Even across repeated runs that all yield high-quality sheaves, the intersecting structure remains minimal, and shrinks further when overlap is explicitly penalised. This behaviour is inconsistent with the view that a task is supported by a small, specific subgraph that discovery methods should eventually converge to. Instead, it suggests that circuit and sheaf discovery are uncovering one of many viable realisations of task-relevant computation, rather than approximations to a single underlying mechanism. In this sense, functional anisotropy appears not as a fundamental property of the model, but as an artefact of searching for a single solution in a space that admits many.

### 3.3. Beyond DiscoGP: The Findings Persist Across Major Sheaf and Circuit Discovery Methods

So far, we have focused on results obtained with DiscoGP. We now ask whether the same conclusions hold for other widely used circuit discovery methods. In particular, we consider ACDC, EAP, and EP, which span different design choices, including heuristic edge removal, gradient-based attribution, and hybrid pruning strategies. Despite these methodological differences, all three approaches are commonly interpreted through the lens of functional anisotropy, namely the assumption that a task is supported by a small, privileged set of components that discovery methods should converge to. Repeating our analysis across these methods, we find the same qualitative pattern as before: multiple high-quality circuits can be discovered for the same task, while their overlap remains small and highly variable. This shows that the breakdown of functional anisotropy is not specific to DiscoGP, but persists across major circuit and sheaf discovery paradigms.

**ACDC Is Sensitive to Traversal Order** ACDC (Conmy et al., 2023) operates in reverse topological order, pruning edges deemed insignificant based on changes in KL divergence. Although largely deterministic for a fixed threshold, we find that ACDC is sensitive to the traversal order. In particular, there is no empirical or theoretical justification for treating attention head indices as semantically meaningful. Accordingly, we vary the traversal order of attention heads within each layer while preserving the overall reverse-

*Table 3.* **Mutual intersection further diminishes across many sheaves.** We report the size of the total intersection and union across 20 discovery runs, together with their ratio (Big IoU), as well as the average quality of the resulting sheaves. Despite consistently strong task performance, the mutual intersection across sheaves remains negligible.

| Task | Method | Mutual Intersection | | | Average Sheaf Quality | | | |
|---|---|---|---|---|---|---|---|---|
| | | $|E_\cap|$ | $|E_\cup|$ | Mutual IoU | Edge D. | $\#E$ | Acc. (%) | Comp. (%) |
| IOI | Random Init | 20 | 6560 | 0.30% | 3.04% | 987.0 | 99.95 | 45.64 |
| | OASR | 11 | 7382 | 0.15% | 2.86% | 928.5 | 99.59 | 45.87 |
| BLiMP | Random Init | 50 | 4858 | 1.03% | 2.62% | 852.0 | 97.26 | 44.78 |
| | OASR | 37 | 5289 | 0.70% | 2.34% | 758.7 | 96.11 | 45.75 |
| AGA | Random Init | 21 | 4758 | 0.44% | 2.42% | 785.2 | 94.67 | 43.10 |
| | OASR | 15 | 5791 | 0.26% | 2.35% | 764.8 | 94.43 | 43.23 |
| ANA | Random Init | 26 | 4531 | 0.57% | 2.21% | 718.8 | 96.40 | 40.10 |
| | OASR | 10 | 4890 | 0.20% | 1.91% | 621.4 | 95.00 | 39.77 |
| DNA | Random Init | 17 | 3134 | 0.54% | 1.51% | 489.2 | 96.73 | 55.38 |
| | OASR | 17 | 3440 | 0.49% | 1.33% | 431.1 | 95.11 | 55.30 |
| DNA i | Random Init | 19 | 3649 | 0.52% | 1.75% | 569.9 | 98.65 | 55.45 |
| | OASR | 9 | 3252 | 0.28% | 1.18% | 384.4 | 96.75 | 54.65 |
| DNA a | Random Init | 23 | 3982 | 0.58% | 2.16% | 701.2 | 96.35 | 49.74 |
| | OASR | 22 | 4469 | 0.49% | 2.11% | 684.6 | 95.26 | 51.77 |
| DNA ia | Random Init | 33 | 4813 | 0.69% | 2.59% | 842.1 | 95.00 | 50.80 |
| | OASR | 25 | 5063 | 0.49% | 2.42% | 786.0 | 93.90 | 52.40 |
| Docstring | Random Init | 39 | 6719 | 0.58% | 3.39% | 1100.9 | 99.61 | 51.65 |
| | OASR | 18 | 7314 | 0.25% | 3.08% | 999.8 | 99.04 | 51.75 |

*Table 4.* ACDC circuit results on IOI at $\tau = 0.00398$ under interchange and zero ablation.

| Ablation | Intersection | | Average Metrics | | |
|---|---|---|---|---|---|
| | $|E_\cap|$ | $|E_\cup|$ | Edge D. | $\#E$ | Acc. (%) |
| Interchange | 174 | 608 | 1.09% | 353.4 | 90.75 |
| Zero | 302 | 6951 | 6.76% | 2194.7 | 87.50 |

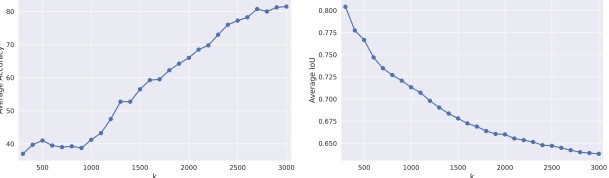

*(a)* Average task accuracy of EAP circuits on IOI as a function of the top-$k$ selection threshold.

*(b)* Average pairwise EAP circuit IoU on IOI as a function of the top-$k$ selection threshold

*Figure 3.* EAP circuits' task performance and pairwise overlap as a function of the top-$k$ selection threshold.

topological structure, in contrast to the default increasing head-index order used by Conmy et al. (2023). Even with an identical threshold, we observe substantially different circuits on IOI across random seeds, with far-from-unity IoUs (Table 4). Under zero ablation, the resulting circuits are denser and exhibit lower structural overlap. Together, these findings indicate that the circuits recovered by ACDC are not uniquely determined, providing further evidence against the Functional Anisotropy Hypothesis.

**EAP Is Sensitive to Task-Irrelevant Information** In the IOI task, the specific names used in the prompt are task-irrelevant: prompts that differ only by replacing `John` and `Mary` with `Alice` and `Bob` instantiate the same template and require identical reasoning. EAP estimates per-edge importance via a first-order approximation (Syed et al., 2024), with the number of retained edges $k$ as its sole hyperparameter. The method is otherwise deterministic, relying on two forward passes and one backward pass to compute attribu-

tions for all edges simultaneously. Nevertheless, when we vary only the names in the IOI prompts, we observe systematically decreasing IoUs between the resulting circuits as $k$ increases (Figure 3). This sensitivity to task-irrelevant variation indicates that EAP does not recover a stable or canonical mechanism even in a highly controlled setting. More broadly, these results challenge the Functional Anisotropy Hypothesis: if a unique, structurally privileged mechanism underlies IOI, its recovered circuit should be invariant to superficial changes that do not alter the task itself.

**Edge Pruning Mirrors DiscoGP** Despite differences in ablation strategies, both Edge Pruning (EP) and DiscoGP aim to identify sparse, task-relevant subgraphs over edges. Using the original EP objective, which minimises KL di-

*Table 5.* Circuit discovery results with EP. The IoUs from using task-specific cross entropy loss are much lower than the IoUs obtained by using the full vocabulary distribution-based KL divergence loss. This implies the full distributional alignment yields the seeming algorithmic stability.

| Task | Loss Type | Consistency Metrics | | | Average Metrics | | |
|------|-----------|--------|--------|--------|-----------|-------|---------|
| | | Min IoU | $\#E_\cap$ | $\#E_\cup$ | Edge D. (%) | $\#E$ | Acc. (%) |
| IOI | CE | 0.064 | 35 | 644 | 0.888 | 319.8 | 98.65 |
| | KL | 0.340 | 226 | 787 | 1.513 | 498.5 | 96.95 |
| Docstring | CE | 0.109 | 329 | 3241 | 4.875 | 1749.0 | 99.75 |
| | KL | 0.344 | 1570 | 5512 | 10.313 | 3427.5 | 88.25 |

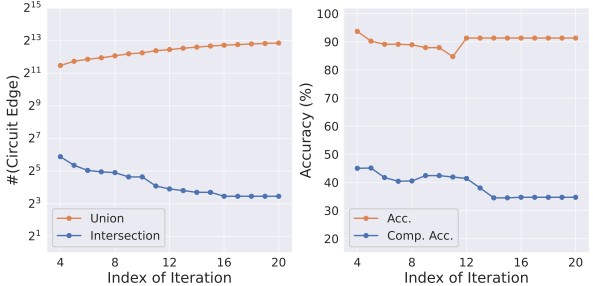

*Figure 4.* The sheaf profiles of iterative intersections and unions of sheaves discovered by DiscoGP with OASR for IOI.

vergence to the full model, we find that circuits discovered across random seeds exhibit relatively high pairwise IoUs (Table 5), likely because optimising over the full vocabulary encourages retention of edges that preserve global output alignment rather than task-specific computation. When we replace the KL-divergence objective with the same task-specific loss used by DiscoGP, EP exhibits substantially lower circuit consistency (sometimes even exceeding the diversity induced by DiscoGP with overlap penalties) while producing sparser but less independent circuits due to interchange ablation. Although EP's Lagrangian sparsity constraint prevents directly applying overlap penalties, these results show that circuit non-uniqueness is not an artefact of DiscoGP but also emerges within the EP paradigm once task-specific objectives are considered, providing further evidence against the Functional Anisotropy Hypothesis.

Together, these results challenge the Functional Anisotropy Hypothesis from complementary directions: existing CSD methods show that single-circuit explanations are operationally inconsistent under minor task-preserving or methodologically arbitrary changes, while OASR shows that low-overlap faithful mechanisms can be recovered more explicitly by directly penalising structural overlap.

## 4. A Three-Edge Sheaf and the Absence of Indispensable Components

Even if CSD often yields multiple low-overlap mechanisms for the same task, one might still hope for a weaker form of

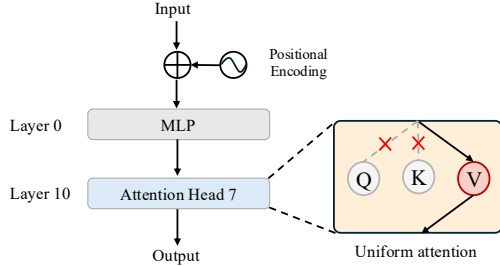

*Figure 5.* The three-edge IOI circuit under zero ablation.

functional anisotropy: that a small set of shared components persists across discoveries and constitutes a canonical core. Under this view, non-uniqueness arises only in peripheral structure, while essential computation is concentrated in a compact, indispensable subcircuit. In this section, we show that this weaker hypothesis also fails by identifying an ultra-sparse three-edge sheaf for IOI and demonstrating that none of its edges are individually indispensable. Focusing on IOI, we intersect edge sets across multiple independently discovered circuits and evaluate the resulting subgraphs under zero ablation. These intersection circuits remain highly functional even as they shrink, as shown in Figure 4, achieving over 90% task accuracy when reduced to 11 edges, while their complements remain near random-guess performance. Starting from this 11-edge core, we exhaustively search for the most compact subgraph that preserves task performance and identify an ultra-sparse three-edge sheaf (illustrated in Figure 5) that achieves 86.7% IOI accuracy in isolation, with its complement remaining close to chance level. Note that even the initial embedding layer is treated as the source node and can only influence downstream computation through selected masked edges. At first glance, this three-edge sheaf appears indispensable. Removing these edges from previously discovered IOI circuits reduces average task accuracy to 52.3% across OASR-discovered circuits, and explicitly prohibiting them during discovery prevents DiscoGP from recovering circuits with high functional fidelity (task accuracy >80%). Taken in isolation, these results suggest that the three-edge sheaf captures a canonical mechanism repeatedly relied upon by the model and the sheaves uncovered.

However, this conclusion depends critically on treating IOI as a single, undifferentiated task. When we decompose IOI into ABBA and BABA templates and perform sheaf discovery under the constraint that none of the three edges are present, we still recover sparse, highly functional circuits, as shown in Table 6, with edge densities below 3.5%. A more fine-grained internal ablation result is given in Table 7. These results establish two levels of non-indispensability. Locally, within the three-edge sheaf, removing any single edge leaves performance essentially unchanged. Globally, each edge can be avoided by alternative discovered sheaves, showing that no individual edge is intrinsically required by

*Table 6.* Functional role of the three-edge core under task aggregation and decomposition. We refer to the 3-edge sheaf as the core in the constraint. IOI sheaves' accuracy and complement accuracy are the average over 20 sheaves.

| Sheaf | Task | Constraint | Acc. | Comp. |
|---|---|---|---|---|
| 3-edge Sheaf | IOI | None | 86.7% | 31.3% |
| $C_{\text{BABA}}$ | IOI-BABA | Core excluded | 94.9% | 48.0% |
| $C_{\text{ABBA}}$ | IOI-ABBA | Core excluded | 96.7% | 49.2% |
| IOI Sheaves | IOI | Core excluded | 52.3% | 48.2% |

*Table 7.* Ablation of the three-edge core in the full model. The three core edges are $e_1 : \text{Input} \rightarrow \text{MLP}_0$, $e_2 : \text{MLP}_0 \rightarrow \text{Attn}_{10}\text{H}_7 V$, and $e_3 : \text{Attn}_{10}\text{H}_7 \rightarrow \text{Output}$. Each row reports IOI accuracy after removing the specified subset of these core edges from the otherwise unmodified full model. Thus, "# kept core edges" refers only to how many of $e_1, e_2, e_3$ remain active; all non-core edges in the full model are kept active throughout.

| Core edge(s) removed | # kept core edges | Acc. |
|---|---|---|
| None | 3 | 100.0% |
| $e_1$ | 2 | 99.9% |
| $e_2$ | 2 | 99.9% |
| $e_3$ | 2 | 99.8% |
| $e_2, e_3$ | 1 | 31.6% |
| $e_1, e_3$ | 1 | 31.4% |
| $e_1, e_2$ | 1 | 31.3% |
| $e_1, e_2, e_3$ | 0 | 31.2% |

all functionally valid mechanisms. The stronger-looking indispensability of the full three-edge core arises only when IOI is treated as an aggregated task.

## 5. A Distributive Dense Circuit Hypothesis

In this section, we present the central theoretical contribution of this work and introduce the *Distributive Dense Circuit Hypothesis*. We challenge a prevailing assumption that a task admits a unique or near-unique sparse internal circuit explanation. Instead, we show that even under strong faithfulness constraints, the same task behaviour can be realised by multiple, structurally distinct circuits.

Our main theoretical result establishes that circuit explanations are, in general, *non-unique*: we prove an *existence* statement showing that there *exist* multiple structurally distinct (low-overlap) circuits that are simultaneously faithful to the same task behaviour. The non-uniqueness of circuits is not accidental but a direct consequence of linear superposition in high-dimensional representations. Under mild local linearity assumptions, task-relevant readouts decompose into sums of edge-wise contributions. As the number of candidate circuits grows combinatorially with model depth and width, distinct edge subsets inevitably produce nearly identical readout vectors via subset-sum collisions. When the task decision is protected by a finite margin, these

readout-level collisions translate into identical predictions despite substantial structural differences between circuits. Our theoretical analysis formalises this intuition and proves the existence of multiple low-overlap, $\varepsilon$-faithful circuits for the same task (as shown in Appendix A, Theorem A.4).

## 6. Conclusion

We presented empirical and theoretical evidence against the *Functional Anisotropy Hypothesis*, the implicit assumption that a given LLM capability is realised by a unique or near-unique internal mechanism. Across tasks and discovery methods, we showed that multiple circuits or sheaves can be simultaneously faithful, sparse, and complete while exhibiting minimal structural overlap, and that even seemingly canonical explanations (including an ultra-sparse three-edge sheaf for IOI) do not contain indispensable components once task structure is taken into account.

Nevertheless, our findings do not undermine CSD: the discovered mechanisms remain meaningful and causally relevant. Rather, we call for a rethinking of how such mechanisms are interpreted and evaluated, moving away from canonical or minimal explanations toward a more distributed view in which task behaviour is realised by a space of functionally equivalent, competing mechanisms.

## Impact Statement

This paper presents work whose goal is to advance the field of machine learning. There are many potential societal consequences of our work, none of which we feel must be specifically highlighted here.

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

# Contents

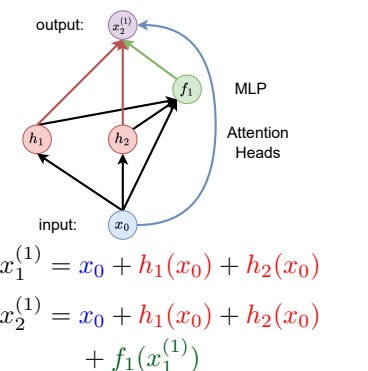

$$x_1^{(1)} = x_0 + h_1(x_0) + h_2(x_0)$$
$$x_2^{(1)} = x_0 + h_1(x_0) + h_2(x_0)$$
$$+ f_1(x_1^{(1)})$$

*(a)* Term-by-term unrolled DAG of a small Transformer and its residual stream.

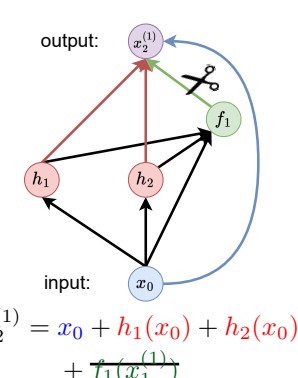

$$x_2^{(1)} = x_0 + h_1(x_0) + h_2(x_0)$$
$$+ \overline{f_1(x_1^{(1)})}$$

*(b)* Edge removal as vanishing residual term in the unrolled residual stream.

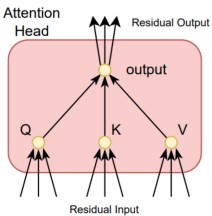

*(c)* Attention head granularity. In practice, attention heads are often decomposed at a finer granularity. Following Conmy et al. (2023), query (Q), key (K), and value (V) activations (together with a separate output node) are treated as distinct components within each attention head in the empirical analysis. For clarity, and without loss of generality, we ignore this decomposition in the proof.

*Figure 6.* Edge pruning viewed as structured removal of residual-stream terms at the level of attention head components. Illustration taken from Yu et al. (2025).

## A. Existence Theorem for Circuit Non-Uniqueness

### A.1. Preliminaries

**Transformer Blocks and Computational Graph** Let $F_\theta$ denote a decoder-only transformer language model with parameters $\theta$. Given an input sequence $x = (x_1, \ldots, x_n)$, the model maps it to a distribution over next-token logits. We focus on the internal computation of $F_\theta$ through its ***residual stream***.

Let $\mathbf{x}_i \in \mathbb{R}^{n \times d}$ denote the residual stream entering the $i$-th transformer block ($i = 0, \ldots, L-1$), where $L$ is the number of blocks and $d$ is the model width. Each block consists of a *multi-head attention (MHA)* module followed by an *MLP* module, each wrapped by residual addition $\oplus$. Abstracting away normalization details (e.g. *Pre-LN* or *RMSNorm*), we write the block computation in an *edge-additive* form over the residual stream:

$$\mathbf{x}_{i+1} = \mathbf{x}_i + \sum_{h \in \mathcal{H}^{(i)}} \Delta_h^{(i)}(\mathbf{x}_i) + \Delta_{\mathrm{mlp}}^{(i)}(\mathbf{x}_i^{\mathrm{mid}}), \qquad \mathbf{x}_i^{\mathrm{mid}} = \mathbf{x}_i + \sum_{h \in \mathcal{H}^{(i)}} \Delta_h^{(i)}(\mathbf{x}_i). \tag{7}$$

Here $\mathcal{H}^{(i)}$ is the set of attention heads in block $i$, $\Delta_h^{(i)}(\cdot) \in \mathbb{R}^{n \times d}$ denotes the residual contribution produced by head $h$, and $\Delta_{\mathrm{mlp}}^{(i)}(\cdot) \in \mathbb{R}^{n \times d}$ denotes the residual contribution produced by the MLP. Crucially, we define a *residual-stream edge* as each additive term that injects information into the residual stream: the identity (skip) edge contributes $\mathbf{x}_i$ to $\mathbf{x}_{i+1}$, each head-edge contributes $\Delta_h^{(i)}(\mathbf{x}_i)$ into $\mathbf{x}_i^{\mathrm{mid}}$ (and hence into $\mathbf{x}_{i+1}$), and the MLP-edge contributes $\Delta_{\mathrm{mlp}}^{(i)}(\mathbf{x}_i^{\mathrm{mid}})$ into $\mathbf{x}_{i+1}$. A circuit will be defined as a sparse subset of these residual-stream edges whose execution preserves the target behaviour.

**Unrolled Residual-Stream Computation** Eq. (7) implies that the residual stream at each depth can be expressed as a sum of *component contributions* (embedding, attention-head outputs, and MLP outputs). For instance, in a one-block transformer with two heads $(h_1, h_2)$ and one MLP $f$, The unrolled form reads

$$\mathbf{x}^{\mathrm{mid}} = \mathbf{x}_0 + \Delta_{h_1}(\mathbf{x}_0) + \Delta_{h_2}(\mathbf{x}_0), \ \mathbf{x}_1 = \mathbf{x}^{\mathrm{mid}} + \Delta_{\mathrm{mlp}}(\mathbf{x}^{\mathrm{mid}}) = \mathbf{x}_0 + \Delta_{h_1}(\mathbf{x}_0) + \Delta_{h_2}(\mathbf{x}_0) + \Delta_{\mathrm{mlp}}\big(\mathbf{x}_0 + \Delta_{h_1}(\mathbf{x}_0) + \Delta_{h_2}(\mathbf{x}_0)\big).$$
$$\tag{8}$$

**Circuit Pruning Over Residual-Stream Edges** For the purposes of the proof, the distinction between circuits and sheaves is immaterial. Without loss of generality, we use the term circuit throughout. Under the edge-additive view in Eq. (7), *circuit pruning* is implemented by selectively *removing residual-stream injections*. Concretely, for each block $i$ we introduce binary masks $m_h^{(i)} \in \{0, 1\}$ for every head $h \in \mathcal{H}^{(i)}$ and $m_{\mathrm{mlp}}^{(i)} \in \{0, 1\}$ for the MLP (we keep the skip/identity edge by default). The circuit-restricted execution replaces Eq. (7) with

$$\mathbf{x}_i^{\mathrm{mid}, C} = \mathbf{x}_i^C + \sum_{h \in \mathcal{H}^{(i)}} m_h^{(i)} \Delta_h^{(i)}(\mathbf{x}_i^C), \qquad \mathbf{x}_{i+1}^C = \mathbf{x}_i^{\mathrm{mid}, C} + m_{\mathrm{mlp}}^{(i)} \Delta_{\mathrm{mlp}}^{(i)}(\mathbf{x}_i^{\mathrm{mid}, C}), \tag{9}$$

where the superscript $C$ denotes the activations produced by the circuit-pruned execution (i.e., with masked edges ablated). Setting $m_h^{(i)} = 0$ *cuts off* the head-edge injection $\Delta_h^{(i)}(\mathbf{x}_i)$, and setting $m_{\mathrm{mlp}}^{(i)} = 0$ cuts off the MLP-edge injection $\Delta_{\mathrm{mlp}}^{(i)}(\mathbf{x}_i^{\mathrm{mid}})$. In the one-block example of Eq. (8), cutting off the MLP-edge yields

$$\mathbf{x}_1^C \;=\; \mathbf{x}_0 + \Delta_{h_1}(\mathbf{x}_0) + \Delta_{h_2}(\mathbf{x}_0), \tag{10}$$

matching the intuition in Fig. 6 that the output residual stream can be reproduced by executing only a subset of residual-stream edges. More generally, a (residual-stream) circuit corresponds to a sparse choice of masks $C = \{m_h^{(i)}, m_{\mathrm{mlp}}^{(i)}\}_{i,h}$ whose execution preserves the target behaviour up to a specified tolerance, while all omitted edges are replaced by a fixed baseline (typically zero ablation).

### A.2. Task Fidelity and Circuit Non-Uniqueness

**Task and Fidelity Metric**    Let $\mathcal{D}$ denote a task distribution over prompts (e.g., IoI and BLiMP). Let $\mathcal{M}(F; \mathcal{D})$ be a scalar performance metric (accuracy). Given an edge-mask circuit $C$, we write $F_\theta^C$ for the circuit-pruned execution in Eq. (9). We say that $C$ is $\varepsilon$-*faithful* on $\mathcal{D}$ if

$$\mathcal{M}(F_\theta^C; \mathcal{D}) \geq \mathcal{M}(F_\theta; \mathcal{D}) - \varepsilon. \tag{11}$$

Here $\varepsilon \geq 0$ is a small tolerance that quantifies the maximum allowable performance drop under pruning: when $\mathcal{M}$ is accuracy, $\varepsilon$ corresponds to at most an $\varepsilon$-absolute decrease in accuracy.

In the circuit discovery area, a "task" is typically specified by a distribution $\mathcal{D}$ over prompts such as the IoI dataset or BLiMP suites. A circuit is said to "implement" the task when the pruned model $F_\theta^C$ retains nearly the same performance as the full model on prompts sampled from $\mathcal{D}$. Importantly, $\mathcal{D}$ represents a *family* of instances: if a circuit is $\varepsilon$-faithful on $\mathcal{D}$, then it preserves performance not just on a single example but across the entire distribution. Consequently, any circuit that is faithful for $\mathcal{D}$ is also faithful for any subset of instances drawn from $\mathcal{D}$ (e.g., any particular prompt or sub-benchmark), up to the same tolerance.

**Circuit Non-Uniqueness**    We say the model exhibits *circuit non-uniqueness* on $\mathcal{D}$ if there exist two distinct circuits $C_1 \neq C_2$ such that both are $\varepsilon$-faithful. To quantify how different two circuits are, we measure the overlap of their active residual-stream edges using *Intersection-over-Union (IoU)*. Let $E(C)$ denote the set of (non-skip) residual-stream edges kept by circuit $C$ (i.e., edges with mask value 1). We define

$$\mathrm{IoU}(C_1, C_2) \triangleq \frac{\big|E(C_1) \cap E(C_2)\big|}{\big|E(C_1) \cup E(C_2)\big|} \in [0, 1]. \tag{12}$$

A smaller IoU indicates more dissimilar circuits, with $\mathrm{IoU}(C_1, C_2) = 0$ corresponding to disjoint edge sets. We say the model exhibits *strong* circuit non-uniqueness on $\mathcal{D}$ if there exist two $\varepsilon$-faithful circuits $C_1, C_2$ whose overlap is small, e.g.,

$$\mathrm{IoU}(C_1, C_2) \leq \tau \tag{13}$$

for some $\tau \ll 1$ (equivalently, their symmetric difference is large).

### A.3. Assumptions and Model Class for the Existence Result

To make the existence result precise, we work with a controlled model class that matches the residual-stream edge formalism in Eq. (9). Our assumptions are mild and standard in theoretical treatments of transformer computations and circuit abstractions.

**Assumption A.1** (Residual-additive edge model)**.** We model circuit pruning as masking *residual-stream injections* in an edge-additive residual stream (Eq. (9)). In particular, when an edge is masked out, its injected contribution is replaced by a fixed baseline (zero ablation by default). We omit normalization layers (e.g., Pre-LN/RMSNorm) to preserve the explicit additive decomposition of residual-stream edges.

Assumption A.1 fixes the semantics of residual-stream edge ablation: a circuit is simply a subset of additive injection terms in the residual stream, and pruning corresponds to removing the corresponding terms while leaving the rest of the computation unchanged.

**Assumption A.2** (Task-relevant readout and margin stability (for accuracy)). There exists a task-relevant readout map $g$ that maps the final residual stream to prediction logits. For each prompt $x \sim \mathcal{D}$, let $\mathbf{z}(x) \triangleq g(\mathbf{x}_L(x)) \in \mathbb{R}^{|\mathcal{V}|}$ denote the logits of the reference execution (e.g., the full model), and define its predicted label

$$\hat{y}(x) \triangleq \arg\max_{y \in \mathcal{V}} \mathbf{z}(x)_y. \tag{14}$$

Define the (top-1) logit margin as

$$\gamma(x) \triangleq \mathbf{z}(x)_{\hat{y}(x)} - \max_{y \neq \hat{y}(x)} \mathbf{z}(x)_y. \tag{15}$$

Assume the reference execution is $\gamma$-separated on $\mathcal{D}$, i.e., $\gamma(x) \geq \gamma$ for all $x \in \operatorname{supp}(\mathcal{D})$. Moreover, for any circuit-pruned execution $F_\theta^C$, its logits $\mathbf{z}^C(x) \triangleq g(\mathbf{x}_L^C(x))$ satisfy

$$\left\| \mathbf{z}^C(x) - \mathbf{z}(x) \right\|_\infty \leq \delta \quad \text{for all } x \in \operatorname{supp}(\mathcal{D}). \tag{16}$$

Assumption A.2 formalises when accuracy is invariant to pruning: if the reference execution has margin at least $\gamma$ and the circuit-induced logit perturbation is bounded by $\delta < \gamma/2$, then the top-1 prediction is unchanged for every $x \in \mathcal{D}$. Indeed, the new margin is at least $\gamma(x) - 2\delta > 0$, so $\arg\max \mathbf{z}^C(x) = \arg\max \mathbf{z}(x)$, and therefore the accuracy metric is preserved under pruning.

**Assumption A.3** (Local linearisation / fixed activation regime). On the task distribution $\mathcal{D}$, the effect of residual-stream edges on the task-relevant readout can be locally approximated by a linear map. That is, there exists a reference execution (e.g., the full model or a reference circuit) such that for all circuits considered in our theorem, the readout satisfies

$$g(\mathbf{x}_L^C(x)) \approx g(\mathbf{x}_L(x)) + \sum_{e \in E(C)} \mathbf{J}_e(x), \tag{17}$$

where $\mathbf{J}_e(x)$ denotes the (first-order) contribution of keeping edge $e$ at input $x$ (e.g., via Jacobian/linearisation), and the approximation error is uniformly bounded by a small constant. Equivalently, for piecewise-linear networks (e.g., ReLU/GELU MLPs), we assume the activation patterns are fixed on $\operatorname{supp}(\mathcal{D})$, so that the computation is exactly linear in the edge injections within this regime.

Assumptions A.1–A.3 allow us to reason about circuits as sparse subsets of residual-stream edges and to establish a constructive existence result for two $\varepsilon$-faithful but low-overlap (i.e., small-IoU) circuits.

## A.4. Main Theorem: Existence of Disjoint Circuits

We now state our main result: under the assumptions of residual additivity and local linearisation, there exists a specific redundancy in the model such that multiple distinct circuits can solve the same task.

**Theorem A.4** (Existence of low-overlap faithful circuits). *Let $F_\theta$ be a Transformer model and $\mathcal{D}$ be a task distribution. Assume Assumptions A.1 and A.3. In addition, assume there exist two edge sets $E_A$ and $E_B$ with $\operatorname{IoU}(E_A, E_B) \leq \tau$ such that their linearized readout contributions match on $\mathcal{D}$:*

$$\max_{x \sim \mathcal{D}} \left\| \sum_{e \in E_A} \mathbf{J}_e(x) - \sum_{e \in E_B} \mathbf{J}_e(x) \right\|_\infty \leq \delta. \tag{18}$$

*If $\mathcal{M}$ is accuracy and Assumption A.2 holds with margin $\gamma$ and perturbation bound $\delta < \gamma/2$, then there exist two circuits $C_1$ and $C_2$ with $E(C_1) = E_A$ and $E(C_2) = E_B$ such that:*

1. *Faithfulness. Both circuits preserve performance on $\mathcal{D}$ up to tolerance:*

$$\mathcal{M}(F_\theta^{C_1}; \mathcal{D}) \geq \mathcal{M}(F_\theta; \mathcal{D}) - \varepsilon, \qquad \mathcal{M}(F_\theta^{C_2}; \mathcal{D}) \geq \mathcal{M}(F_\theta; \mathcal{D}) - \varepsilon, \tag{19}$$

   *and in fact $\varepsilon = 0$ for accuracy under the margin condition $\delta < \gamma/2$.*

2. *Structural disparity (low overlap). The circuits rely on substantially different residual-stream edges:*

$$\operatorname{IoU}(C_1, C_2) = \frac{|E(C_1) \cap E(C_2)|}{|E(C_1) \cup E(C_2)|} \leq \tau, \qquad \text{with } \tau \ll 1. \tag{20}$$

   *In particular, $\operatorname{IoU}(C_1, C_2) \leq \tau < 1$ implies $C_1 \neq C_2$ (equivalently, $E(C_1) \neq E(C_2)$).*

Theorem A.4 shows that whenever two low-overlap edge subsets are redundant in their task-relevant contributions, the task admits multiple faithful circuit explanations: distinct circuits with $\text{IoU} \ll 1$ can preserve accuracy.

## A.5. Proof of Theorem A.4

### Step 1: Reduce Circuits to Subset Sums

**Lemma A.5** (Subset-sum representation of circuit effects). *Fix a finite evaluation set $\mathcal{D} = \{x^{(1)}, \ldots, x^{(N)}\}$ and a task-relevant readout map $g : \mathbb{R}^{n \times d} \to \mathbb{R}^m$. Let $\mathbf{x}_L(x)$ denote the final residual stream under a* reference execution, *which we take to be the full model $F_\theta$ (i.e., the circuit that keeps all prunable residual-stream edges). Define the reference readout*

$$\mathbf{z}(x^{(j)}) \triangleq g(\mathbf{x}_L(x^{(j)})) \in \mathbb{R}^m, \qquad j = 1, \ldots, N.$$

*Assume Assumption A.3. Then for any circuit $C$ considered in the theorem, there exist edge-wise contributions $\mathbf{J}_e(x^{(j)}) \in \mathbb{R}^m$ for each prunable residual-stream edge $e$ and a remainder term $\mathbf{r}_C(x^{(j)}) \in \mathbb{R}^m$ such that*

$$\mathbf{z}^C(x^{(j)}) \triangleq g(\mathbf{x}_L^C(x^{(j)})) = \mathbf{z}(x^{(j)}) + \sum_{e \in E(C)} \mathbf{J}_e(x^{(j)}) + \mathbf{r}_C(x^{(j)}), \qquad j = 1, \ldots, N. \tag{21}$$

*Interpretation of $\mathbf{J}_e(x)$.* We may view each residual-stream edge $e$ as equipped with a continuous gate $\alpha_e \in [0,1]$ that scales its injected contribution. Then $\mathbf{J}_e(x)$ corresponds to the first-order change in the readout with respect to $\alpha_e$ around the reference execution, i.e., a Jacobian direction capturing the logit-level effect of retaining edge $e$.

*Let $D \triangleq Nm$ and define the stacked readouts*

$$\mathbf{Z} \triangleq [\mathbf{z}(x^{(1)}); \ldots; \mathbf{z}(x^{(N)})] \in \mathbb{R}^D, \qquad \mathbf{Z}^C \triangleq [\mathbf{z}^C(x^{(1)}); \ldots; \mathbf{z}^C(x^{(N)})] \in \mathbb{R}^D.$$

*Define the* edge signature *(edge contribution) vectors*

$$\mathbf{s}_e \triangleq [\mathbf{J}_e(x^{(1)}); \ldots; \mathbf{J}_e(x^{(N)})] \in \mathbb{R}^D, \tag{22}$$

*and the stacked remainder*

$$\mathbf{R}_C \triangleq [\mathbf{r}_C(x^{(1)}); \ldots; \mathbf{r}_C(x^{(N)})] \in \mathbb{R}^D.$$

*Then the circuit-pruned readout admits the* subset-sum form

$$\mathbf{Z}^C = \mathbf{Z} + \sum_{e \in E(C)} \mathbf{s}_e + \mathbf{R}_C. \tag{23}$$

*Consequence.* Eq. (23) reduces circuit non-uniqueness to a combinatorial collision problem over edge signatures. In particular, it suffices to find two distinct edge subsets $E_A \neq E_B$ such that

$$\left\| \sum_{e \in E_A} \mathbf{s}_e - \sum_{e \in E_B} \mathbf{s}_e \right\|_\infty \leq \delta. \tag{24}$$

Indeed, let $C_A, C_B$ be the circuits with $E(C_A) = E_A$ and $E(C_B) = E_B$. Subtracting (23) for $C_A$ and $C_B$ gives

$$\mathbf{Z}^{C_A} - \mathbf{Z}^{C_B} = \sum_{e \in E_A} \mathbf{s}_e - \sum_{e \in E_B} \mathbf{s}_e + (\mathbf{R}_{C_A} - \mathbf{R}_{C_B}), \tag{25}$$

$$\Rightarrow \quad \left\| \mathbf{Z}^{C_A} - \mathbf{Z}^{C_B} \right\|_\infty \leq \left\| \sum_{e \in E_A} \mathbf{s}_e - \sum_{e \in E_B} \mathbf{s}_e \right\|_\infty + \left\| \mathbf{R}_{C_A} - \mathbf{R}_{C_B} \right\|_\infty.$$

Combining (24) with (25) yields

$$\left\| \mathbf{Z}^{C_A} - \mathbf{Z}^{C_B} \right\|_\infty \leq \delta + \left\| \mathbf{R}_{C_A} - \mathbf{R}_{C_B} \right\|_\infty.$$

The remainder term will be explicitly controlled in Step 3 when invoking margin stability.

*Proof.* Fix $j \in \{1, \ldots, N\}$. By Assumption A.3, on inputs from $\mathcal{D}$ the effect of retaining residual-stream edges on the task-relevant readout is locally linear around the reference execution. Therefore there exist vectors $\{\mathbf{J}_e(x^{(j)})\}_e$ and a remainder $\mathbf{r}_C(x^{(j)})$ such that (21) holds.

Stacking the $N$ identities in (21) yields

$$\mathbf{Z}^C = \mathbf{Z} + \sum_{e \in E(C)} [\mathbf{J}_e(x^{(1)}); \ldots; \mathbf{J}_e(x^{(N)})] + \mathbf{R}_C.$$

By the definition of the edge signature vectors in (22), this is exactly the subset-sum representation (23). The final claim (24) then follows by applying the triangle inequality to (23). $\qquad\square$

In summary, each circuit $C$ is identified with its active edge set $E(C)$, and its task-relevant effect is (up to a small remainder) the subset sum $\sum_{e \in E(C)} \mathbf{s}_e$; thus it suffices to find two distinct edge subsets whose subset sums nearly coincide, which we do in the next step by a pigeonhole argument.

### Step 2: Quantisation and a Pigeonhole Collision

**Lemma A.6** (Quantisation–pigeonhole collision of subset sums). *Let $E = L(H + 1)$ be the number of prunable (non-skip) residual-stream edges, and fix a sparsity budget $s \in \{1, \ldots, E\}$. Let $\mathcal{C}_s \triangleq \{C : |E(C)| = s\}$ so that $|\mathcal{C}_s| = \binom{E}{s}$. Assume the stacked edge signatures $\{\mathbf{s}_e\}_{e \in \mathcal{E}} \subset \mathbb{R}^D$ satisfy the uniform bound*

$$\|\mathbf{s}_e\|_\infty \leq B \qquad \text{for all } e \in \mathcal{E}. \tag{26}$$

*For each circuit $C \in \mathcal{C}_s$, define its linearized subset-sum contribution*

$$\mathbf{S}(C) \triangleq \sum_{e \in E(C)} \mathbf{s}_e \in \mathbb{R}^D.$$

*Fix a resolution $\delta > 0$ and define the* grid-quantisation *map $Q_\delta : \mathbb{R}^D \to (\delta\mathbb{Z})^D$ coordinate-wise by*

$$\left(Q_\delta(\mathbf{v})\right)_i \triangleq \delta \cdot \text{round}\left(\frac{v_i}{\delta}\right), \qquad i = 1, \ldots, D, \tag{27}$$

*where $\text{round}(\cdot)$ rounds to the nearest integer (ties broken arbitrarily).*

*Then:*

1. *(**Bounded range**). For all $C \in \mathcal{C}_s$, $\mathbf{S}(C) \in [-sB, sB]^D$.*

2. *(**Finite number of bins**). The number of possible quantized values $\{Q_\delta(\mathbf{S}(C)) : C \in \mathcal{C}_s\}$ is at most*

$$K \leq \left(\left\lceil \frac{2sB}{\delta} \right\rceil + 1\right)^D. \tag{28}$$

3. *(**Collision**). If $\binom{E}{s} > K$, then there exist two distinct circuits $C_A \neq C_B$ in $\mathcal{C}_s$ such that*

$$\|\mathbf{S}(C_A) - \mathbf{S}(C_B)\|_\infty \leq \delta. \tag{29}$$

*Equivalently, letting $E_A \triangleq E(C_A)$ and $E_B \triangleq E(C_B)$,*

$$\left\|\sum_{e \in E_A} \mathbf{s}_e - \sum_{e \in E_B} \mathbf{s}_e\right\|_\infty \leq \delta.$$

*Proof.* **(1) Bounded range.** By the triangle inequality and (26), for any $C \in \mathcal{C}_s$,

$$\|\mathbf{S}(C)\|_\infty = \left\|\sum_{e \in E(C)} \mathbf{s}_e\right\|_\infty \leq \sum_{e \in E(C)} \|\mathbf{s}_e\|_\infty \leq sB,$$

hence $\mathbf{S}(C) \in [-sB, sB]^D$.

**(2) Finite number of bins.** Fix any coordinate $i$. Since $(\mathbf{S}(C))_i \in [-sB, sB]$, we have $(\mathbf{S}(C))_i/\delta \in [-sB/\delta, sB/\delta]$. The rounded integer $\mathrm{round}((\mathbf{S}(C))_i/\delta)$ therefore lies in an interval of integers of length at most $\lceil 2sB/\delta \rceil + 1$. Because quantisation is coordinate-wise, the total number of distinct grid points in $(\delta\mathbb{Z})^D$ hit by $Q_\delta(\mathbf{S}(C))$ is at most the product bound (28).

**(3) Collision and the rounding constant.** Define $\Phi : \mathcal{C}_s \to (\delta\mathbb{Z})^D$ by $\Phi(C) \triangleq Q_\delta(\mathbf{S}(C))$. If $\binom{E}{s} > |\mathrm{Im}(\Phi)|$ then by the pigeonhole principle there exist $C_A \neq C_B$ with $\Phi(C_A) = \Phi(C_B)$, i.e., $Q_\delta(\mathbf{S}(C_A)) = Q_\delta(\mathbf{S}(C_B))$.

We now show this equality implies (29) with *no hidden constant*. For any scalar $u \in \mathbb{R}$, by definition of rounding,

$$\left| u - \delta \cdot \mathrm{round}(u/\delta) \right| \leq \delta/2.$$

Applying this coordinate-wise to $\mathbf{S}(C_A)$ and $\mathbf{S}(C_B)$ and using that their quantizations coincide, we get for each coordinate $i$:

$$\left| (\mathbf{S}(C_A))_i - (\mathbf{S}(C_B))_i \right| \leq \left| (\mathbf{S}(C_A))_i - (Q_\delta(\mathbf{S}(C_A)))_i \right| + \left| (Q_\delta(\mathbf{S}(C_B)))_i - (\mathbf{S}(C_B))_i \right| \leq \delta/2 + \delta/2 = \delta.$$

Taking the maximum over $i$ yields $\|\mathbf{S}(C_A) - \mathbf{S}(C_B)\|_\infty \leq \delta$, proving (29). □

In summary, Step 2 establishes a combinatorial collision: because the number of $s$-edge circuits grows exponentially in the total edge count $E = L(H+1)$ while the $\delta$-quantized readout space has only polynomially many bins in $D$, two distinct circuits must produce nearly identical linearized readout effects.

### Step 3: From Readout Collision to Accuracy Faithfulness via Margin Stability

**Lemma A.7** (Margin preservation under $\ell_\infty$ logit perturbations). *Let $x$ be an input and let $\mathbf{z}(x) \in \mathbb{R}^m$ denote reference logits with predicted label $y^\star(x) \triangleq \arg\max_{k \in [m]} z_k(x)$. Define the (logit) margin at $x$ by*

$$\gamma(x) \triangleq z_{y^\star(x)}(x) - \max_{k \neq y^\star(x)} z_k(x). \tag{30}$$

*Assume a uniform margin $\gamma > 0$ on $\mathrm{supp}(\mathcal{D})$, i.e., $\gamma(x) \geq \gamma$ for all $x \in \mathrm{supp}(\mathcal{D})$. Let $\mathbf{z}'(x) \in \mathbb{R}^m$ satisfy*

$$\|\mathbf{z}'(x) - \mathbf{z}(x)\|_\infty \leq \rho. \tag{31}$$

*If $\rho < \gamma/2$, then $\arg\max_k z_k'(x) = \arg\max_k z_k(x)$. Consequently, if (31) holds for all $x \in \mathrm{supp}(\mathcal{D})$ with $\rho < \gamma/2$, then the alternative execution induces the same predictions as the reference on $\mathrm{supp}(\mathcal{D})$, and hence achieves the same accuracy on $\mathcal{D}$.*

*Proof.* Fix $x \in \mathrm{supp}(\mathcal{D})$ and write $y^\star \triangleq y^\star(x)$. For any $k$, (31) implies $z_k'(x) \in [z_k(x) - \rho, z_k(x) + \rho]$. Thus $z_{y^\star}'(x) \geq z_{y^\star}(x) - \rho$ and for any $k \neq y^\star$, $z_k'(x) \leq \max_{j \neq y^\star} z_j(x) + \rho$. Therefore,

$$z_{y^\star}'(x) - \max_{k \neq y^\star} z_k'(x) \geq \gamma(x) - 2\rho \geq \gamma - 2\rho.$$

If $\rho < \gamma/2$ then $\gamma - 2\rho > 0$, hence $y^\star$ remains the unique maximizer, proving the claim. □

Let $\mathbf{z}(x) \triangleq g(\mathbf{x}_L(x))$ denote the reference logits produced by the full model, and let $\mathbf{z}^C(x) \triangleq g(\mathbf{x}_L^C(x))$ denote the logits produced by a circuit-pruned execution.

**Linearisation With a Uniform Remainder Bound** By Assumption A.3, for each circuit $C$ and each $x \in \mathrm{supp}(\mathcal{D})$, there exists $\mathbf{r}_C(x)$ such that

$$\mathbf{z}^C(x) = \mathbf{z}(x) + \sum_{e \in E(C)} \mathbf{J}_e(x) + \mathbf{r}_C(x), \qquad \|\mathbf{r}_C(x)\|_\infty \leq \eta. \tag{32}$$

**Pairwise Logit Proximity Between Two Circuits** Let $C_1, C_2$ be the circuits returned by Step 2 with $E(C_1) = E_A$ and $E(C_2) = E_B$. Then for all $x \in \mathrm{supp}(\mathcal{D})$,

$$\|\mathbf{z}^{C_1}(x) - \mathbf{z}^{C_2}(x)\|_\infty \leq \delta + 2\eta. \tag{33}$$

**Logit Perturbation Relative to the Reference Model** Moreover, each $C_i \in \{C_1, C_2\}$ satisfies, uniformly for all $x \in \mathrm{supp}(\mathcal{D})$,

$$\|\mathbf{z}^{C_i}(x) - \mathbf{z}(x)\|_\infty \leq \delta + \eta \triangleq \delta_{\mathrm{tot}}. \tag{34}$$

**Accuracy Preservation via Margin Stability** If

$$\delta_{\mathrm{tot}} < \gamma/2, \tag{35}$$

then applying Lemma A.7 with $\mathbf{z}'(x) = \mathbf{z}^{C_i}(x)$ and $\rho = \delta_{\mathrm{tot}}$ yields

$$\arg\max_y \mathbf{z}^{C_i}(x)_y = \arg\max_y \mathbf{z}(x)_y, \qquad \forall x \in \mathrm{supp}(\mathcal{D}), \; i \in \{1, 2\}.$$

Consequently, when $\mathcal{M}$ is accuracy,

$$\mathcal{M}(F_\theta^{C_1}; \mathcal{D}) = \mathcal{M}(F_\theta; \mathcal{D}), \qquad \mathcal{M}(F_\theta^{C_2}; \mathcal{D}) = \mathcal{M}(F_\theta; \mathcal{D}),$$

i.e., both circuits are $\varepsilon$-faithful with $\varepsilon = 0$.

Step 3 converts a collision in the linearized readout space into exact task faithfulness. Specifically, Step 2 guarantees the existence of two distinct circuits whose linearized logit contributions differ by at most $\delta$, while Assumption A.3 controls the higher-order error by $\eta$. Under the margin stability condition $\delta_{\mathrm{tot}} = \delta + \eta < \gamma/2$, Lemma A.7 ensures that such bounded $\ell_\infty$ perturbations cannot change the predicted label. As a result, both circuits preserve the reference predictions on $\mathcal{D}$ and achieve the same accuracy as the full model, completing the faithfulness guarantee required in Theorem A.4.

**Step 4: Selecting a Low-IoU Colliding Pair** Step 2 guarantees the existence of a *collision* in the quantized readout space: two distinct circuits whose linearized subset sums fall into the same quantisation bin, hence are $\delta$-close in $\ell_\infty$. We now strengthen this guarantee by showing that, under an additional counting condition, one can choose the colliding pair to have *low overlap* (small IoU), i.e., to be structurally disparate.

**Lemma A.8** (Low-IoU pair inside a quantisation bin). *Fix an edge budget $s$ and consider the family $\mathcal{C}_s = \{C : |E(C)| = s\}$. Let $\mathcal{B} \subseteq \mathcal{C}_s$ be any quantisation bin induced by the map*

$$\Phi(C) \triangleq Q_\delta\left(\sum_{e \in E(C)} \mathbf{s}_e\right),$$

*and let $M \triangleq |\mathcal{B}|$.*

*For a target overlap threshold $\tau \in (0, 1)$, define*

$$t_\tau \triangleq \left\lfloor \frac{2\tau}{1 + \tau} s \right\rfloor, \tag{36}$$

*and the corresponding high-overlap neighborhood size upper bound*

$$V_\tau \triangleq \sum_{t=t_\tau+1}^{s} \binom{s}{t}\binom{E-s}{s-t}. \tag{37}$$

*If*

$$M > 1 + V_\tau, \tag{38}$$

*then there exist two distinct circuits $C_1, C_2 \in \mathcal{B}$ such that*

$$\mathrm{IoU}(C_1, C_2) = \frac{|E(C_1) \cap E(C_2)|}{|E(C_1) \cup E(C_2)|} \leq \tau. \tag{39}$$

*Moreover, since $C_1$ and $C_2$ lie in the same bin $\mathcal{B}$, their subset-sum contributions are $\delta$-close:*

$$\left\| \sum_{e \in E(C_1)} \mathbf{s}_e - \sum_{e \in E(C_2)} \mathbf{s}_e \right\|_\infty \leq \delta. \tag{40}$$

*Proof.* **Step 1: IoU as an intersection constraint.** For any two size-$s$ edge sets $A, B \subseteq [E]$, let $t \triangleq |A \cap B|$. Then

$$\text{IoU}(A, B) = \frac{|A \cap B|}{|A \cup B|} = \frac{t}{2s - t}.$$

Hence $\text{IoU}(A, B) \leq \tau$ is equivalent to $t \leq \frac{2\tau}{1+\tau} s$, and thus to the integer condition $t \leq t_\tau$ with $t_\tau$ defined in (36).

**Step 2: Bounding the high-overlap neighborhood.** Fix any $A \subseteq [E]$ with $|A| = s$ and define its $\tau$-overlap neighborhood

$$\mathcal{N}_\tau(A) \triangleq \{B \subseteq [E] : |B| = s, \ |A \cap B| \geq t_\tau + 1\}.$$

For any $t \in \{t_\tau + 1, \ldots, s\}$, the number of $B$ with $|A \cap B| = t$ is $\binom{s}{t}\binom{E-s}{s-t}$, since one chooses $t$ elements from $A$ and $s - t$ elements from $[E] \setminus A$. Summing over $t$ yields

$$|\mathcal{N}_\tau(A)| \leq \sum_{t=t_\tau+1}^{s} \binom{s}{t}\binom{E-s}{s-t} = V_\tau,$$

where $V_\tau$ is as in (37).

**Step 3: A packing argument inside the bin.** Pick any $C \in \mathcal{B}$ and set $A = E(C)$. Suppose, for contradiction, that every other $C' \in \mathcal{B} \setminus \{C\}$ satisfies $\text{IoU}(E(C), E(C')) > \tau$. By Step 1 this implies $|A \cap E(C')| \geq t_\tau + 1$, i.e., $E(C') \in \mathcal{N}_\tau(A)$. Therefore,

$$\mathcal{B} \subseteq \{C\} \cup \{C' \in \mathcal{C}_s : E(C') \in \mathcal{N}_\tau(A)\},$$

and hence

$$M = |\mathcal{B}| \leq 1 + |\mathcal{N}_\tau(A)| \leq 1 + V_\tau,$$

contradicting (38). Thus there exists some $C' \in \mathcal{B}$ with $\text{IoU}(C, C') \leq \tau$. Taking $(C_1, C_2) = (C, C')$ proves (39).

**Step 4: Collision radius within a bin.** Finally, since $C_1$ and $C_2$ are in the same bin, $\Phi(C_1) = \Phi(C_2)$. By the definition of the coordinate-wise rounding quantizer $Q_\delta$, equality of quantized coordinates implies that each coordinate differs by at most $\delta$, hence (40) holds. $\qquad\square$

**Lemma A.9** (Existence of a low-IoU collision pair)**.** *Under the setting of Step 2, the quantisation map $\Phi$ partitions $\mathcal{C}_s$ into at most*

$$K \ \leq \ \left(\left\lceil \frac{2sB}{\delta} \right\rceil + 1\right)^D \tag{41}$$

*bins (cf. Step 2). If*

$$\frac{\binom{E}{s}}{K} > 1 + V_\tau, \tag{42}$$

*then there exist two circuits $C_1 \neq C_2$ such that: (i) $\Phi(C_1) = \Phi(C_2)$ (hence their subset sums are $\delta$-close in $\ell_\infty$), and (ii) $\text{IoU}(C_1, C_2) \leq \tau$.*

*Proof.* Since $\Phi$ induces at most $K$ bins, by averaging there exists a bin $\mathcal{B}$ with

$$|\mathcal{B}| \geq \left\lceil \frac{|\mathcal{C}_s|}{K} \right\rceil = \left\lceil \frac{\binom{E}{s}}{K} \right\rceil.$$

If (42) holds, then in particular $|\mathcal{B}| > 1 + V_\tau$. Applying Lemma A.8 to this bin yields two circuits $C_1, C_2 \in \mathcal{B}$ with $\text{IoU}(C_1, C_2) \leq \tau$ and $\delta$-close subset sums, as claimed. $\qquad\square$

*Summary of Step 4.* Step 4 upgrades the mere existence of a collision (Step 2) to the existence of a *structurally disparate* collision pair: provided that the average bin occupancy $\binom{E}{s}/K$ exceeds the volume $1 + V_\tau$ of the high-overlap neighborhood in the constant-weight space, there exist two circuits in the same quantisation bin whose edge sets have IoU $\leq \tau$ (with $\tau \ll 1$) while their linearized readout contributions remain $\delta$-close. Combined with Step 3 (margin stability), this yields two $\varepsilon$-faithful circuits with IoU $\ll 1$, completing the non-uniqueness construction in Theorem A.4.

## B. Sources of Randomness in ACDC and EAP

ACDC (Conmy et al., 2023) operates in the reverse topological order to examine the importance of model edges based on the difference in KL divergence caused by the removal against a predefined threshold. Note that in every layer, the attention heads operate in parallel: There is no intrinsic priority ordering among attention heads. ACDC defaults to examine edges based on the increasing order of attention head index. Varying this does not significantly harm the quality of the circuits uncovered, but demonstrates the instability of ACDC which can be simply induced by merely reordering the indices of the attention heads in the same layer. EAP (Syed et al., 2024) estimates edge attribution using first-order approximation involving the task-specific metric. The only hyperparameter is the number of edges kept, $k$. We find that EAP generally uncovers much denser circuits: Using the same sparsity budget, the circuits perform much worse than those uncovered by ACDC. Moreover, varying only the names in the IOI task yields structurally different (based on the IoU metric) circuits. It is then concluded that EAP is sensitive to batching and task-irrelevant information like names of the same tokenization length.

## C. Sources of Randomness in Edge Pruning and DiscoGP

Edge Pruning (Bhaskar et al., 2024) models the edge mask $z$ using the following computations:

$$\mathbf{u} \sim \mathrm{Uniform}(\epsilon, 1 - \epsilon) \tag{43}$$

$$\mathbf{s} = \sigma\left(\frac{1}{\beta} \cdot \log \frac{\mathbf{u}}{1 - \mathbf{u}} + \log \boldsymbol{\alpha}\right) \tag{44}$$

$$\tilde{\mathbf{s}} = \mathbf{s} \times (r - l) + l \tag{45}$$

$$\mathbf{z} = \min\big(1, \max(0, \tilde{\mathbf{s}})\big) \tag{46}$$

where $\sigma$ is the sigmoid function, $\epsilon$ is a hyperparameter, $\frac{1}{\beta}$ is the temperature hyperparameter. The key point to pay attention to is the stochastic noise $\mathbf{u}$ introduced for every edge $e \in E$. This sampling is part of the intrinsic randomness of EP, in addition to the initialisation of the log alphas $\log \boldsymbol{\alpha}$, which are the learnable parameters. EP is stochastic by design to allow differentiable masking. The similar situation applies to DiscoGP. DiscoGP models the edge mask $m_e$ with the following:

$$s_e = \sigma\left(\frac{l_e - \log \frac{\log \mathcal{U}_1}{\log \mathcal{U}_2}}{\tau}\right), \tag{47}$$

where $\tau \in (0, \inf)$ is a temperature hyperparameter, $l_e$ is a learnable parameter of a sigmoid function $\sigma(\cdot)$, and $\mathcal{U}_1, \mathcal{U}_2 \sim$ Uniform$(0, 1)$ are random variables drawn from a uniform distribution. The straight-through estimator (Bengio et al., 2013) is used to convert the sampled $s_e$ into a binary mask variable:

$$m_e = [\mathbb{1}_{s_e > 0.5} - s_e]_{\mathrm{detach}} + s_e, \tag{48}$$

The source of randomness comes from the sampling of $\mathcal{U}_1, \mathcal{U}_2$ and also the initialisation of $l_e$ for every edge $e \in E$. DiscoGP is similarly stochastic by design.

## D. Node-Level Overlap

Our main analysis measures structural overlap at the edge level, since the discovered mechanisms are defined as edge-selected subgraphs in the residual-stream computation graph. Nevertheless, one may ask whether low edge-level overlap simply reflects different wiring among largely shared nodes. To examine this, we also compute node-level intersection-over-union (IoU) for the paired sheaves discovered by OASR.

As shown in Table 8, node-level overlap is substantially higher than edge-level overlap. This is expected because our discovery objective does not explicitly optimise for node sparsity, and individual sheaves can already contain a large fraction of nodes. However, edge-level IoU remains consistently low, indicating that the discovered mechanisms differ primarily in how information flows between components rather than merely in which components appear.

*Table 8.* Node-level and edge-level overlap between paired OASR-discovered sheaves. Node-level overlap is higher because the objective optimises edge sparsity rather than node sparsity, but edge-level overlap remains low across tasks.

| Task | Node IoU | Node D. A | Node D. B | Edge IoU |
|------|----------|-----------|-----------|----------|
| IOI | 64.2% | 67.7% | 73.0% | 4.1% |
| BLiMP | 66.9% | 76.7% | 77.1% | 5.1% |
| AGA | 62.1% | 62.8% | 76.0% | 6.2% |
| ANA | 61.4% | 66.4% | 74.7% | 5.3% |
| DNA | 55.3% | 57.9% | 54.2% | 5.8% |
| DNA i | 49.0% | 51.1% | 60.4% | 6.2% |
| DNA a | 67.2% | 71.5% | 66.7% | 7.5% |
| DNA ia | 71.8% | 76.2% | 64.8% | 6.4% |
| Docstring | 93.0% | 89.0% | 88.0% | 11.0% |

## E. Constructing Intersection-Derived Candidate Sheaves

In Section 4, we construct compact candidate sheaves by taking intersections of edge sets across independently discovered IOI sheaves. We emphasise that this construction is used only for the intersection-derived compact sheaf analysis, and not for computing the IoU statistics reported in the main non-uniqueness experiments.

Given a collection of discovered sheaves with edge sets $E_1, \ldots, E_m$, we first compute their edge-wise intersection

$$E_\cap = \bigcap_{i=1}^{m} E_i.$$

Since an arbitrary edge-wise intersection need not itself form a connected computation graph, we then retain only edges that lie on at least one directed path from the input node to the output node. Formally, if $G_\cap = (V, E_\cap)$ is the subgraph induced by the intersected edge set, we define

$$E_\text{path} = \{e \in E_\cap : e \text{ lies on a directed path from Input to Output in } G_\cap\}.$$

The resulting path-connected subgraph $G_\text{path} = (V, E_\text{path})$ is then evaluated under the same zero-ablation protocol used for sheaf evaluation. If $E_\text{path}$ is empty, the intersection is treated as containing no valid candidate sheaf.

This path filtering step is intended only to ensure that the intersection-derived candidate corresponds to a valid information-flow subgraph. It should not be interpreted as an additional discovery objective, and it is not applied when measuring edge overlap between independently discovered mechanisms.

## F. Worst-case Sheaf Results using DiscoGP

To ensure all the sheaves discovered are of satisfactory quality, we report the worst-case statistics in Table 9. All sheaves uncovered by DiscoGP with different random seeds with and without OASR are faithful and compact based on task performance and sheaf sparsity. Even in the worst seed across all tasks, DiscoGP recovers circuits that retain high accuracy ($\geq 82\%$, which is close to the worst performance obtained by baseline using Random Init at $84\%$) while using at most $4.3\%$ of edges, demonstrating that functional faithfulness and compactness are robust to both the random initialisation and the case with the additional OASR.

Overall, these worst-case results indicate that functional faithfulness and sparsity are not fragile artefacts of lucky random seeds, but stable properties of DiscoGP. This supports our central claim that multiple sparse, task-faithful circuits can reliably emerge despite randomness and additional constraints, e.g., the OASR.

*Table 10.* **Evaluation of two sheaves discovered for common benchmark tasks.** Here, the sheaves are discovered with different random seeds and without OASR.

| Task | Sheaf $A$ Acc. | Sheaf $B$ Acc. | IoU$(A, B)$ |
|---|---|---|---|
| IOI | 100.0% | 100.0% | 18.5% |
| BLiMP | 99.0% | 98.8% | 26.4% |
| AGA | 97.3% | 97.3% | 21.0% |
| ANA | 99.3% | 98.7% | 21.1% |
| DNA | 99.2% | 99.2% | 22.8% |
| DNA i | 100.0% | 100.0% | 19.7% |
| DNA a | 99.2% | 98.5% | 23.5% |
| DNA ia | 99.0% | 99.0% | 25.9% |
| Docstring | 100.0% | 100.0% | 21.3% |

*Table 9.* Worst-case circuit statistics for circuit discovery results using DiscoGP on GPT-2 Small. We report the minimum task accuracy across runs and the densest discovered circuit, measured by maximum edge density and maximum edge count.

| Task | Method | Worst-Case Metrics | | |
|---|---|---|---|---|
| | | Min Acc. (%) | Max Edge D. (%) | Max $\#E$ |
| IOI | Random Init | 99.70 | 3.484 | 1132 |
| | OASR | 98.80 | 3.490 | 1134 |
| BLiMP | Random Init | 95.30 | 3.133 | 1018 |
| | OASR | 94.70 | 2.847 | 925 |
| AGA | Random Init | 90.67 | 2.850 | 926 |
| | OASR | 90.00 | 3.376 | 1097 |
| ANA | Random Init | 92.67 | 2.718 | 883 |
| | OASR | 89.33 | 2.247 | 730 |
| DNA | Random Init | 90.98 | 1.927 | 626 |
| | OASR | 90.23 | 1.927 | 626 |
| DNA i | Random Init | 94.00 | 2.074 | 674 |
| | OASR | 91.00 | 1.600 | 520 |
| DNA a | Random Init | 91.73 | 2.512 | 816 |
| | OASR | 87.22 | 4.260 | 1384 |
| DNA ia | Random Init | 84.00 | 3.016 | 980 |
| | OASR | 82.00 | 3.032 | 985 |
| Docstring | Random Init | 97.10 | 3.924 | 1275 |
| | OASR | 95.60 | 3.672 | 1193 |

## G. Pairwise IoU with Only Random Init

By varying random seeds during the sheaf discovery process in DiscoGP, we establish a baseline for consistency in sheaf identification. The results are reported in Table 10. Even without OASR, we observe that pairwise intersection-over-union (IoU) scores across all tasks remain substantially higher than would be expected from random edge selection. From an optimisation perspective, these results suggest that while gradient-based pruning can identify faithful subgraphs, a naive approach may fall into an "illusion of algorithmic consistency." Without a repulsion mechanism, standard optimisation fails to capture the internal competing components that exhibit functional equivalence. Consequently, the issue of sheaf multiplicity is easily overlooked when only a small number of random seeds are evaluated, as an IoU of approximately 20% may be misinterpreted as algorithmic stability, given the gradient-based nature of the method.

## H. DiscoGP with OASR Results on Pythia-160M

We report similarly the sheaf discovery results on Pythia-160M. The same pattern emerges in Pythia-160M: Using OASR yields more structurally distinct circuits, despite similar task performance and sparsity levels. The cumulative intersection continues to shrink more aggressively compared to the baseline by varying only the random seed. This empirical evidence

*Table 11.* **Mutual intersection across circuits in Pythia-160M.** We report the size of the total intersection and union across 20 discovery runs, together with their ratio (Mutual IoU), as well as the average quality of the resulting sheaves.

| Task | Method | Mutual Intersection | | | Average Sheaf Quality | | | |
|---|---|---|---|---|---|---|---|---|
| | | $|E_\cap|$ | $|E_\cup|$ | Mutual IoU | Edge D. | $\#E$ | Acc. (%) | Comp. (%) |
| IOI | Random Init | 50 | 11112 | 0.45% | 7.09% | 2303 | 80.10 | 47.01 |
| | OASR | 39 | 11470 | 0.34% | 6.82% | 2215 | 79.63 | 47.81 |
| BLiMP | Random Init | 56 | 10722 | 0.52% | 7.36% | 2393 | 85.35 | 48.32 |
| | OASR | 35 | 10795 | 0.32% | 6.76% | 2197 | 84.62 | 49.98 |
| AGA | Random Init | 9 | 7740 | 0.12% | 4.86% | 1578 | 93.26 | 65.45 |
| | OASR | 4 | 7288 | 0.05% | 3.87% | 1257 | 93.08 | 69.02 |
| ANA | Random Init | 44 | 7710 | 0.57% | 5.87% | 1906 | 84.29 | 45.89 |
| | OASR | 20 | 7815 | 0.26% | 4.84% | 1573 | 84.52 | 48.39 |
| DNA | Random Init | 100 | 8366 | 1.20% | 6.67% | 2168 | 80.93 | 48.93 |
| | OASR | 64 | 8336 | 0.77% | 5.74% | 1865 | 80.21 | 48.14 |
| DNA i | Random Init | 42 | 7064 | 0.59% | 5.80% | 1884 | 81.86 | 34.90 |
| | OASR | 20 | 7361 | 0.27% | 4.82% | 1567 | 82.65 | 34.61 |
| DNA a | Random Init | 125 | 7558 | 1.65% | 6.52% | 2118 | 85.22 | 45.97 |
| | OASR | 87 | 7807 | 1.11% | 5.88% | 1910 | 84.78 | 46.87 |
| DNA ia | Random Init | 31 | 6709 | 0.46% | 5.31% | 1726 | 93.52 | 42.96 |
| | OASR | 15 | 6842 | 0.22% | 4.29% | 1394 | 93.06 | 42.31 |
| Docstring | Random Init | 28 | 6935 | 0.40% | 3.28% | 1067 | 82.56 | 55.13 |
| | OASR | 5 | 8303 | 0.06% | 3.81% | 1237 | 80.94 | 53.05 |

validates that the phenomenon is not specific to a particular transformer model (e.g. GPT-2 Small). Similar to the results reported in Table 3, with OASR, DiscoGP continues to explore more edges across most of the tasks, while having a much smaller cumulative intersection across multiple runs. This empirically supports that the argument of multiple distinct computational pathways supporting the same behaviour is not unique to a particular transformer language model.

## I. Experimental Results on ACDC and EAP

Following Conmy et al. (2023), we run ACDC under both interchange ablation and zero ablation with the result summarised in Table 4. The only source of randomness is the ordering of nodes during the backward search: while layers are always traversed in decreasing order, we allow the order of attention heads within each layer to vary. Despite using the same hyperparameter $\tau$, this induces substantial variation among the discovered circuits, even though ACDC is typically regarded as a largely deterministic circuit discovery method. There is no explicit source of randomness in EAP. However, by only varying the names in the task of IOI, we perform multiple runs of EAP to reveal its sensitivity to task-irrelevant information, i.e., the names appearing in the IOI dataset. Despite being extremely efficient using only two forward passes and one backward pass, the circuits uncovered by EAP perform relatively poorly on the IOI task, as shown in Figure 3.

## J. Sample Prompts from the Datasets

Example prompts from different datasets, together with their corresponding correct and incorrect continuations, are reported in Figure 7. Task-specific performance is evaluated by checking whether the circuit (or sheaf) assigns a higher probability to the correct continuation than to the incorrect one.

**Indirect Object Identification (IOI)** This task was formulated and introduced by Wang et al. (2022a). The sentences in the IOI task have two clauses, where the initial clause introduces the subject (S) and the indirect object (IO), followed by the main clause referring to S and IO again. Depending on the type of IOI template (BABA or ABBA), in the main clause, there are two cases for the order of S and IO. Throughout this paper, we refer to IOI as an equal mix of IOI_BABA and IOI_ABBA. Using the logit difference metric, GPT-2 Small performs near perfectly on the IOI task. We filter the dataset such that the base model can perform perfectly. The specific IOI templates are given in Table 12 with the candidate names in Table 13.

*Table 12.* Sentence templates for generating the IOI dataset.

| |
|---|
| Then, [B] and [A] went to the [PLACE]. [B] gave a [OBJECT] to [A] |
| Then, [B] and [A] had a lot of fun at the [PLACE]. [B] gave a [OBJECT] to [A] |
| Then, [B] and [A] were working at the [PLACE]. [B] decided to give a [OBJECT] to [A] |
| Then, [B] and [A] were thinking about going to the [PLACE]. [B] wanted to give a [OBJECT] to [A] |
| Then, [B] and [A] had a long argument, and afterwards [B] said to [A] |
| After [B] and [A] went to the [PLACE], [B] gave a [OBJECT] to [A] |
| When [B] and [A] got a [OBJECT] at the [PLACE], [B] decided to give it to [A] |
| When [B] and [A] got a [OBJECT] at the [PLACE], [B] decided to give the [OBJECT] to [A] |
| While [B] and [A] were working at the [PLACE], [B] gave a [OBJECT] to [A] |
| While [B] and [A] were commuting to the [PLACE], [B] gave a [OBJECT] to [A] |
| After the lunch, [B] and [A] went to the [PLACE]. [B] gave a [OBJECT] to [A] |
| Afterwards, [B] and [A] went to the [PLACE]. [B] gave a [OBJECT] to [A] |
| Then, [B] and [A] had a long argument. Afterwards [B] said to [A] |
| The [PLACE] [B] and [A] went to had a [OBJECT]. [B] gave it to [A] |
| Friends [B] and [A] found a [OBJECT] at the [PLACE]. [B] gave it to [A] |

*Table 13.* Candidate infilling words of IOI sentence templates.

| Placeholder Type | Candidate Infilling Words |
|---|---|
| [A] and [B] (names) | Michael, Christopher, Jessica, Matthew, Ashley, Jennifer, Joshua, Daniel, David, James, Robert, John, Joseph, Andrew, Ryan, Brandon, Justin, Sarah, William, Jonathan, Stephanie, Brian, Nicole, Nicholas, Heather, Eric, Elizabeth, Adam, Megan, Melissa, Kevin, Steven, Timothy, Christina, Kyle, Rachel, Laura, Lauren, Amber, Brittany, Richard, Kimberly, Jeffrey, Amy, Crystal, Michelle, Tiffany, Jeremy, Mark, Emily, Aaron, Charles, Rebecca, Jacob, Stephen, Patrick, Kelly, Samantha, Nathan, Sara, Dustin, Paul, Angela, Tyler, Scott, Andrea, Gregory, Erica, Mary, Travis, Lisa, Kenneth, Bryan, Lindsey, Jose, Alexander, Jesse, Katie, Lindsay, Shannon, Vanessa, Courtney, Alicia, Cody, Allison, Bradley, Samuel. |
| [PLACE] | store, garden, restaurant, school, hospital, office, house, station. |
| [OBJECT] | ring, kiss, bone, basketball, computer, necklace, drink, snack. |

---

**IOI_BABA**

**Prompt:** Then, Brian and Justin had a long argument. Afterwards Brian said to
**Correct Answer:** Justin, **Incorrect Answer:** Brian

---

**IOI_ABBA**

**Prompt:** Friends Jason and Dustin found a necklace at the office. Dustin gave it to
**Correct Answer:** Jason, **Incorrect Answer:** Dustin

---

**AGA**

**Prompt:** Margaret is describing
**Correct Answer:** herself, **Incorrect Answer:** himself

---

**ANA**

**Prompt:** Ruth wasn't discussing
**Correct Answer:** herself, **Incorrect Answer:** themselves

---

**DNA**

**Prompt:** This woman sees these
**Correct Answer:** eyes, **Incorrect Answer:** eye

---

**DNA a**

**Prompt:** Sara hasn't bored this displeased
**Correct Answer:** senator, **Incorrect Answer:** senators

---

**DNA i**

**Prompt:** Guy wouldn't argue about that
**Correct Answer:** hypothesis, **Incorrect Answer:** hypotheses

---

**DNA ia**

**Prompt:** Broccoli astounds this unemployed
**Correct Answer:** child, **Incorrect Answer:** children

---

**Docstring**

**Prompt:** `def fields(self, count, version):`
`    """address release rule object connection plant port`
`    :param count:  account determination stock`
`    :param`
**Correct Answer:** `version`, **Incorrect Answer:** `count`

---

*Figure 7.* Example prompts from each dataset together with their correct and incorrect continuations.

**BLiMP**   BLiMP (Warstadt et al., 2020) consists of individual datasets about phenomena in syntax, morphology, and semantics. Each of the datasets contain minimally different sentence pairs that contrast in grammatical acceptability. BLiMP was initially designed for bidirectional models such as BERT, as a consequence, the sentence pairs do not necessarily differ in the last token position. Excessive prompt augmentation would defeat the purpose of CSD, and so we select the 6 BLiMP paradigms that would differ only in the last token position, which are applicable to unidirectional decoder-only LMs.

**Docstring**   The prompts in the Docstring task are Python code, in particular, function signatures to be completed (Heimersheim & Janiak, 2023). The next token prediction in this task is about predicting variable names previously declared, immediately after `:param`. The goal of this task is to measure the model's ability to map code structure to natural language descriptions. Compared to the IOI task, this requires the sheaf to handle additional tokens involved in the structured text.

## K. Results for Individual Sheaves

The IOI sheaves discovered by the original DiscoGP, as shown in Table 14, ultimately shrinks to a final intersection $E_\cap$ of 20 edges, with the final cumulative union $E_\cup$ being of 6560 edges. The edge densities are consistently below 3.5%, indicating that DiscoGP is successful in uncovering functional and compact sheaves for the IOI task. With OASR, DiscoGP explores more edges and converges to a smaller intersection of seemingly essential components. Across the remaining BLiMP subtasks, as reported from Table 15 to 21, and the Docstring task, as shown in Table 22, we observe the same qualitative pattern. In all cases, successive runs produce sheaves with comparable sparsity (typically in the 1–4% edge density range) and high task accuracy, while the cumulative union grows steadily and the cumulative intersection shrinks to a relatively small core. This indicates that many distinct, low-overlap circuits can independently realise the same function, and that only a small subset of edges is consistently reused across runs. Introducing OASR systematically accelerates this diversification: the size of $E_\cup$ increases more rapidly, while $E_\cap$ converges to a smaller set of edges, without a significant loss in accuracy or completeness. Taken together, these results suggest that DiscoGP does not recover a unique minimal circuit for a task, but rather a family of functionally equivalent, sparse realizations whose shared intersection can be interpreted as a set of highly robust or essentially indispensable components, and whose large union reflects the substantial degree of functional redundancy and non-identifiability in the underlying network. The extra edges can also be seen as additional noise that may or may not contribute to the functionality of the sheaf.

## L. Per-Layer Incoming Edge Counts

Figure 8 reports the per-layer incoming edge counts for attention and MLP components of two low-IoU sheaf pairs for the BLiMP task. The two sheaves exhibit broadly similar high-level layer-wise trends, such as concentration of attention edges in middle layers and increased MLP usage toward later layers, yet differ substantially in the precise allocation of edges across layers. For the BLiMP, Sheaf A shows a pronounced peak in attention incoming edges around layers 6–7, whereas Sheaf B distributes attention more evenly, with relatively higher density in later layers. In the MLP blocks, Sheaf A places more mass in mid-layers and exhibits an abrupt drop in the final layer, while Sheaf B allocates a larger fraction of edges to the top layers. Despite these structural discrepancies, both circuits achieve comparable task performance, indicating that similar syntactic behaviour can be supported by markedly different depth-wise routing patterns.

A similar phenomenon appears for other BLiMP subtasks. The pairs of sheaves mostly agree qualitatively on the overall shape of attention and MLP utilisation, but differ in where peak connectivity occurs and how sharply it decays across layers.

For the Docstring sheaves, as shown in Figure 8 although the intersection-over-union (IoU) is only about 0.11, the per-layer edge count distributions exhibit highly similar overall shapes for both the attention and MLP components, aside from a few mild localised spikes. This suggests that, for processing the structured patterns in the Docstring task, there may exist an implicit layer-wise preference for where the relevant computation is carried out.

To examine the fine-grained structure of the computational graphs induced by the sheaves, we visualize representative sheaf pairs for various tasks as given in Figures 9, 10, 11. Despite structural differences, these pairs exhibit near-identical task performance, providing clear empirical evidence of functional equivalence under substantial structural disparity.

*Table 14.* IOI: Per-sheaf cumulative overlap and sheaf quality (Random Init and OASR).

**IOI – Random Init**

| ID | Cumulative Statistics | | Sheaf Quality | | | |
|---|---|---|---|---|---|---|
| | $|E_\cap|$ | $|E_\cup|$ | Edge D. (%) | #E | Acc. (%) | Comp. (%) |
| 0 | 897 | 897 | 2.76% | 897 | 100.00 | 42.80 |
| 1 | 292 | 1576 | 2.99% | 971 | 100.00 | 45.30 |
| 2 | 128 | 2183 | 2.98% | 967 | 100.00 | 45.90 |
| 3 | 98 | 2709 | 3.28% | 1067 | 99.90 | 45.70 |
| 4 | 64 | 3089 | 2.83% | 921 | 99.90 | 45.50 |
| 5 | 47 | 3465 | 2.84% | 924 | 99.90 | 45.40 |
| 6 | 41 | 3826 | 3.22% | 1046 | 100.00 | 45.80 |
| 7 | 38 | 4200 | 3.43% | 1114 | 100.00 | 46.30 |
| 8 | 36 | 4467 | 2.97% | 964 | 100.00 | 46.80 |
| 9 | 32 | 4752 | 3.26% | 1058 | 100.00 | 45.50 |
| 10 | 31 | 4973 | 3.12% | 1013 | 100.00 | 44.60 |
| 11 | 31 | 5224 | 3.48% | 1132 | 99.80 | 46.20 |
| 12 | 30 | 5488 | 3.39% | 1102 | 99.90 | 45.20 |
| 13 | 29 | 5620 | 2.55% | 827 | 99.90 | 46.20 |
| 14 | 28 | 5760 | 2.76% | 897 | 99.70 | 45.90 |
| 15 | 23 | 5989 | 3.15% | 1023 | 100.00 | 45.30 |
| 16 | 23 | 6153 | 2.97% | 966 | 100.00 | 46.40 |
| 17 | 22 | 6367 | 3.43% | 1116 | 100.00 | 45.40 |
| 18 | 21 | 6478 | 2.87% | 931 | 100.00 | 45.80 |
| 19 | 20 | 6560 | 2.48% | 805 | 100.00 | 46.70 |

**IOI – OASR**

| ID | Cumulative Statistics | | Sheaf Quality | | | |
|---|---|---|---|---|---|---|
| | $|E_\cap|$ | $|E_\cup|$ | Edge D. (%) | #E | Acc. (%) | Comp. (%) |
| 0 | 918 | 918 | 2.83% | 918 | 99.40 | 45.80 |
| 1 | 213 | 1654 | 2.92% | 949 | 99.10 | 45.60 |
| 2 | 105 | 2240 | 2.81% | 912 | 98.80 | 45.50 |
| 3 | 59 | 2845 | 3.14% | 1020 | 99.20 | 45.70 |
| 4 | 41 | 3399 | 3.28% | 1067 | 100.00 | 45.30 |
| 5 | 33 | 3710 | 2.28% | 742 | 99.70 | 46.00 |
| 6 | 31 | 3944 | 2.12% | 690 | 100.00 | 45.60 |
| 7 | 30 | 4283 | 2.80% | 911 | 99.60 | 45.10 |
| 8 | 25 | 4672 | 3.36% | 1093 | 100.00 | 46.00 |
| 9 | 25 | 4856 | 2.11% | 684 | 99.10 | 44.70 |
| 10 | 17 | 5328 | 3.45% | 1120 | 100.00 | 46.10 |
| 11 | 15 | 5590 | 2.84% | 922 | 99.20 | 46.90 |
| 12 | 14 | 5923 | 3.20% | 1040 | 100.00 | 46.10 |
| 13 | 13 | 6205 | 3.42% | 1112 | 100.00 | 47.90 |
| 14 | 13 | 6504 | 3.49% | 1134 | 98.90 | 46.00 |
| 15 | 11 | 6732 | 3.04% | 988 | 100.00 | 46.10 |
| 16 | 11 | 6905 | 2.65% | 861 | 99.90 | 45.80 |
| 17 | 11 | 7106 | 2.61% | 847 | 99.90 | 45.70 |
| 18 | 11 | 7241 | 2.27% | 736 | 99.80 | 45.80 |
| 19 | 11 | 7382 | 2.54% | 824 | 99.20 | 45.60 |

*Table 15.* BLiMP: Per-sheaf cumulative overlap and sheaf quality (Random Init and OASR).

**BLiMP – Random Init**

| ID | Cumulative Statistics | | Sheaf Quality | | | |
|---|---|---|---|---|---|---|
| | $|E_\cap|$ | $|E_\cup|$ | Edge D. (%) | #E | Acc. (%) | Comp. (%) |
| 0 | 885 | 885 | 2.72% | 885 | 98.10 | 50.40 |
| 1 | 311 | 1323 | 2.31% | 749 | 97.30 | 43.30 |
| 2 | 192 | 1837 | 2.83% | 920 | 95.50 | 43.80 |
| 3 | 143 | 2148 | 2.48% | 806 | 96.40 | 43.90 |
| 4 | 116 | 2547 | 3.13% | 1018 | 96.60 | 43.80 |
| 5 | 105 | 2785 | 2.64% | 857 | 99.00 | 47.80 |
| 6 | 99 | 2955 | 2.31% | 751 | 98.80 | 47.50 |
| 7 | 89 | 3136 | 2.51% | 816 | 97.90 | 49.90 |
| 8 | 82 | 3361 | 2.88% | 937 | 97.00 | 42.70 |
| 9 | 72 | 3539 | 2.65% | 862 | 95.30 | 43.60 |
| 10 | 67 | 3734 | 2.55% | 830 | 96.90 | 42.80 |
| 11 | 63 | 3874 | 2.59% | 840 | 97.60 | 42.90 |
| 12 | 62 | 4015 | 2.71% | 879 | 97.40 | 43.30 |
| 13 | 58 | 4179 | 2.70% | 878 | 96.60 | 44.00 |
| 14 | 56 | 4323 | 2.90% | 941 | 97.60 | 43.40 |
| 15 | 53 | 4413 | 2.32% | 753 | 96.70 | 45.00 |
| 16 | 53 | 4516 | 2.44% | 792 | 97.60 | 43.50 |
| 17 | 52 | 4582 | 2.17% | 705 | 98.00 | 42.90 |
| 18 | 52 | 4750 | 2.91% | 946 | 97.20 | 43.90 |
| 19 | 50 | 4858 | 2.69% | 875 | 97.70 | 47.20 |

**BLiMP – OASR**

| ID | Cumulative Statistics | | Sheaf Quality | | | |
|---|---|---|---|---|---|---|
| | $|E_\cap|$ | $|E_\cup|$ | Edge D. (%) | #E | Acc. (%) | Comp. (%) |
| 0 | 885 | 885 | 2.72% | 885 | 98.10 | 50.40 |
| 1 | 258 | 1405 | 2.39% | 778 | 96.50 | 43.80 |
| 2 | 154 | 1953 | 2.85% | 925 | 97.30 | 43.10 |
| 3 | 108 | 2357 | 2.50% | 812 | 95.20 | 43.40 |
| 4 | 91 | 2689 | 2.45% | 795 | 95.40 | 49.60 |
| 5 | 77 | 2980 | 2.51% | 817 | 97.00 | 46.30 |
| 6 | 70 | 3154 | 2.07% | 673 | 96.50 | 54.00 |
| 7 | 66 | 3436 | 2.54% | 825 | 96.20 | 44.90 |
| 8 | 65 | 3610 | 2.18% | 708 | 96.20 | 43.80 |
| 9 | 57 | 3836 | 2.45% | 797 | 95.50 | 55.50 |
| 10 | 53 | 4007 | 2.22% | 720 | 95.20 | 43.00 |
| 11 | 49 | 4215 | 2.49% | 808 | 95.60 | 44.80 |
| 12 | 46 | 4388 | 2.04% | 663 | 95.80 | 46.50 |
| 13 | 44 | 4569 | 2.28% | 742 | 96.80 | 43.40 |
| 14 | 44 | 4717 | 2.22% | 722 | 96.30 | 43.70 |
| 15 | 41 | 4896 | 2.35% | 763 | 97.10 | 43.90 |
| 16 | 39 | 4979 | 2.07% | 673 | 96.60 | 44.00 |
| 17 | 37 | 5081 | 2.10% | 681 | 95.10 | 42.50 |
| 18 | 37 | 5175 | 2.00% | 650 | 94.70 | 44.50 |
| 19 | 37 | 5289 | 2.27% | 737 | 95.20 | 43.80 |

*Table 16.* AGA: Per-sheaf cumulative overlap and sheaf quality (Random Init and OASR).

**AGA – Random Init**

| ID | Cumulative Statistics | | Sheaf Quality | | | |
|---|---|---|---|---|---|---|
| | $|E_\cap|$ | $|E_\cup|$ | Edge D. (%) | #E | Acc. (%) | Comp. (%) |
| 0 | 838 | 838 | 2.58% | 838 | 93.00 | 41.00 |
| 1 | 237 | 1410 | 2.49% | 809 | 95.00 | 38.00 |
| 2 | 132 | 1938 | 2.73% | 886 | 93.00 | 61.00 |
| 3 | 81 | 2277 | 2.30% | 748 | 96.00 | 60.00 |
| 4 | 59 | 2588 | 2.46% | 800 | 93.00 | 52.00 |
| 5 | 49 | 2859 | 2.42% | 786 | 96.00 | 74.00 |
| 6 | 44 | 3150 | 2.72% | 884 | 94.00 | 59.00 |
| 7 | 38 | 3339 | 2.26% | 734 | 94.00 | 28.00 |
| 8 | 35 | 3524 | 2.44% | 793 | 92.00 | 34.00 |
| 9 | 29 | 3710 | 2.39% | 776 | 92.00 | 46.00 |
| 10 | 27 | 3854 | 2.37% | 771 | 95.00 | 37.00 |
| 11 | 24 | 3952 | 2.12% | 688 | 92.00 | 27.00 |
| 12 | 23 | 4038 | 2.10% | 683 | 93.00 | 27.00 |
| 13 | 22 | 4119 | 1.86% | 603 | 95.00 | 27.00 |
| 14 | 22 | 4187 | 1.95% | 635 | 93.00 | 58.00 |
| 15 | 22 | 4279 | 2.39% | 777 | 96.00 | 43.00 |
| 16 | 22 | 4415 | 2.85% | 926 | 95.00 | 41.00 |
| 17 | 22 | 4549 | 2.78% | 904 | 96.00 | 41.00 |
| 18 | 22 | 4692 | 2.75% | 895 | 94.00 | 45.00 |
| 19 | 21 | 4758 | 2.36% | 768 | 95.00 | 43.00 |

**AGA – OASR**

| ID | Cumulative Statistics | | Sheaf Quality | | | |
|---|---|---|---|---|---|---|
| | $|E_\cap|$ | $|E_\cup|$ | Edge D. (%) | #E | Acc. (%) | Comp. (%) |
| 0 | 838 | 838 | 2.58% | 838 | 93.00 | 41.00 |
| 1 | 175 | 1268 | 1.86% | 605 | 95.00 | 45.00 |
| 2 | 79 | 1774 | 2.34% | 761 | 93.00 | 28.00 |
| 3 | 60 | 2151 | 2.07% | 671 | 94.00 | 59.00 |
| 4 | 36 | 2530 | 2.16% | 702 | 91.00 | 43.00 |
| 5 | 33 | 2887 | 2.28% | 741 | 95.00 | 34.00 |
| 6 | 29 | 3201 | 2.32% | 755 | 94.00 | 42.00 |
| 7 | 26 | 3471 | 2.07% | 674 | 89.00 | 43.00 |
| 8 | 24 | 3611 | 1.56% | 507 | 94.00 | 30.00 |
| 9 | 21 | 3702 | 1.47% | 479 | 94.00 | 37.00 |
| 10 | 18 | 3817 | 1.52% | 494 | 94.00 | 32.00 |
| 11 | 17 | 4099 | 2.62% | 852 | 95.00 | 57.00 |
| 12 | 16 | 4349 | 2.61% | 849 | 95.00 | 44.00 |
| 13 | 15 | 4569 | 2.23% | 723 | 93.00 | 43.00 |
| 14 | 15 | 4783 | 2.83% | 920 | 95.00 | 43.00 |
| 15 | 15 | 4905 | 2.16% | 703 | 94.00 | 55.00 |
| 16 | 15 | 5187 | 3.38% | 1097 | 94.00 | 42.00 |
| 17 | 15 | 5347 | 2.72% | 885 | 94.00 | 43.00 |
| 18 | 15 | 5604 | 3.22% | 1047 | 96.00 | 52.00 |
| 19 | 15 | 5791 | 3.06% | 993 | 95.00 | 66.00 |

*Table 17.* ANA: Per-sheaf cumulative overlap and sheaf quality (Random Init and OASR).

| ID | Cumulative Statistics | | Sheaf Quality | | | |
|---|---|---|---|---|---|---|
| | $|E_\cap|$ | $|E_\cup|$ | Edge D. (%) | $\#E$ | Acc. (%) | Comp. (%) |
| | | | **ANA – Random Init** | | | |

| ID | $|E_\cap|$ | $|E_\cup|$ | Edge D. (%) | $\#E$ | Acc. (%) | Comp. (%) |
|---|---|---|---|---|---|---|
| 0 | 635 | 635 | 1.95% | 635 | 94.00 | 41.00 |
| 1 | 224 | 1242 | 2.56% | 831 | 95.00 | 41.00 |
| 2 | 122 | 1636 | 2.18% | 709 | 95.00 | 39.00 |
| 3 | 80 | 1851 | 1.69% | 550 | 98.00 | 41.00 |
| 4 | 67 | 2122 | 2.16% | 703 | 97.00 | 41.00 |
| 5 | 56 | 2312 | 1.89% | 615 | 100.00 | 41.00 |
| 6 | 49 | 2559 | 2.35% | 765 | 95.00 | 41.00 |
| 7 | 43 | 2820 | 2.38% | 772 | 96.00 | 41.00 |
| 8 | 38 | 3033 | 2.43% | 790 | 96.00 | 59.00 |
| 9 | 34 | 3246 | 2.19% | 712 | 99.00 | 55.00 |
| 10 | 32 | 3398 | 2.06% | 669 | 97.00 | 41.00 |
| 11 | 30 | 3499 | 1.96% | 637 | 99.00 | 39.00 |
| 12 | 29 | 3650 | 2.29% | 743 | 98.00 | 41.00 |
| 13 | 28 | 3746 | 2.14% | 695 | 93.00 | 41.00 |
| 14 | 27 | 3969 | 2.72% | 883 | 97.00 | 41.00 |
| 15 | 27 | 4071 | 2.10% | 682 | 96.00 | 41.00 |
| 16 | 27 | 4205 | 2.31% | 749 | 100.00 | 41.00 |
| 17 | 27 | 4316 | 2.41% | 783 | 97.00 | 41.00 |
| 18 | 26 | 4454 | 2.48% | 805 | 92.00 | 41.00 |
| 19 | 26 | 4531 | 1.99% | 648 | 95.00 | 41.00 |

| ID | Cumulative Statistics | | Sheaf Quality | | | |
|---|---|---|---|---|---|---|
| | | | **ANA – OASR** | | | |
| | $|E_\cap|$ | $|E_\cup|$ | Edge D. (%) | $\#E$ | Acc. (%) | Comp. (%) |
| 0 | 635 | 635 | 1.95% | 635 | 94.00 | 41.00 |
| 1 | 166 | 1133 | 2.04% | 664 | 99.00 | 41.00 |
| 2 | 91 | 1589 | 2.18% | 709 | 93.00 | 42.00 |
| 3 | 58 | 1978 | 2.05% | 666 | 94.00 | 41.00 |
| 4 | 48 | 2290 | 2.01% | 653 | 96.00 | 43.00 |
| 5 | 35 | 2655 | 2.23% | 726 | 94.00 | 31.00 |
| 6 | 33 | 2896 | 1.97% | 639 | 96.00 | 41.00 |
| 7 | 29 | 3191 | 2.25% | 730 | 90.00 | 41.00 |
| 8 | 20 | 3401 | 1.80% | 584 | 93.00 | 41.00 |
| 9 | 18 | 3634 | 2.21% | 718 | 87.00 | 53.00 |
| 10 | 16 | 3809 | 1.99% | 648 | 95.00 | 41.00 |
| 11 | 15 | 3949 | 1.72% | 559 | 92.00 | 41.00 |
| 12 | 15 | 4122 | 2.05% | 667 | 98.00 | 45.00 |
| 13 | 13 | 4287 | 1.83% | 593 | 98.00 | 41.00 |
| 14 | 13 | 4374 | 1.53% | 497 | 98.00 | 41.00 |
| 15 | 13 | 4485 | 1.71% | 555 | 97.00 | 41.00 |
| 16 | 13 | 4644 | 2.12% | 690 | 96.00 | 41.00 |
| 17 | 13 | 4758 | 1.69% | 548 | 98.00 | 43.00 |
| 18 | 11 | 4830 | 1.57% | 509 | 97.00 | 41.00 |
| 19 | 10 | 4890 | 1.35% | 438 | 96.00 | 39.00 |

*Table 18.* DNA: Per-sheaf cumulative overlap and sheaf quality (Random Init and OASR).

**DNA – Random Init**

| ID | Cumulative Statistics | | Sheaf Quality | | | |
|---|---|---|---|---|---|---|
| | $|E_\cap|$ | $|E_\cup|$ | Edge D. (%) | #$E$ | Acc. (%) | Comp. (%) |
| 0 | 626 | 626 | 1.93% | 626 | 94.00 | 54.00 |
| 1 | 156 | 997 | 1.62% | 527 | 97.00 | 56.00 |
| 2 | 99 | 1261 | 1.54% | 499 | 99.00 | 58.00 |
| 3 | 82 | 1563 | 1.91% | 622 | 96.00 | 53.00 |
| 4 | 66 | 1716 | 1.38% | 450 | 98.00 | 55.00 |
| 5 | 59 | 1832 | 1.38% | 449 | 99.00 | 54.00 |
| 6 | 48 | 1995 | 1.55% | 502 | 95.00 | 54.00 |
| 7 | 46 | 2087 | 1.36% | 441 | 99.00 | 50.00 |
| 8 | 41 | 2177 | 1.39% | 453 | 97.00 | 50.00 |
| 9 | 39 | 2274 | 1.33% | 432 | 98.00 | 57.00 |
| 10 | 38 | 2356 | 1.34% | 437 | 95.00 | 54.00 |
| 11 | 36 | 2443 | 1.30% | 422 | 90.00 | 49.00 |
| 12 | 35 | 2539 | 1.49% | 485 | 98.00 | 51.00 |
| 13 | 33 | 2666 | 1.70% | 553 | 91.00 | 49.00 |
| 14 | 19 | 2756 | 1.34% | 434 | 88.00 | 53.00 |
| 15 | 17 | 2814 | 1.26% | 409 | 98.00 | 54.00 |
| 16 | 17 | 2918 | 1.82% | 592 | 97.00 | 52.00 |
| 17 | 17 | 3044 | 1.81% | 589 | 93.00 | 48.00 |
| 18 | 17 | 3096 | 1.42% | 460 | 96.00 | 51.00 |
| 19 | 17 | 3134 | 1.24% | 403 | 97.00 | 53.00 |

**DNA – OASR**

| ID | Cumulative Statistics | | Sheaf Quality | | | |
|---|---|---|---|---|---|---|
| | $|E_\cap|$ | $|E_\cup|$ | Edge D. (%) | #$E$ | Acc. (%) | Comp. (%) |
| 0 | 626 | 626 | 1.93% | 626 | 94.00 | 54.00 |
| 1 | 137 | 868 | 1.17% | 379 | 98.00 | 50.00 |
| 2 | 86 | 1250 | 1.80% | 584 | 95.00 | 54.00 |
| 3 | 62 | 1494 | 1.42% | 460 | 97.00 | 52.00 |
| 4 | 52 | 1687 | 1.31% | 425 | 94.00 | 50.00 |
| 5 | 46 | 1816 | 1.11% | 361 | 96.00 | 54.00 |
| 6 | 42 | 2030 | 1.48% | 482 | 92.00 | 55.00 |
| 7 | 39 | 2209 | 1.43% | 465 | 94.00 | 56.00 |
| 8 | 32 | 2312 | 1.10% | 356 | 88.00 | 53.00 |
| 9 | 28 | 2492 | 1.56% | 507 | 90.00 | 54.00 |
| 10 | 28 | 2610 | 1.24% | 403 | 94.00 | 56.00 |
| 11 | 26 | 2777 | 1.51% | 490 | 95.00 | 49.00 |
| 12 | 24 | 2870 | 1.28% | 415 | 98.00 | 51.00 |
| 13 | 22 | 2975 | 1.35% | 438 | 98.00 | 46.00 |
| 14 | 21 | 3078 | 1.28% | 417 | 89.00 | 53.00 |
| 15 | 20 | 3144 | 1.23% | 400 | 95.00 | 53.00 |
| 16 | 20 | 3179 | 0.78% | 255 | 90.00 | 49.00 |
| 17 | 18 | 3275 | 1.27% | 413 | 96.00 | 54.00 |
| 18 | 18 | 3371 | 1.34% | 436 | 90.00 | 54.00 |
| 19 | 17 | 3440 | 0.96% | 311 | 92.00 | 52.00 |

*Table 19.* DNA i: Per-sheaf cumulative overlap and sheaf quality (Random Init and OASR).

**DNA i – Random Init**

| ID | Cumulative Statistics | | Sheaf Quality | | | |
|---|---|---|---|---|---|---|
| | $|E_\cap|$ | $|E_\cup|$ | Edge D. (%) | #$E$ | Acc. (%) | Comp. (%) |
| 0 | 517 | 517 | 1.59% | 517 | 100.00 | 57.00 |
| 1 | 157 | 912 | 1.70% | 552 | 99.00 | 55.00 |
| 2 | 76 | 1246 | 1.71% | 554 | 94.00 | 55.00 |
| 3 | 44 | 1561 | 1.92% | 623 | 100.00 | 48.00 |
| 4 | 41 | 1791 | 1.90% | 616 | 100.00 | 49.00 |
| 5 | 36 | 2012 | 1.85% | 602 | 100.00 | 50.00 |
| 6 | 33 | 2139 | 1.47% | 478 | 96.00 | 55.00 |
| 7 | 30 | 2274 | 1.75% | 567 | 100.00 | 58.00 |
| 8 | 27 | 2367 | 1.29% | 418 | 97.00 | 61.00 |
| 9 | 26 | 2564 | 2.06% | 669 | 100.00 | 59.00 |
| 10 | 26 | 2676 | 1.83% | 595 | 100.00 | 55.00 |
| 11 | 26 | 2785 | 1.57% | 510 | 99.00 | 58.00 |
| 12 | 24 | 2926 | 1.84% | 598 | 99.00 | 55.00 |
| 13 | 21 | 3113 | 1.99% | 646 | 95.00 | 59.00 |
| 14 | 20 | 3207 | 1.70% | 551 | 95.00 | 53.00 |
| 15 | 19 | 3349 | 2.07% | 674 | 100.00 | 59.00 |
| 16 | 19 | 3418 | 1.57% | 509 | 99.00 | 56.00 |
| 17 | 19 | 3500 | 1.77% | 575 | 100.00 | 60.00 |
| 18 | 19 | 3567 | 1.71% | 556 | 100.00 | 54.00 |
| 19 | 19 | 3649 | 1.81% | 588 | 100.00 | 53.00 |

**DNA i – OASR**

| ID | Cumulative Statistics | | Sheaf Quality | | | |
|---|---|---|---|---|---|---|
| | $|E_\cap|$ | $|E_\cup|$ | Edge D. (%) | #$E$ | Acc. (%) | Comp. (%) |
| 0 | 517 | 517 | 1.59% | 517 | 100.00 | 57.00 |
| 1 | 113 | 924 | 1.60% | 520 | 100.00 | 54.00 |
| 2 | 62 | 1242 | 1.54% | 499 | 99.00 | 55.00 |
| 3 | 43 | 1522 | 1.55% | 505 | 99.00 | 56.00 |
| 4 | 31 | 1762 | 1.37% | 446 | 93.00 | 53.00 |
| 5 | 29 | 1869 | 0.98% | 320 | 98.00 | 56.00 |
| 6 | 20 | 2091 | 1.43% | 466 | 95.00 | 56.00 |
| 7 | 19 | 2302 | 1.59% | 516 | 94.00 | 55.00 |
| 8 | 19 | 2449 | 1.30% | 423 | 100.00 | 56.00 |
| 9 | 15 | 2552 | 1.11% | 361 | 100.00 | 57.00 |
| 10 | 13 | 2677 | 1.18% | 385 | 91.00 | 56.00 |
| 11 | 13 | 2754 | 0.86% | 280 | 99.00 | 55.00 |
| 12 | 11 | 2861 | 1.18% | 384 | 99.00 | 46.00 |
| 13 | 10 | 2997 | 1.28% | 415 | 99.00 | 52.00 |
| 14 | 10 | 3029 | 0.73% | 238 | 95.00 | 53.00 |
| 15 | 9 | 3069 | 0.85% | 277 | 92.00 | 57.00 |
| 16 | 9 | 3097 | 0.73% | 237 | 98.00 | 56.00 |
| 17 | 9 | 3126 | 0.86% | 280 | 98.00 | 57.00 |
| 18 | 9 | 3209 | 1.03% | 335 | 94.00 | 50.00 |
| 19 | 9 | 3252 | 0.87% | 284 | 92.00 | 56.00 |

*Table 20.* DNA a: Per-sheaf cumulative overlap and sheaf quality (Random Init and OASR).

**DNA a – Random Init**

| ID | Cumulative Statistics | | Sheaf Quality | | | |
|---|---|---|---|---|---|---|
| | $\|E_\cap\|$ | $\|E_\cup\|$ | Edge D. (%) | #E | Acc. (%) | Comp. (%) |
| 0 | 619 | 619 | 1.91% | 619 | 99.00 | 50.00 |
| 1 | 207 | 1081 | 2.06% | 669 | 96.00 | 54.00 |
| 2 | 129 | 1464 | 2.25% | 731 | 96.00 | 53.00 |
| 3 | 93 | 1796 | 2.18% | 707 | 96.00 | 52.00 |
| 4 | 81 | 1978 | 1.87% | 608 | 98.00 | 52.00 |
| 5 | 66 | 2155 | 1.96% | 636 | 97.00 | 50.00 |
| 6 | 57 | 2328 | 1.86% | 603 | 96.00 | 54.00 |
| 7 | 41 | 2560 | 2.14% | 694 | 96.00 | 54.00 |
| 8 | 34 | 2668 | 1.51% | 491 | 95.00 | 55.00 |
| 9 | 31 | 2824 | 2.33% | 756 | 97.00 | 55.00 |
| 10 | 31 | 3016 | 2.43% | 791 | 96.00 | 54.00 |
| 11 | 29 | 3194 | 2.44% | 792 | 98.00 | 54.00 |
| 12 | 29 | 3343 | 2.27% | 739 | 92.00 | 54.00 |
| 13 | 29 | 3426 | 2.16% | 701 | 97.00 | 54.00 |
| 14 | 25 | 3562 | 2.27% | 737 | 94.00 | 54.00 |
| 15 | 23 | 3676 | 2.51% | 816 | 93.00 | 54.00 |
| 16 | 23 | 3769 | 2.32% | 753 | 97.00 | 53.00 |
| 17 | 23 | 3850 | 2.43% | 791 | 99.00 | 52.00 |
| 18 | 23 | 3915 | 2.15% | 699 | 98.00 | 55.00 |
| 19 | 23 | 3982 | 2.13% | 692 | 97.00 | 53.00 |

**DNA a – OASR**

| ID | Cumulative Statistics | | Sheaf Quality | | | |
|---|---|---|---|---|---|---|
| | $\|E_\cap\|$ | $\|E_\cup\|$ | Edge D. (%) | #E | Acc. (%) | Comp. (%) |
| 0 | 663 | 663 | 2.04% | 663 | 97.00 | 54.00 |
| 1 | 190 | 1139 | 2.05% | 666 | 99.00 | 50.00 |
| 2 | 94 | 1690 | 2.46% | 800 | 98.00 | 58.00 |
| 3 | 69 | 2099 | 2.44% | 794 | 91.00 | 54.00 |
| 4 | 57 | 2338 | 2.06% | 669 | 100.00 | 54.00 |
| 5 | 47 | 2595 | 2.20% | 714 | 97.00 | 54.00 |
| 6 | 42 | 2795 | 1.83% | 593 | 96.00 | 52.00 |
| 7 | 36 | 2935 | 1.55% | 505 | 87.00 | 54.00 |
| 8 | 33 | 3231 | 2.53% | 822 | 97.00 | 54.00 |
| 9 | 32 | 3402 | 2.02% | 656 | 96.00 | 56.00 |
| 10 | 30 | 3506 | 1.83% | 593 | 96.00 | 54.00 |
| 11 | 29 | 3656 | 2.10% | 681 | 96.00 | 55.00 |
| 12 | 28 | 3772 | 2.10% | 682 | 97.00 | 56.00 |
| 13 | 28 | 3897 | 2.12% | 690 | 93.00 | 52.00 |
| 14 | 26 | 3982 | 1.67% | 542 | 93.00 | 58.00 |
| 15 | 25 | 4054 | 1.78% | 579 | 95.00 | 55.00 |
| 16 | 22 | 4118 | 1.59% | 516 | 95.00 | 54.00 |
| 17 | 22 | 4165 | 1.67% | 543 | 95.00 | 55.00 |
| 18 | 22 | 4206 | 1.85% | 601 | 98.00 | 55.00 |
| 19 | 22 | 4469 | 4.26% | 1384 | 93.00 | 55.00 |

*Table 21.* DNA ia: Per-sheaf cumulative overlap and sheaf quality (Random Init and OASR).

**DNA ia – Random Init**

| ID | Cumulative Statistics | | Sheaf Quality | | | |
|---|---|---|---|---|---|---|
| | $|E_\cap|$ | $|E_\cup|$ | Edge D. (%) | #$E$ | Acc. (%) | Comp. (%) |
| 0 | 677 | 677 | 2.08% | 677 | 84.00 | 51.00 |
| 1 | 247 | 1205 | 2.39% | 775 | 86.00 | 49.00 |
| 2 | 147 | 1537 | 2.21% | 718 | 93.00 | 50.00 |
| 3 | 111 | 1815 | 2.44% | 794 | 97.00 | 50.00 |
| 4 | 94 | 2153 | 2.69% | 873 | 93.00 | 52.00 |
| 5 | 82 | 2529 | 3.02% | 980 | 98.00 | 50.00 |
| 6 | 75 | 2815 | 2.70% | 878 | 98.00 | 50.00 |
| 7 | 69 | 3022 | 2.52% | 819 | 95.00 | 50.00 |
| 8 | 67 | 3252 | 2.64% | 858 | 97.00 | 50.00 |
| 9 | 60 | 3546 | 3.01% | 977 | 99.00 | 48.00 |
| 10 | 57 | 3701 | 2.39% | 775 | 92.00 | 49.00 |
| 11 | 54 | 3796 | 2.18% | 709 | 99.00 | 56.00 |
| 12 | 48 | 4002 | 2.88% | 936 | 95.00 | 50.00 |
| 13 | 44 | 4168 | 2.58% | 838 | 88.00 | 47.00 |
| 14 | 42 | 4313 | 2.82% | 917 | 99.00 | 55.00 |
| 15 | 40 | 4389 | 2.30% | 747 | 98.00 | 55.00 |
| 16 | 38 | 4488 | 2.71% | 879 | 97.00 | 50.00 |
| 17 | 38 | 4588 | 2.70% | 877 | 99.00 | 50.00 |
| 18 | 37 | 4728 | 2.84% | 924 | 97.00 | 51.00 |
| 19 | 33 | 4813 | 2.74% | 891 | 96.00 | 53.00 |

**DNA ia – OASR**

| ID | Cumulative Statistics | | Sheaf Quality | | | |
|---|---|---|---|---|---|---|
| | $|E_\cap|$ | $|E_\cup|$ | Edge D. (%) | #$E$ | Acc. (%) | Comp. (%) |
| 0 | 785 | 785 | 2.42% | 785 | 99.00 | 41.00 |
| 1 | 244 | 1445 | 2.78% | 904 | 96.00 | 51.00 |
| 2 | 143 | 1860 | 2.43% | 790 | 96.00 | 55.00 |
| 3 | 80 | 2091 | 1.67% | 541 | 86.00 | 51.00 |
| 4 | 62 | 2487 | 2.60% | 844 | 87.00 | 56.00 |
| 5 | 53 | 2869 | 3.03% | 985 | 96.00 | 51.00 |
| 6 | 48 | 3053 | 2.21% | 718 | 95.00 | 55.00 |
| 7 | 46 | 3229 | 2.39% | 775 | 100.00 | 53.00 |
| 8 | 44 | 3411 | 2.25% | 731 | 88.00 | 55.00 |
| 9 | 43 | 3612 | 2.56% | 833 | 97.00 | 53.00 |
| 10 | 41 | 3807 | 2.58% | 838 | 100.00 | 55.00 |
| 11 | 37 | 4007 | 2.48% | 807 | 87.00 | 45.00 |
| 12 | 31 | 4296 | 2.96% | 963 | 82.00 | 50.00 |
| 13 | 31 | 4409 | 2.18% | 707 | 95.00 | 50.00 |
| 14 | 30 | 4527 | 2.43% | 791 | 98.00 | 50.00 |
| 15 | 29 | 4602 | 2.08% | 676 | 99.00 | 55.00 |
| 16 | 27 | 4700 | 1.96% | 636 | 96.00 | 51.00 |
| 17 | 27 | 4859 | 2.59% | 842 | 97.00 | 61.00 |
| 18 | 25 | 4954 | 2.19% | 711 | 87.00 | 55.00 |
| 19 | 25 | 5063 | 2.60% | 844 | 97.00 | 55.00 |

*Table 22.* Docstring: Per-sheaf cumulative overlap and sheaf quality (Random Init and OASR).

| | | | | | | |
|---|---|---|---|---|---|---|
| | **Docstring – Random Init** | | | | | |
| **ID** | **Cumulative Statistics** | | **Sheaf Quality** | | | |
| | $|E_\cap|$ | $|E_\cup|$ | Edge D. (%) | #E | Acc. (%) | Comp. (%) |
| 0 | 1081 | 1081 | 3.33% | 1081 | 100.00 | 51.00 |
| 1 | 356 | 1930 | 3.71% | 1205 | 98.40 | 54.00 |
| 2 | 220 | 2466 | 3.41% | 1107 | 100.00 | 50.90 |
| 3 | 149 | 3087 | 3.92% | 1275 | 97.10 | 51.30 |
| 4 | 122 | 3543 | 3.59% | 1166 | 99.70 | 51.90 |
| 5 | 105 | 3868 | 3.27% | 1064 | 100.00 | 49.40 |
| 6 | 92 | 4105 | 2.82% | 916 | 99.80 | 51.70 |
| 7 | 81 | 4413 | 3.40% | 1105 | 99.80 | 52.60 |
| 8 | 67 | 4767 | 3.83% | 1244 | 99.40 | 50.90 |
| 9 | 64 | 4980 | 3.38% | 1099 | 99.60 | 52.00 |
| 10 | 59 | 5162 | 2.91% | 944 | 100.00 | 52.60 |
| 11 | 53 | 5262 | 2.39% | 775 | 100.00 | 49.40 |
| 12 | 51 | 5488 | 3.59% | 1166 | 99.80 | 50.70 |
| 13 | 49 | 5674 | 3.51% | 1139 | 100.00 | 51.80 |
| 14 | 47 | 5854 | 3.15% | 1025 | 100.00 | 51.10 |
| 15 | 46 | 5986 | 3.04% | 988 | 100.00 | 50.30 |
| 16 | 40 | 6209 | 3.55% | 1154 | 98.60 | 53.60 |
| 17 | 40 | 6397 | 3.62% | 1175 | 100.00 | 52.30 |
| 18 | 40 | 6568 | 3.66% | 1188 | 100.00 | 53.20 |
| 19 | 39 | 6719 | 3.70% | 1202 | 100.00 | 52.40 |

| | | | | | | |
|---|---|---|---|---|---|---|
| | **Docstring – OASR** | | | | | |
| **ID** | **Cumulative Statistics** | | **Sheaf Quality** | | | |
| | $|E_\cap|$ | $|E_\cup|$ | Edge D. (%) | #E | Acc. (%) | Comp. (%) |
| 0 | 1081 | 1081 | 3.33% | 1081 | 100.00 | 51.00 |
| 1 | 236 | 1842 | 3.07% | 997 | 100.00 | 51.80 |
| 2 | 140 | 2601 | 3.64% | 1182 | 99.60 | 52.80 |
| 3 | 92 | 3125 | 3.06% | 993 | 96.30 | 51.40 |
| 4 | 58 | 3729 | 3.56% | 1156 | 98.90 | 51.60 |
| 5 | 48 | 4106 | 2.77% | 899 | 99.90 | 52.50 |
| 6 | 44 | 4418 | 2.66% | 863 | 100.00 | 50.80 |
| 7 | 39 | 4788 | 3.50% | 1137 | 95.60 | 53.00 |
| 8 | 37 | 5077 | 3.28% | 1065 | 99.70 | 53.30 |
| 9 | 33 | 5338 | 3.00% | 974 | 100.00 | 51.80 |
| 10 | 32 | 5627 | 3.19% | 1035 | 100.00 | 52.30 |
| 11 | 22 | 5778 | 1.87% | 607 | 100.00 | 52.00 |
| 12 | 22 | 6049 | 3.59% | 1167 | 98.20 | 50.20 |
| 13 | 21 | 6373 | 3.67% | 1193 | 100.00 | 51.90 |
| 14 | 21 | 6542 | 2.90% | 941 | 99.30 | 51.10 |
| 15 | 19 | 6742 | 2.96% | 962 | 99.10 | 50.70 |
| 16 | 19 | 6873 | 2.65% | 861 | 96.30 | 54.30 |
| 17 | 19 | 7017 | 2.97% | 966 | 100.00 | 52.00 |
| 18 | 19 | 7132 | 2.65% | 862 | 100.00 | 52.20 |
| 19 | 18 | 7314 | 3.25% | 1055 | 97.80 | 48.40 |

Per-Layer Incoming Edge Counts

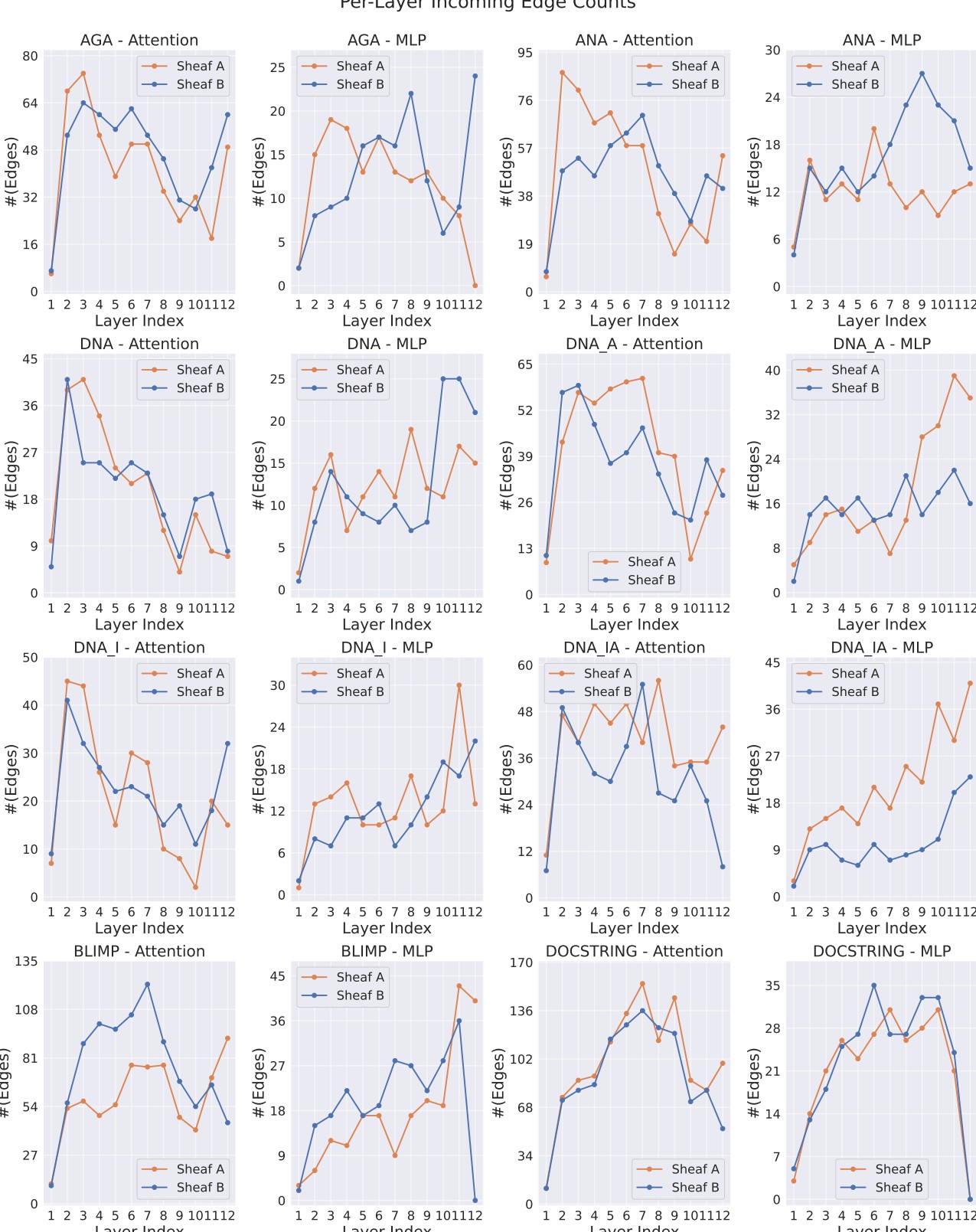

*Figure 8.* The distributions of incoming edge count across layers and different types of components in pairs of sheaves discovered.

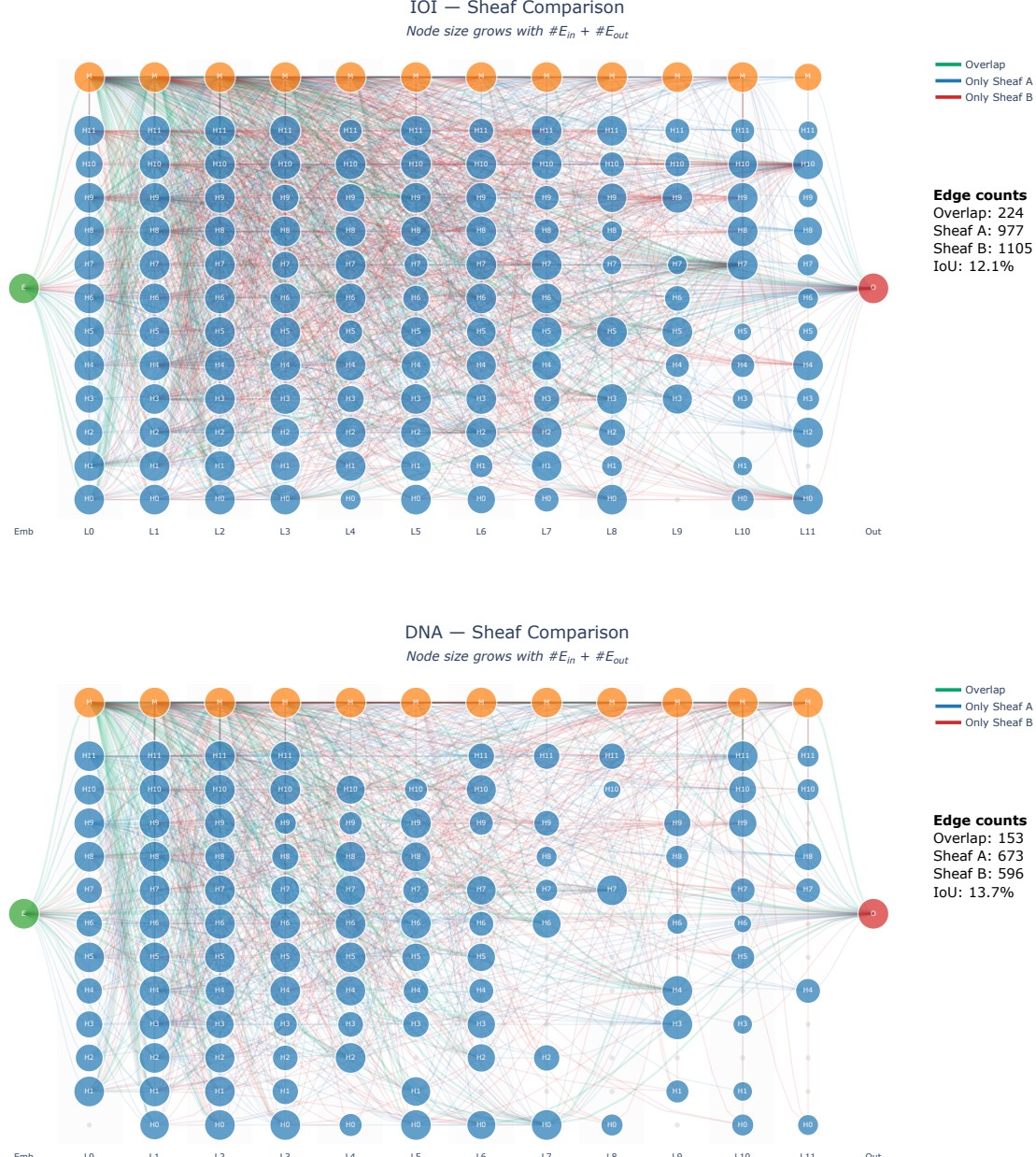

*Figure 9.* We visualize the computational graphs of the sheaves discovered for the IOI and DNA tasks. When the computation graph is rendered at a very fine granularity (e.g., with explicit Q/K/V/O decomposition), the resulting edge density leads to severe visual clutter that obscures the global structure. Note that this also results in fewer edge counts by re-merging the Q/K/V/O nodes. In contrast, representing the model at the level of attention heads and MLP blocks yields a more compact and interpretable graph that preserves the salient connectivity patterns. These visualizations further illustrate that substantial structural differences can nonetheless give rise to functionally equivalent computational substructures.

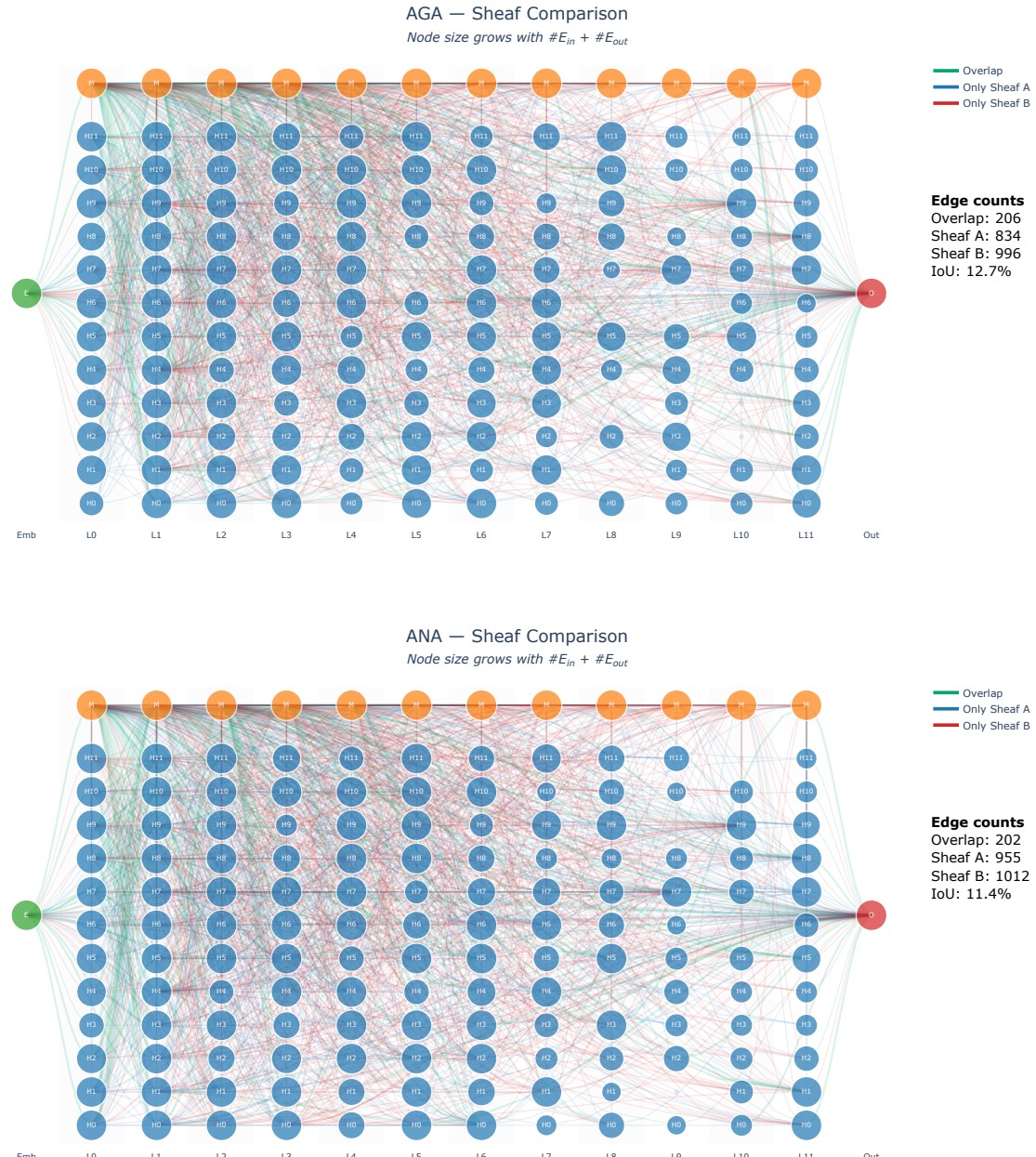

*Figure 10.* We visualize the computational graphs of the sheaves discovered for the AGA and ANA tasks. The sparseness of the overlapping edges supports the similar argument as given above for Figure 9.

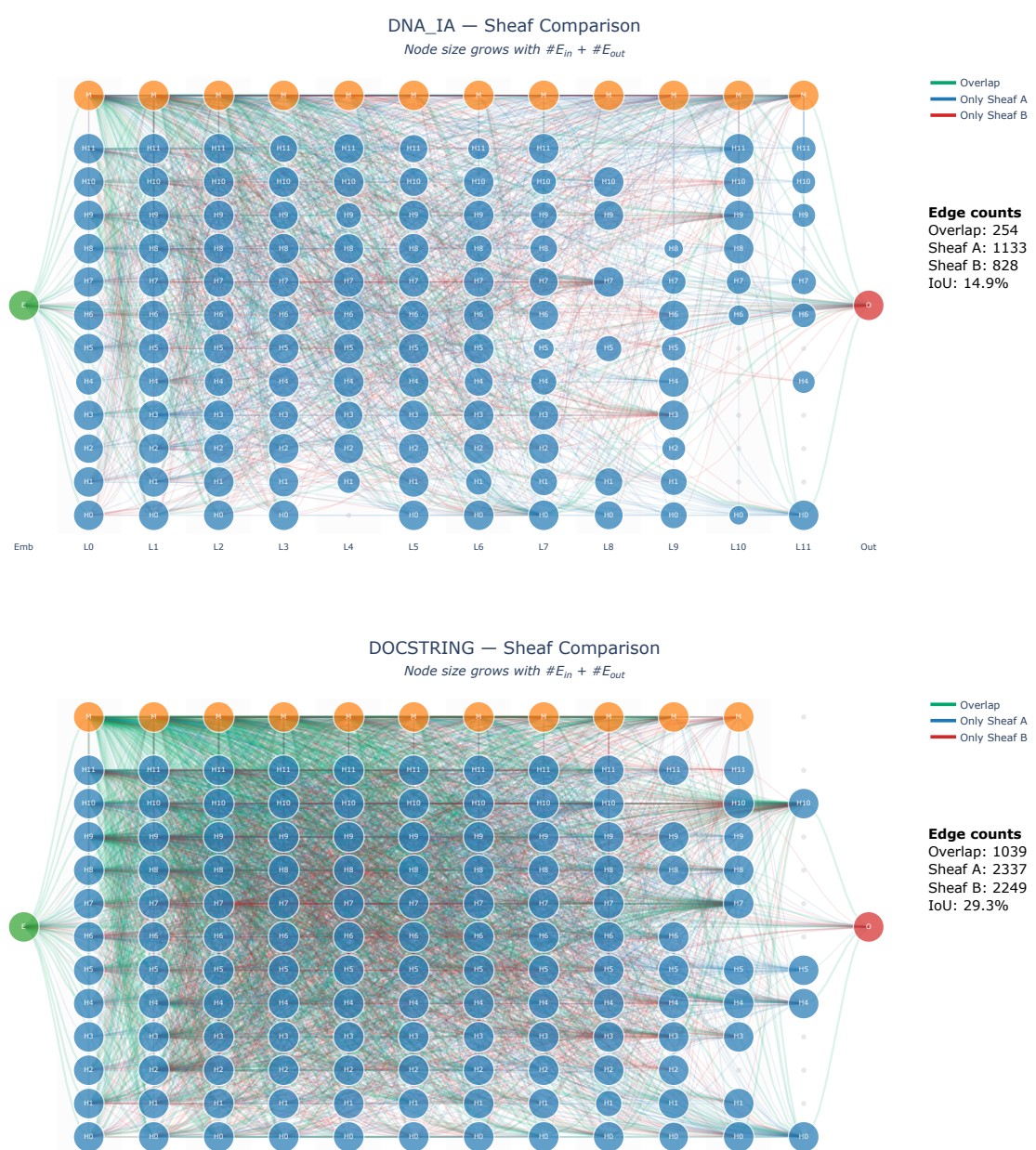

*Figure 11.* We visualize the computational graphs of the sheaves discovered for the DNA ia and Docstring tasks. The same conclusion still holds for these two tasks given the visual similarity of the connectivity patterns across all sheaf visualizations.

# M. Extended Related Work

## M.1. Explainability of Language Models

With the remarkable performance of LLMs in natural language processing tasks, understanding the mechanisms behind their decisions (i.e., "interpretability") has become a key area of research (Lee et al., 2025; Du et al., 2019; Zhao et al., 2024; 2025; Shu et al., 2025). Existing research on LLM interpretability can be broadly categorised into four main directions: feature attribution methods, representation analysis and probing methods, mechanistic interpretability, and natural language-based interpretation.

**Feature Attribution Methods**   These methods aim to quantify how much each input token contributes to a model's final prediction, thereby producing token-level evidence distributions (Gao et al., 2024). Early work on Transformer interpretability often relied on attention-based analyses, treating attention weights as direct proxies for feature importance (Serrano & Smith, 2019). However, subsequent studies argue that attention is not inherently faithful: substantially different attention distributions can yield nearly the same outputs, and attention weights may correlate weakly with more intervention-relevant importance measures, calling into question their use as explanations in a strict causal sense (Jain & Wallace, 2019; Bai et al., 2021). Motivated by these limitations, the literature increasingly adopts gradient-based explanations (e.g., saliency maps) that measure local sensitivity of the output to the input representation (Simonyan et al., 2013; Ancona et al., 2018). To improve stability and satisfy principled axioms such as sensitivity and implementation invariance, Integrated Gradients computes path-integrated attributions from a baseline to the input and has become a standard tool for token-level interpretation in NLP (Sundararajan et al., 2017). Complementary smoothing techniques such as SmoothGrad reduce visual and numerical noise in gradient-based maps (Smilkov et al., 2017). Beyond gradients, perturbation/erasure-based approaches evaluate token importance by removing or masking parts of the input (or intermediate representations) and measuring the induced change in prediction, offering a more direct intervention test of attribution faithfulness (Li et al., 2016). Finally, game-theoretic attribution frameworks grounded in the Shapley value formalism motivate additive feature-credit decompositions; modern instantiations such as LIME and SHAP provide practical approximations for local explanations and unify several additive attribution families (Lundberg & Lee, 2017; Ribeiro et al., 2016; Shapley, 1953).

**Representation Analysis and Probing Methods**   These methods aim to understand what linguistic knowledge is encoded in the internal hidden states of neural language models. A dominant paradigm in this line of work is the use of probing classifiers, which train lightweight supervised models—typically linear classifiers or shallow neural networks—on frozen representations to predict specific linguistic properties (Conneau et al., 2018; Adi et al., 2017; Belinkov, 2022). These properties range from low-level features such as part-of-speech tags and morphological attributes to higher-level structures including syntactic dependency depth, constituency structure, semantic roles, and entity relations. Successful probe performance is commonly interpreted as evidence that the probed linguistic information is linearly or simply recoverable from the representations at a given layer (Hewitt & Manning, 2019; Conneau et al., 2018; Hewitt & Liang, 2019; Tenney et al., 2019). Beyond standard probes, specialised techniques such as structural probes explicitly test whether syntactic structures are embedded in representation geometry, for example by learning a low-rank transformation under which Euclidean distances correspond to dependency tree distances (Hewitt & Manning, 2019). However, the probing paradigm has also raised methodological concerns: high probe accuracy may reflect the expressive power of the probe rather than the information genuinely encoded in the representation. To address this issue, prior work introduces control tasks, selectivity measures, and probe-capacity constraints to disentangle representational content from probe learning effects (Tenney et al., 2019). Complementary perspectives frame probing as an instance of information estimation, arguing that probes should be interpreted as bounds on mutual information between representations and linguistic variables rather than definitive evidence of functional usage (Pimentel et al., 2020). At a broader level, extensive layer-wise analyses reveal a consistent hierarchical organisation in Transformer-based language models. Empirical studies show that lower layers tend to encode surface-level and morphological features, intermediate layers capture syntactic relations and structural regularities, and higher layers increasingly represent semantic, contextual, and discourse-level information (Tenney et al., 2019). This progression has often been summarised as pretrained language models "rediscovering" a classical NLP processing pipeline. Comprehensive surveys further systematise these findings and discuss the limitations of probing-based analysis, particularly regarding faithfulness, causality, and cross-task generalisation (Rogers et al., 2020; Belinkov, 2022; Niu et al., 2022; Jin et al., 2025b; 2026; Shi et al., 2025).

**Mechanistic Interpretability**   This line of work seeks to understand neural networks by reverse engineering their internal computations. Rather than treating models as black boxes or focusing solely on input–output correlations, it aims to

identify the functional roles of concrete model components, such as individual neurons, attention heads, and MLP channels, to explain model behaviour in terms of explicit, human-interpretable computational mechanisms (Sharkey et al., 2025; Zhao et al., 2025; Dreyer et al., 2025). A central methodology in this paradigm is circuit analysis, which decomposes complex model behaviours into structured interaction patterns among a small set of components that collectively implement an algorithmic function (Olah et al., 2020). Subsequent mechanistic studies on Transformers demonstrated that specific attention heads, most notably induction heads, play a critical role in tasks involving in-context pattern matching and sequence continuation by attending to repeated token pairs and copying information across positions (Olsson et al., 2022). These findings provide concrete evidence that Transformer models internally instantiate algorithmic primitives, such as prefix matching and copy mechanisms, through localised and reusable circuits (Elhage et al., 2021). By grounding explanations in identifiable modules and causal interventions, mechanistic interpretability offers a path toward faithful, causal accounts of model behaviour beyond descriptive attribution or representational probing (Wang et al., 2022a; Jin et al., 2025a).

In particular, a series of studies leverage causal tracing and activation patching to localise where and how specific factual, syntactic, or reasoning-related information is represented and propagated through the model, enabling fine-grained intervention at the level of attention heads, MLP weights and neurons, and residual streams (Meng et al., 2022; Geva et al., 2021; Wang et al., 2022a; Niu et al., 2024; 2025b; Yu et al., 2024a; Tiblias et al., 2026). Parallel to this line, feature-based decompositions such as sparse autoencoders (SAEs) have been introduced to disentangle superposed representations into more interpretable latent features, facilitating neuron- and subspace-level analysis at scale and revealing structured computational motifs underlying abstract concepts and behaviours (Bricken et al., 2023; Huben et al., 2023; Han et al., 2026; He et al., 2025). A complementary line of work isolates individual model behaviours along single linear directions in activation space, including refusal, verbal uncertainty, and hallucination-reduction steering (Yu et al., 2024b; Ji et al., 2025; Yang et al., 2025). These approaches have enabled the discovery of higher-level circuits responsible for phenomena such as indirect object identification, multi-token copying, and semantic role tracking, moving mechanistic interpretability beyond single-head analyses toward multi-component, distributed circuits (Elhage et al., 2022). At the same time, recent studies emphasise the non-uniqueness and redundancy of mechanistic explanations: multiple distinct circuits or feature combinations may implement functionally equivalent behaviours, and a single direction or mechanism can itself mediate distinct functional regimes such as memorisation versus algorithmic processing, challenging the notion of a single canonical decomposition (Elhage et al., 2022; Dreyer et al., 2025; Méloux et al., 2025; Hong et al., 2025; Niu et al., 2025a). This observation has motivated a shift from identifying minimal circuits toward characterising families of functionally valid mechanisms and understanding their stability across model scales, training checkpoints, and distribution shifts. Collectively, these post-2022 advances position mechanistic interpretability as a unifying, causal framework that bridges attribution, representation analysis, and intervention-based model editing, while also highlighting the need for principled abstractions that remain faithful yet scalable for modern large language models.

**Natural Language-Based Interpretation**   This approach takes a different route from attribution, probing, or circuit analysis: instead of explaining a model by inspecting internal signals, it asks the model (or an auxiliary model) to produce human-readable explanations e.g., free-text rationales, step-by-step chain-of-thought (CoT), or self-critiques alongside predictions (Wei et al., 2022). Early work operationalised this idea by learning to generate rationales as textual justifications, either by extracting input fragments as "sufficient" explanations or by producing free-form explanations supervised from human-written rationales (Lei et al., 2016). With the rise of instruction-tuned LLMs, prompting-based approaches such as CoT have become a widely used interface for eliciting intermediate reasoning and improving performance on multi-step tasks (Jacovi & Goldberg, 2020). However, a growing body of work questions whether such generated rationales faithfully reflect the model's underlying computation, showing that CoT traces can be post-hoc, inconsistent with intervention-based evidence, or sensitive to prompt-level perturbations. Recent work therefore emphasises that natural language explanations should not be treated as self-authenticating evidence of "how"the model reasoned," and instead should be validated via consistency checks, counterfactual tests, or complementary causal analyses; accordingly, some approaches explicitly incorporate counterfactual or faithfulness-oriented objectives to improve the reliability of rationale-based reasoning pipelines (Wang et al., 2022b).

## M.2. Circuit Discovery History

There have been multiple variants of the definition of a circuit. Olah et al. (2020) defined it as a subgraph of a neural network consisting of a set of features and the weighted connections between them. The term has since been used more broadly to refer to a collection of network components and a subset of their connections (Wang et al., 2022a; Conmy et al., 2023). However, what constitutes a network component remains debated in the community: common choices include attention

heads and MLPs, while edges typically refer to the weighted contributions of these components to the residual stream. A major breakthrough in circuit theory was moving away from seeing the model as a sequence of layers stacked together and instead viewing it as a residual stream. In this view, the residual stream is a shared communication channel. Circuits are then defined by which components read from the stream and which write to it. Elhage et al. (2021) formalised the idea that attention heads can be understood as independent circuits. By analysing the product of weight matrices ($W_O W_V$ circuit for content and $W_Q^\top W_K$ circuit for allocation), one can identify "copying" and "composition" model behavioural mechanism purely from weights.

Early work relied on manual search through interventions (Meng et al., 2022) and logit lens–style readouts of intermediate representations (Geva et al., 2022) to identify circuits for simple tasks in the pursuit of mechanistic explanations of model behaviour. More recently, ACDC (Conmy et al., 2023) automated the process of circuit discovery by evaluating, in reverse topological order, the local importance of component connections (edges) using KL divergence, and pruning those below a predefined threshold. Syed et al. (2024) improved upon ACDC by introducing efficient first-order estimation of edge attribution. There has been follow up work on attribution patching with improved approximation and incorporation of the positional information (Zhang et al., 2025; Hanna et al., 2024). Gradient-based methods further enable efficient global assessment of component importance under sparsity regularisation (Bhaskar et al., 2024; Yu et al., 2025), empirically uncovering more compact and functional circuits without relying on local approximations.

## N. The Use of Large Language Models (LLMs)

Large Language Models (LLMs), such as ChatGPT, were used solely as general-purpose tools for **(i)** language editing and stylistic polishing of manuscript drafts and **(ii)** limited coding assistance for minor boilerplate components (e.g., plotting scripts and small utility functions). All LLM-generated outputs were carefully reviewed, revised, and validated by the authors prior to use. All research ideas, algorithmic designs, experimental setups, datasets, analyses, and conclusions were independently conceived, executed, and verified by the authors. LLMs were **not** used to generate experimental results, annotations, ground-truth labels, or to make methodological or scientific decisions. The authors take full responsibility for the content of this paper.

