# OpenReview forum: "All Circuits Lead to Rome: Rethinking Functional Anisotropy in Circuit and Sheaf Discovery for LLMs"
_ICML.cc/2026/Conference — ICML 2026 regular_

### Official Review · Reviewer_NXvU · 2026-02-13

**Soundness:** 2
**Presentation:** 2
**Significance:** 2
**Originality:** 2
**Overall Recommendation:** 3
**Confidence:** 3

**Summary:**

This paper argues against what they termed the Functional Anisotropy Hypothesis, which they defines as the implicit assumption that LLM capabilities are supported by a single, well-defined internal mechanism. They made the observation that non-overlapping sheaves can each faithfully perform the same task independently, and they introduced a new method (overlap-aware
sheaf repulsion process) to show that this is true.

**Compliance With Llm Reviewing Policy:**

Affirmed.

**Final Justification:**

I thank the authors for the rebuttal.

After the rebuttal, I remain unconvinced of the biggest soundness weakness I pointed out in the paper regarding how "Functional Anisotropy Hypothesis" was formulated in a rigorous fashion from prior works. Thus, I remain skeptical of the significance as well as the soundness of the paper.

The authors' follow-up rebuttal reinforced my assessment, the author mentions "Méloux et al.‘s (2025)" but giving no reference to it in the reply. I had to look it up in the manuscript itself, and the cited reference does not even shown the name "Méloux", but "Mloux" instead.

Moreover, in my review, I have never said anything related to this:

> Our work is novel. Méloux et al. (2025) provide useful theoretical grounding for reasoning about the uniqueness of explanations by formalizing mappings between low-level representations and high-level functions. We thank the reviewer for pointing this out, which helped us clarify our formulation and its conceptual grounding. However, the object of study differs: Méloux et al. ...

Yet the authors thanked me for pointing this out, and tried to correct me on a point that I did not mention to begin with. This strikes me as unusual and speak to the overall quality of the paper. It gives me the same impression that I originally get from reading the paper: the author identify a non-relevant hypothesis and go about disproving it themselves.

Given that, I will keep my rating. Of course, as I don't have backgrounds in this field, I mentioned that this might reflect limits in my own understanding on this particular line of work rather than a definitive flaw in the paper. Thus, I'll defer to the AC and other reviewers with stronger subject-specific expertise to evaluate whether my questioning of the "Functional Anisotropy Hypothesis" was fair.

**Key Questions For Authors:**

1. I think the paper could be made stronger by more rigorous framing of the proposed hypotheses. I find it very hand-wavy for now, but I will be open to raising the scores once the hypotheses are better framed theoretically. Are there plans to make this stronger?

2. The experimental section seems comprehensive, are there any potential limitations to the set of experiments performed?

**Limitations:**

I don't find that the limitations were adequately discussed. There is little discussion about the limitations of the experiments or theoretical framing. I think adding a clear limitation section detailing the paper's own assumptions, and limits to their experimental result would make the paper stronger.

**Strengths And Weaknesses:**

## Strengths:

* **Soundness:**
    * I find the paper technically sound, where the experiment section is quite comprehensive covering a wide range of benchmarks under different prior methods.
    * The experiments align with their central claim for the proposed Distributive Dense Circuit Hypothesis.
* **Presentation:**
    * I find the paper easy to follow, with the main claims and contributions clearly stated.
    * The work properly positions itself in the context of prior/concurrent literature and clearly discusses how it differs.
    * Very detailed explanations of previous works.
* **Significance & Originality:**
    * The paper points out an interesting effect, which I see as insights that could lead to better understanding on LLMs internal mechanism.
    * These effects could potentially be exploited for a more structured exploration of how LLMs work.

## Weaknesses:

* **Soundness:**
    * The author argues against an "implicit" assumptions made by previous work, which they coined "Functional Anisotropy Hypothesis." However, the authors do not clearly show whether these assumptions were that "LLM capabilities are supported by **only a single**, well-defined internal mechanism" or "LLM capabilities are supported by **at least a single**, well-defined internal mechanism." They try to justify it in section 2.2, but I find it very hand-wavy, as they only quote very vague language from previous papers. I find that this is a strong assumption made by the author that could invalidate the central thesis of their paper. Reading previous works cited I found that it is not clear that prior works made the assumption as claimed in this paper.
    * I find that the theoretical statement of the paper is quite weak, and, in relation to the previous point, such statement could probably be made from careful consideration of prior works.
    * The paper does not provide strong theoretical backups for their hypothesis; most of the paper is dedicated to empirical observations and then a final claim about their "theoretical" findings.
    * There is no open discussion relating to either experimental or theoretical limits of their works, despite claiming what is seemingly a strong theoretical result.
* **Presentation:**
    * The abstract is a bit too long and presents too much detail of their work. Large parts of the abstract should belong in the main body of the paper.

---

> ### Author Rebuttal · Authors · 2026-03-31
>
> We thank the reviewer for recognizing the soundness, significance, originality, and clarity of our work. We greatly appreciate this feedback and are encouraged by the recognition.
>
> The main question raised is about our identification of the Functional Anisotropy Hypothesis, namely the idea that prior work implicitly assumes a one-to-one relationship between mechanism and task. We are happy to take this opportunity to clarify our position.
>
> ### **Q1: Functional Anisotropy Hypothesis**
>
> While we agree that prior work does not explicitly state the Functional Anisotropy Hypothesis, our goal is not to attribute a formal claim to the literature, but to *surface a previously under-articulated assumption that is implicitly reflected in how existing methods are designed and evaluated*. Here, we clarify this point from three perspectives: (1) what prior work assumes in practice, (2) how this is reflected in existing methods, and (3) how our contribution relates to this observation.
>
> To clarify, we are not claiming that prior work explicitly assumes that a task is supported by only a single mechanism. Rather, our point is that many existing methods are designed to identify and evaluate a single dominant mechanism for a given task. In this sense, the one-to-one mapping arises operationally from how these methods are formulated and evaluated, even if it is not explicitly stated as a theoretical assumption.
>
> Our justification given in Section 2.2 is derived from a thorough examination of how existing methods are formulated and evaluated, rather than relying solely on individual quotes. While we include representative excerpts from prior work, these serve only as illustrative examples; our argument is grounded in the underlying optimization objectives, selection criteria, and evaluation protocols. In particular, across prior work, these components are consistently structured around identifying a single circuit that is sufficient to recover task performance. This typically involves searching for a minimal or highest-scoring structure, under criteria such as performance preservation, sparsity, or faithfulness (fidelity), and evaluating success based on whether this structure alone can account for the behavior of the full model. As a result, the methodology itself centers on isolating a single dominant mechanism for the task.
>
> Particularly, to expand on our points in the paper, prior work often defines circuits through criteria such as faithfulness, completeness, and minimality. For example, Wang et al. (2022) require that a circuit “can perform the task as well as the whole model,” “contains all task-relevant nodes,” and “excludes irrelevant ones,” while Shi et al. (2024) similarly emphasize performance preservation, localization, and minimality. In parallel, several approaches optimize for a single structure that minimizes the loss gap under sparsity constraints (e.g., Li et al., 2024; Bhaskar et al., 2024). Taken together, these formulations consistently frame the goal as identifying a single self-contained mechanism for the task.
>
> Our goal is therefore not to attribute a formal claim to prior work, but to identify and make explicit a recurring assumption reflected in existing methodologies. By formulating this assumption as the Functional Anisotropy Hypothesis, we make it possible to systematically evaluate and test it. Our empirical results are designed precisely for this purpose, and show that multiple distinct mechanisms can faithfully realize the same task, challenging the one-to-one framing suggested by prior approaches.
>
> In summary, our intention is not to ascribe an explicit assumption to prior work, but to surface and examine an implicit pattern reflected in existing methodologies. By making this assumption explicit, we are able to evaluate it systematically and uncover limitations that are otherwise difficult to observe. We want to thank the reviewer again for this insightful and constructive question, which has helped clarify and sharpen our argument significantly. We will for sure incorporate these discussion into  the camera-ready version!
>
> ### **Q2: Limitation Section**
>
> Thank you for the careful and thoughtful reading. We agree that the limitations of our experimental setup can be articulated more clearly, and we will include a dedicated discussion in the camera-ready version.
>
> First, most experiments are conducted on GPT-2 and Pythia; extending to larger and more recent models remains future work. We note, however, that this level of coverage already matches or exceeds that of prior work. Second, our analysis assumes a fixed circuit granularity induced by the residual-stream computation graph, which should be viewed as a preliminary design choice. Third, while we establish the *existence* of multiple faithful task-supporting mechanisms, we do not determine which mechanism is preferentially used during standard inference, even though each is independently sufficient in isolation.

---

> > ### Author Rebuttal · Reviewer_NXvU · 2026-04-01
> >
> > Thank you for the detailed rebuttal. The authors did address my concern by clarifying that their claim is not that prior work explicitly states a formal “Functional Anisotropy Hypothesis,” but rather that such an assumption is implicitly reflected in how prior methods are formulated and evaluated. This clarification is helpful and improves the framing.
> >
> > That said, I remain unconvinced that the paper’s central theoretical framing is yet sufficiently rigorous. My original concern was not only whether this assumption exists implicitly in prior work, but whether the paper formulates that implicit pattern with enough precision to support the strength of its claims. At present, the argument still feels somewhat interpretive and hand-wavy to me, and I am not fully persuaded that the paper has cleanly identified and then refuted a well-specified hypothesis.
> >
> > It is also possible that this reflects limits in my own background on this particular line of work rather than a definitive flaw in the paper. Given that, while I appreciate the clarification in the rebuttal, I would prefer to defer the final judgment on how much this issue should affect the decision to the AC and other reviewers with stronger subject-specific expertise.

---

> > > ### Author Response · Authors · 2026-04-04
> > >
> > > Thank you for acknowledging our response. We want tofurther clarify the rigor of our formulation. While our specific statement of the Functional Anisotropy Hypothesis is not explicitly given in prior circuit or sheaf discovery work, we ground it in closely related assumptions from the broader mechanistic interpretability literature. We hope this makes the framing clearer, including for readers less familiar with this line of work.
> > >
> > > Particularly, Méloux et al.‘s (2025) work do have a particular formulation that distinguish the “only a single” vs. “at least a single” distinction highlighted by the reviewer.
> > >
> > > > In this work, we question a property of explanation that appears to be tacitly taken for granted: do MI criteria guarantee a unique explanation of a fixed behavior?
> > > >
> > >
> > > Although both works use the term “circuit,” they refer to fundamentally different objects, as clarified in our response to Reviewer Pkpb. The term “circuit” is overloaded: Méloux et al. (2025) define circuits in fully connected networks with explicit neuron-to-neuron edges, whereas we study Transformer models where edges reflect information flow in the residual stream. In particular, computation follows $f(x_i) = x_{i-1} + \text{Attn}(x_{i-1}) + \text{MLP}(x_{i-1} + \text{Attn}(x_{i-1}))$, rather than direct neuron-level connections.
> > >
> > > Given this, we would like to highlight Definition 2 in Méloux et al. (2025), which provides an existing and rigorous formulation that we can build on.
> > >
> > > > A mapping between low-level values … and high-level values […]. Each associates the neural network activations with the values of the corresponding high-level variable. […] Each mapping should be surjective $(\forall f_i, \exists h \in \mathbb{R}^{|V_j|} : \tau_j(h) = f_i)$ and have a non-empty pre-image $(\forall f_i, \tau_j^{-1}(f_i) \neq \emptyset)$.
> > > >
> > >
> > > We adapt this perspective to our setting by replacing neuron-level activations with subgraphs, and by binding the explanatory algorithm to the task solver. Concretely, let $M$ be a Transformer, and let $\mathcal{G}(M)$ denote the set of admissible residual-stream subgraphs of $M$ under a fixed circuit framework. Let $\mathcal{A}_T$ denote the set of task-solvers for task $T$. We say that a subgraph $G \in \mathcal{G}(M)$ realizes an explanatory algorithm $A \in \mathcal{A}_T$ if, under the fixed evaluation protocol, the behavior induced by $G$ is consistent with $A$ on task $T$, up to tolerance $\varepsilon$. Formally, this induces a relation $\rho \subseteq \mathcal{G}(M) \times \mathcal{A}_T,$ where $(G,A) \in \rho$ means that $G$ is an $\varepsilon$-faithful realization of $A$ for task $T$.
> > >
> > > Under this adaptation, uniqueness is not the claim that there exists at least one faithful circuit for $T$, but that the family of faithful subgraphs solving $T$ collapses to a single canonical explanatory object. Equivalently, if there exist subgraphs $G_1, G_2 \in \mathcal{G}(M)$ such that both are $\varepsilon$-faithful task-solvers for $T$, yet remain structurally $\delta$-low-overlap, then the explanatory object for solving $T$ is not unique.
> > >
> > > Our work is novel. Méloux et al. (2025) provide useful theoretical grounding for reasoning about the uniqueness of explanations by formalizing mappings between low-level representations and high-level functions. We thank the reviewer for pointing this out, which helped us clarify our formulation and its conceptual grounding. However, the object of study differs: Méloux et al. consider fully connected networks with neuron-to-neuron edges, whereas we study Transformer models where edges capture information flow in the residual stream. As such, their formulation does not directly apply. We instead adapt their framework to our setting and study a different question, showing that multiple distinct, faithful mechanisms can support the same function in Transformer circuits.
> > >
> > > Respectfully, we would also like to highlight that other reviewers independently agreed that our work makes an important, yet previously implicit, assumption in circuit discovery explicit. In particular:
> > >
> > > - Reviewer Pkpb: “The fundamental claim is timely given that single-circuit assumptions underlie the dominant approaches for circuit evaluation such as the compiled Tracr circuits in InterpBench (Gupta et al. 2025) and MIB (Mueller et al. 2025).”
> > > - Reviewer kHxp: “Most of the community focuses on discovering a single circuit for a task, so I think this is a very relevant finding!”
> > > - Reviewer QBks: “The authors highlight a prevailing assumption in the mechanistic interpretability field, which assumes that a single circuit is responsible for model behavior.”
> > >
> > > That said, we are very grateful that the reviewer raised this point, which ultimately helped us refine our framing and make our claim clearer and more accessible, both to the circuit discovery community and to the broader AI/ML/NLP audience.

---

### Official Review · Reviewer_QBks · 2026-03-11

**Soundness:** 3
**Presentation:** 2
**Significance:** 3
**Originality:** 3
**Overall Recommendation:** 4
**Confidence:** 3

**Summary:**

In this paper, the authors examine the Functional Anisotropy Hypothesis, an assumption underlying many circuit and sheaf discovery (CSD) methods. They present empirical evidence suggesting that this hypothesis does not hold in practice. Moreover, they analyze three different CSD methods and find that these methods identify multiple sheaves rather than a single unique mechanism. Based on these observations, the authors propose a new hypothesis, the Distributive Dense Circuit Hypothesis, which proposes that multiple structurally distinct, low-overlap mechanisms (circuits or sheaves) can simultaneously and faithfully implement the same task behavior. The authors further support this claim with a theoretical proof.

**Compliance With Llm Reviewing Policy:**

Affirmed.

**Final Justification:**

I thank the authors for their rebuttal. They have addressed my main concerns, so I maintain my positive score. I believe the paper is very interesting and can further our understanding of how LLMs work.

**Key Questions For Authors:**

1. If multiple low-overlap mechanisms exist, which ones are used by the LLM to perform the task under no interventions? The paper mentions that they compete but does not specify further on the topic. Does the model realize that there are multiple mechanisms and have some process to select the answer? In a way, would that satisfy the Functional Anisotropy Hypothesis?
2. When finding the various mechanisms, did you compare it to the ground truth circuit? For example, in IOI, each attention head type has specific attention patterns to indicate its functional role. Then, in the alternative circuits / sheaves, do the nodes follow the same patterns?
3. I personally find it odd that a circuit with only 11 edges can achieve over 90% accuracy, since it indicates that the circuit only has 12 nodes. I skimmed the code and saw that in the forward method, the masking is performed on the parameter outputs. However, the residual stream also includes the input embeddings, which could influence the circuit’s performance. What would the circuit performance be if the input embeddings were removed from the residual stream? Or, conversely, you could mask only the circuit nodes and measure the drop in performance, which eliminates these confounding variables.

**Limitations:**

Yes

**Strengths And Weaknesses:**

Soundness: The paper is generally technically sound and presents a coherent argument supported by both empirical and theoretical analysis. The authors provide empirical results that demonstrate the Functional Anisotropy Hypothesis does not hold true in various experiment settings and CSD methods. Furthermore, they propose their own hypothesis and support it with a theoretical proof, providing a formal explanation of their findings. However, the authors limited their experiments to only GPT-2, which is a common model used for CSD, but they could have explored other models to ensure generalizability of their results.

Presentation: Overall, the paper is well structured and follows a clear narrative that explains the relevance of the findings and motivates the proposed hypothesis. On the other hand, the paper appears somewhat imbalanced between the empirical results and the theoretical analysis. In particular, Section 5 is relatively brief given that a central contribution of the paper is the theoretical proof that multiple mechanisms can exist simultaneously. Expanding this section with more detailed explanations, intuition, or discussion of the assumptions and implications of the proof would strengthen the presentation.

Significance: The authors highlight a prevailing assumption in the mechanistic interpretability field, which assumes that a single circuit is responsible for model behavior, despite the well-known distributed nature of functionality in LLMs. Through their experiments, the authors revealed a major limitation, but it would have been more impactful if they also provided potential solutions to address this limitation. Nonetheless, if the authors’ claims hold broadly, the implications could be substantial because this would affect how researchers interpret discovered circuits and may require new evaluation criteria for interpretability methods.

Originality: The paper is original in its framing and perspective because it critically examines a foundational assumption underlying many existing methods. It offers an alternative perspective on circuits, that multiple mechanisms can co-exist for a single task, as well as evidence supporting this perspective.

---

> ### Author Rebuttal · Authors · 2026-03-30
>
> Thank you so much for recognizing the soundness, originality, significance, and presentation of our work. We are highly motivated by these kind words! We are also grateful for the interesting questions and suggestions, which help clarify several important aspects of our work. We will incorporate these clarifications into the revised version to further improve the paper.
>
> ### **Q1. Clarifying Coexistence vs. Competition of Mechanisms**
>
> Thank you for this interesting question. At present, we do not know which specific mechanism is utilized during standard (no-intervention) inference, although understanding this is an important direction for future work.
>
> What we show instead is that multiple low-overlap, functionally valid mechanisms exist within the same model, each of which is sufficient to support the task and remains causally relevant (i.e., removing any one leads to a substantial degradation in performance).
>
> Your question also helps us make our phrasing more precise. In particular, we will revise terms such as “compete” to better reflect that multiple mechanisms **coexist** within the same model and can redundantly realize the same behavior.
>
> This perspective challenges strong forms of the Functional Anisotropy Hypothesis that assume a single dominant mechanism, while remaining compatible with weaker forms where multiple mechanisms jointly or redundantly support the task.
>
> ### **Q2: Ground-Truth Circuits**
>
> We do not compare our discovered mechanisms to a single “ground truth” circuit for two reasons. First, the notion of a canonical circuit has been increasingly challenged. Recent work (e.g., Hanna et al., 2025) shows that structural agreement with a reference circuit is not a reliable indicator of mechanism, and that functional fidelity is a more appropriate criterion.
>
> Second, the notion of a “circuit” differs across these works. In IOI-style analyses, circuits are defined in terms of attention heads and their characteristic attention patterns, whereas our circuits (sheaves), as well as those in ACDC and DiscoGP, are defined as subgraphs over residual stream components. As a result, there is no shared notion of ground truth across these formulations, and we do not expect alternative sheaves to reproduce the same attention patterns.
>
> We further support this perspective with results on InterpBench. On its IOI task, we recover a functionally faithful circuit using OASR that achieves 88.9% accuracy with 15.25% edge density (169/1108), compared to the gold circuit’s 100% accuracy and 15.52% density (172/1108), while exhibiting low overlap (IoU = 11.08%). This demonstrates that even in settings with a designated reference circuit, multiple low-overlap yet functionally sufficient mechanisms can exist.
>
> ### **Q3: Implementation Clarification**
>
> Thank you for this careful observation. In our implementation, masking is applied directly to the residual stream contributions at the edge level. Concretely, each source node (including attention heads, MLPs, and the input embeddings) is treated as a potential writer to downstream components, and each such connection is associated with a learned mask.
>
> As shown in our code, the residual stream is first organized into contributions from all previous nodes (`pre_layer_node_id`), where the last index corresponds to the input embeddings:
>
> ```python
> n_pre_layer_nodes = (cfg.n_heads + 1) * layer_id + 1
> ```
>
> We then apply masking via a linear projection:
>
> ```python
> masked = einsum(
>     "batch pos pre_layer_node_id d_model, pre_layer_node_id head_id -> batch pos head_id d_model",
>     resid_pre, mask
> )
> ```
>
> This operation explicitly mixes and gates all incoming residual contributions—including those from the input embeddings—through the learned mask. As a result, embeddings can only influence downstream computation through the corresponding masked edges, and are not an unaccounted pathway.
>
> Therefore, any performance achieved by a small circuit necessarily includes the embedding-related edges that are selected by the optimization. The ability to find circuits with very few edges thus reflects redundancy in the model, rather than uncontrolled information flow outside the discovered subgraph.

---

> > ### Author Rebuttal · Reviewer_QBks · 2026-04-02
> >
> > Thanks for your explanations. I believe all my concerns have been addressed. However, I do have a follow up question on your response to Question 1. The coexisting mechanisms all implement the same task, but removing any one of them would cause a significant drop. This suggests that the model performs the same task in multiple different implementations and uses the combination of the answers to generate a response? I think it would be an interesting direction to explore.

---

> > > ### Author Response · Authors · 2026-04-04
> > >
> > > Thank you so much for acknowledging our response. We are glad that all of your concerns have been addressed.
> > >
> > > We believe the follow-up question points to an important direction for future work: understanding what circuits—or more broadly, mechanisms—actually represent. Building on your observation, we see two possible hypotheses.
> > >
> > > First, it is possible that the model performs the same task through multiple implementations and combines their outputs to produce a final response. Under this view, different mechanisms may contribute partial evidence, with later components implicitly aggregating or selecting among them. This perspective is broadly consistent with prior work on factual recall in LLMs (e.g., Meng et al. 2022; Geva et al. 2023; Niu et al. 2024; 2025; Hernandez et al. 2024), which suggest that lower layers gather relevant information while upper layers consolidate it into a final prediction. Notably, these works do not operate at the level of circuits; a more fine-grained circuit-level analysis could help confirm and refine this view.
> > >
> > > Second, our findings also support a *distributed view*, where no single mechanism is solely responsible for a given behavior. Rather than viewing model behavior as arising from a collection of separate mechanisms, this perspective suggests that the mechanism itself is inherently distributed across multiple components, potentially spanning different levels of the model. In this sense, the underlying process is not a set of distinct mechanisms, but a single distributed mechanism expressed across the system. Under this view, what we identify as a “circuit” may not correspond to a unique causal pathway, but rather to one of many valid decompositions of a fundamentally distributed mechanism, more like a *field* than a discrete, localized circuit.
> > >
> > > We believe that distinguishing between these hypotheses, or understanding how they may coexist, would be an exciting direction for future work. We thank the reviewer again for this insightful question. We are genuinely excited about this direction and look forward to incorporating this discussion into the final version of the paper. If you feel that our revisions have adequately addressed your concerns, we would greatly appreciate it if you could consider reflecting this in your final evaluation.

---

### Official Review · Reviewer_kHxp · 2026-03-12

**Soundness:** 2
**Presentation:** 3
**Significance:** 4
**Originality:** 4
**Overall Recommendation:** 4
**Confidence:** 3

**Summary:**

This paper questions the assumption that a model uses a single circuit to perform a task. The authors show that there can actually be multiple faithful and distinct circuits that preform the same task. To find these circuits, they introduce a method called Overlap-Aware Sheaf Repulsion. In addition they provide a theoretical analysis to support this obbservation.

**Compliance With Llm Reviewing Policy:**

Affirmed.

**Final Justification:**

The authors have answered all of my questions and committed to clarifying the parts that were unclear in the camera-ready version. Accordingly I have decided to increase my score.

**Key Questions For Authors:**

1. Please clarify how you measure accuracy/performance of a circuit.
2. I would be interested in seeing the overlap at the node level. Do the circuits share nodes but differ in their edges?


In Section 4 you mention that you found circuits consisting of 3 and 11 edges that achieve 90% and 86% accuracy under zero ablation respectively.

3. Why should we expect the intersection of circuits to form a connected circuit?
4. Table 4 shows that ACDC using zero ablation finds circuits with 2,194 edges on average that achieve 87% accuracy. Do you have an explanation for how circuits with only 11/3 edges can achieve comparable performance? Additionally, in the original IOI paper the authors measured minimality and showed that removing any node from the circuit drastically decreases circuit performance.

5. According to the theorem in App. A, accuracy is defined based on the argmax over logits. And under this definition, it is possible to discover multiple circuits that preserve the same argmax prediction. Would it also be possible to discover multiple circuits that preserve the full output distribution (kl score for example)? or logit difference (that is being used in the original IOI)?

**Limitations:**

I don't see any potential negative societal impact of this work

**Strengths And Weaknesses:**

### Strengths
-  This paper is written very nicely and clearly.
-  The paper provides both empirical evidence across multiple tasks and a theoretical analysis supporting the existence of many faithful and distinct circuits. Most of the community focuses on discovering a single circuit for a task, so I think this is a very relevant finding!

## Weaknesses
- I could not find in the paper how accuracy/performance is measured. This strongly affect the results! In addition, see the questions in the next section.

---

> ### Author Rebuttal · Authors · 2026-03-30
>
> Thank you so much for your thoughtful and insightful review. We are very encouraged that you find the paper clear and the results relevant to the broader community.
>
> The main questions focus on clarifying technical details. We appreciate your careful reading and will incorporate these clarifications into the paper.
>
> ### **Q1: Accuracy & Performance**
>
> Thank you for the great suggestion! We will certainly clarify this in the paper and make the evaluation protocol more explicit in the camera-ready version.
>
> *Evaluation:* A prediction is correct if $P(✔️) > P(❌)$.
>
> *Accuracy:* Reported as the fraction of inputs satisfying this condition.
>
> *Implementation:* All evaluations use zero-ablation under the masked computation graph, without modifying model weights.
>
> ### **Q2: Node-level Overlap**
>
> Thank you for this very interesting question. We observe a similar pattern at the node level. However, this effect is less pronounced than at the edge level, as individual circuits already exhibit relatively high node density (we do not explicitly optimize for node sparsity), making node-level overlap naturally high and less discriminative. In contrast, edge-level overlap provides a sharper signal of differences between circuits.
>
> **IoU_E / IoU_N: edge / node IoU (%); Node Density (%).**
>
> | Task | IoU Node (%) | Node Density A (%) | Node Density B (%) | IoU Edge (%) |
> | --- | --- | --- | --- | --- |
> | IOI | 64.2 | 67.7 | 73.0 | 4.1 |
> | BLiMP | 66.9 | 76.7 | 77.1 | 5.1 |
> | AGA | 62.1 | 62.8 | 76.0 | 6.2 |
> | ANA | 61.4 | 66.4 | 74.7 | 5.3 |
> | DNA | 55.3 | 57.9 | 54.2 | 5.8 |
> | DNA i | 49.0 | 51.1 | 60.4 | 6.2 |
> | DNA a | 67.2 | 71.5 | 66.7 | 7.5 |
> | DNA ia | 71.8 | 76.2 | 64.8 | 6.4 |
> | Docstring | 93.0 | 89.0 | 88.0 | 11.0 |
>
> ### **Q3: Circuit Intersection Clarification**
>
> We do not assume intersections are connected. We compute edge-wise intersections and retain only edges that lie on a valid path from embedding to unembedding; otherwise, the intersection is discarded. E.g., circuits A → B → C and A → B → D, their intersection does not form a valid path and is just the empty set. This ensures functional validity and yields a connected component in practice. This step (implemented in `discover_circuit.py`) is only used when constructing the 3-edge circuit and does not affect IoU or other main results. We will update the paper to clarify this point.
>
> ### **Q4: ACDC & the Original IOI**
>
> The difference primarily arises from the optimization objective. ACDC uses a local, edge-wise criterion (KL change), which retains edges that are individually important but jointly redundant. In contrast, our setting directly optimizes task performance under sparsity, identifying globally sufficient subgraphs. OASR further exploits this redundancy by exploring alternative pathways in the model, enabling the discovery of substantially smaller circuits that achieve comparable performance.
>
> The original IOI results are not directly comparable to ours due to differences in what is meant by a “circuit.” The term has been used to refer to different structures across the literature: the original IOI paper (Wang et al., 2022) operates at the level of collections of attention heads, whereas ACDC, DiscoGP, and our work consider circuits as subgraphs over residual components. As a result, minimality findings in IOI are tied to that particular definition and extraction procedure, and do not directly transfer to our setting.
>
> ### **Q5. Argmax vs. KL**
>
> We thank the reviewer for suggesting the use of KL divergence as an alternative metric. Indeed, our framework can be naturally extended to a KL-based notion of functional equivalence. Intuitively, if two circuits produce sufficiently close logits (e.g., bounded in $\ell_\infty$ norm), then by the smoothness of the softmax function, the induced output distributions will also be close in KL divergence. This implies that controlling logit perturbations provides an implicit bound on KL.
>
> Formally, one can define a *faithfulness threshold* under KL:
>
> $\text{KL}(P_\text{model}||P_\text{circuit})<\varepsilon,$
>
> where $\varepsilon$ is a small constant. Under this criterion, multiple circuits may satisfy the KL constraint simultaneously, since KL is relatively insensitive to small redistributions of probability mass—especially when the top-1 prediction remains unchanged.
>
> This highlights a key distinction: KL captures distribution-level similarity, but does not necessarily enforce agreement at the decision boundary (argmax). As a result, different circuits can yield low KL divergence while exhibiting structurally distinct internal mechanisms or even differing behaviors in edge cases. This further supports our use of argmax as a stricter notion of functional equivalence in circuit discovery.
>
> Nevertheless, we agree that KL provides a valuable complementary perspective, and we will include additional KL-based analysis in the camera-ready version to strengthen our empirical validation.

---

> > ### Author Rebuttal · Reviewer_kHxp · 2026-04-04
> >
> > I thank the authors for answering all of my questions and addressing my concerns. I have decided to adjust my score.

---

> > > ### Author Response · Authors · 2026-04-06
> > >
> > > Thank you so much for your acknowledgement. It is really great to see that all of your concerns have been addressed. We truly appreciate your careful reading and thoughtful feedback throughout the process. This has been a very productive discussion, and we will reflect it in the camera-ready version.
> > >
> > > Thank you again for the great discussion and for your thoughtful review. We really appreciate the time and care you put into the evaluation. We also sincerely appreciate your updated evaluation.

---

### Official Review · Reviewer_Pkpb · 2026-03-13

**Soundness:** 3
**Presentation:** 3
**Significance:** 3
**Originality:** 3
**Overall Recommendation:** 4
**Confidence:** 4

**Summary:**

This paper targets a key assumption in current methods for circuit discovery and sheaf discovery in the mechanistic interpretability field, which is that there is one true ground truth circuit in the model for any given task or dataset. The authors refer to this as the Functional Anisotropy Hypothesis. Currently, circuit discovery algorithms are benchmarked based on accuracy with respect to ground truth circuits. In this paper, the authors demonstrate that for a given LLM task there can exist multiple structurally distinct mechanisms with low overlap that are simultaneously faithful to the model behavior on that task. This phenomenon is different from the backup heads and Hydra effect reported in prior work, since those are referring to mechanisms that only activate when other components are ablated out -- in this setting, even without ablation the distinct faithful circuits can co-exist. Furthermore, the components in the intersection between these different mechanisms need not all be necessary for the model to perform the task. To substantiate these claims in the case of sheaf discovery, the authors introduce an overlap-aware sheaf repulsion method that is used to discover examples of functionally equivalent sheaves with minimal overlap. There is also an existence theorem in the appendix showing that multiple low-overlap faithful circuits must exist.

**Compliance With Llm Reviewing Policy:**

Affirmed.

**Final Justification:**

The rebuttal addressed several of my concerns, particularly the framing of the "simultaneously activated circuits" claim and the individual edge ablations for the 3-edge sheaf. The Functional Anisotropy Hypothesis (FAH) seems clearly intended as a model-level rather than method-level claim. However, the Section 3.3 results for ACDC, EAP, etc. appear to reflect variability in the methods themselves under arbitrary configuration choices, rather than directly establishing model-level mechanism multiplicity in those formalisms, and this was not resolved in the rebuttal. I appreciated the InterpBench IOI addition, though Tracr-compiled tasks would more directly test FAH and the conditions of Theorem A.4 since they carry a stronger ground-truth-circuit claim by construction. Overall, I am keeping my score as a weak accept: the core FAH claim could be better substantiated, but the evidence within limited settings is solid.

**Key Questions For Authors:**

Aside from the concerns in the Weaknesses section, I have the following questions:

1. The paper mentions Tracr-based benchmarks (e.g. InterpBench) as an example of how Functional Anisotropy is assumed in circuit evaluations. If you ran the overlap-aware repulsion method on InterpBench models, would you find multiple low-overlap faithful circuits? If not, how do these models sidestep the conditions in Theorem A.4?
2. Meloux et al. 2025 also demonstrated that multiple circuits can replicate the same behavior, which they frame as non-identifiability of mechanistic explanations. The core thesis here overlaps substantially, yet this work is only cited briefly in Appendix K. It would be helpful to explicitly discuss how the paper's contributions extend beyond this prior work, aside from the inclusion of sheaves alongside the circuit discovery methods.

**Limitations:**

Yes

**Strengths And Weaknesses:**

### Strengths

The fundamental claim is timely given that single-circuit assumptions underlie the dominant approaches for circuit evaluation such as the compiled Tracr circuits in InterpBench (Gupta et al. 2025) and MIB (Mueller et al. 2025). It is also in line with prior work on the non-uniqueness of circuits (Meloux et al. 2025, cited in the appendix).

The proposed overlap-aware sheaf repulsion method appears well motivated and a clever alternative to random search or resampling. The empirical demonstrations of the method are across a wide enough set of datasets to be convincing, and included layer-wise edge connectivity analysis to demonstrate that the sheaves they recovered are markedly different. The discovery of the 3-edge IOI sheaf is particularly surprising. The random init baseline helps understand the effect of the repulsion term as distinct from the inherent noise in the circuit discovery procedure.

### Weaknesses

**Circuit discovery methods:** The overlap-aware repulsion approach is only applicable to DiscoGP, the sheaf discovery method, and not to ACDC, EAP, and EP, which are circuit discovery methods that are more commonly used in practice. There are still evaluations on the circuit discovery methods, but they focus on variability across runs, hyperparameters, traversal orders, etc. These experiments demonstrate that the methods can return multiple different circuits, but it is not clear how they support the broader claim of "the model itself has multiple independent mechanisms for the same task".

**The 3-edge sheaf:** This argument is somewhat confusing. On one hand, there appears to be a 3-edge sheaf that on its own is sufficient for 86.7% performance and also necessary since removing it from the set of IOI sheaves they discovered drops the performance to 52.3%. However, when splitting the IOI dataset into subsets corresponding to the different task templates (ABBA and BABA), the 3-edge sheaf is not necessary any more. The abstract claims that "none of its edges is individually indispensable" based on this result.
1. The table only shows numbers for the ablation of all three edges at once, and not ablating individual edges, so it seems mismatched with the claim. It is possible that some edges are essential for one sub-template but not the other.
2. The experiment supports the narrower conclusion that task granularity affects what appears to be essential for task performance, but it is not clear how this addresses the broader claim of "the absence of indispensable components".
3. It would also be helpful to verify if this result also holds on the circuit discovery methods.

**Simultaneously active circuits:** It is not clear how the given results support the claim in the intro that multiple circuits are "simultaneously active and often compete during standard inference", in contrast with backup heads and hydras that are active when another component is ablated. It appears as though the current evaluations are on the sufficiency and necessity of each sheaf in isolation while other components are zero-ablated, and complements are also evaluated under a zero-ablation regime. It would be helpful to further clarify how the results demonstrate that the circuits are simultaneously active without ablation.

---

> ### Author Rebuttal · Authors · 2026-03-30
>
> Thank you for the thoughtful and constructive review. We appreciate the positive assessment of our work and its relevance. Below, we clarify the main points raised (Comments 1–3 and Q1,2) and outline how we will revise the paper to better reflect these insights.
>
> ### **Comment 1: OASP on Other CSD Methods**
> We clarify that the distinction between sheaves and circuits is not the main focus of our argument, but it is nevertheless conceptually relevant. Yu et al. (2025) coined the term sheaf to disambiguate the previously overloaded notion of circuit. In our setting (edge masking without weight pruning), both formulations correspond to subgraphs that mediate task behavior, and the primary differences is about evaluation rather than in their underlying computational role. Moreover, we have discussed other CSD methods in Section 3.3.
>
> Regarding method applicability, the key distinction is whether a method involves a *trainable process* into which an overlap term can be introduced. In the edge-only setting, EP is equivalent to DiscoGP, and OASR applies. In contrast, methods such as ACDC or EAP do not involve a learning process that allows such a loss term.
>
> Importantly, our claim does not hinge on OASR being applicable to every method. All CSD approaches aim to identify subnetworks that perform the task. Our key observation is that one can find circuits/sheaves that perform the same task with minimal overlap, regardless of how they are defined or discovered.
>
> ### **Comment 2: 3-Edge Sheaf Elaboration**
> We would like to take the opportunity to clarify our argument around the 3-edge sheaf.
>
> Recall that the 3 edges are:
> - **Edge 1**: Input→MLP₀;
> - **Edge 2**: MLP₀→Attn₁₀H7 (V);
> - **Edge 3**: Attn₁₀·H7→Output.
>
> Edge ablation results:
>
> | Edge 1 | Edge 2 | Edge 3 | Active Edges | Acc % (selected edge(s) removed) |
> | --- | --- | --- | --- | --- |
> | ❌ | ❌ | ❌ | 0 | 100 |
> | ❌ | ❌ | ✔️ | 1 | 99.9 |
> | ❌ | ✔️ | ❌ | 1 | 99.9 |
> | ❌ | ✔️ | ✔️ | 2 | 99.8 |
> | ✔️ | ❌ | ❌ | 1 | 31.6 |
> | ✔️ | ❌ | ✔️ | 2 | 31.4 |
> | ✔️ | ✔️ | ❌ | 2 | 31.3 |
> | ✔️ | ✔️ | ✔️ | 3 | 31.2 |
>
> These results show a clear progression. First, removing any single edge leaves performance largely unchanged, indicating that no edge is indispensable within this particular sheaf. Moreover, we extend the analysis by asking whether each of these edges can also be avoided globally, i.e., whether there exist other sheaves that achieve strong performance without using that edge at all. As shown in Table 6, for each of the 3 edges, we can identify alternative sheaves that exclude it and still achieve strong performance. Taken together, these results show that each edge can be avoided by alternative, functionally valid mechanisms, and is therefore not intrinsically indispensable.
>
> ### **Comment 3: "Simultaneously Activated Circuits" Phrasing**
> Thank you so much for the constructive suggestions regarding the phrasing “simultaneously active and often compete.” We will follow your suggestion to make our message more concise and unambiguous.
>
> What we want to convey is that there are multiple distinct circuits (or sheaves) in LLMs that (1) can each support the same task, while (2) both remain causally relevant: removing either one leads to a substantial degradation in task performance. This is indeed different from backup-style or hydra effects.
>
> We will revise our wording in the introduction and abstract to better reflect this point. In particular, we will replace the original phrasing (line 088-091) with something along the lines of:
>
> > … for a single task, multiple distinct circuits or sheaves **coexist** within the same model during inference—each can support the task, while removing either one degrades performance.
>
> ### **Q1: InterpBench**
> Thank you for this insightful suggestion. We can indeed recover a functionally faithful circuit using OASR on InterpBench’s IOI task, demonstrating that low-overlap alternative circuits can also be identified in this setting. Our circuit achieves 88.9% accuracy with 15.25% edge density (169/1108), compared to the gold circuit’s 100% accuracy and 15.52% density (172/1108), while exhibiting low overlap (IoU = 11.08%). Due to time constraints during the rebuttal, we evaluate only this task; the result is consistent with our main findings, and we will extend this analysis on other InterBench tasks in the camera-ready version.
>
> ### **Q2: Méloux et al. 2025**
> Thank you for pointing us to this relevant paper. We briefly discussed it in the current draft (line 2221), and will expand this comparison in the revision.
>
> There are two key differences. First, Méloux et al. (2025) study simple models and tasks, whereas we evaluate pretrained language models (e.g., GPT-2, Pythia) on realistic linguistic tasks. Second, their notion of “circuit” differs from our residual-based circuits/sheaves; their work is better viewed as providing foundational evidence for superposition in neural networks, rather than directly characterizing circuits in transformer LMs.

---

> > ### Author Rebuttal · Reviewer_Pkpb · 2026-04-04
> >
> > Thank you for the detailed rebuttal, these have addressed my primary concerns. In particular, the individual edge ablation data for the 3-edge sheaf now more clearly supports the claim that no single edge is indispensable within this sheaf, and the preliminary InterpBench result is a useful addition. Regarding other InterpBench tasks, the Tracr-compiled tasks seem like the most informative for this setting since they are designed with the goal of having a unique ground truth circuit.

---

> > > ### Author Response · Authors · 2026-04-06
> > >
> > > Thank you for the thoughtful acknowledgement. We are glad that all of your concerns have been addressed.
> > >
> > > We also really appreciate your suggestion to incorporate Tracr-compiled tasks. They provide an additional perspective through a setting where a unique ground-truth circuit is known within a semi-synthetic transformer. This is a great direction, and we will certainly include it in the camera-ready version.
> > >
> > > Thank you again for the great discussion and for your constructive feedback. We would greatly appreciate it if you could take this into account when finalizing your evaluation.

---

### Decision · Program_Chairs · 2026-04-30

**Decision:**

Accept (regular)

**Comment:**

This paper is on discovery of Circuits and Sheafs in the context of LLMs.  The paper floats the Functional Anisotropy Hypothesis, there exists a unique circuit in LLM for each function, which is central to existing algorithms. The paper then proposes a Distributive Dense Circuit hypothesis and provides a theoretical analysis to establish that there exists (almost) non-overlapping circuits. This paper should be of interest to
audience involved in mechanistic interpretability.

During the rebuttal most of the concerns by the referees were addressed. However, there was no strong support for this paper. At this point it is very hard to make a very strong case for accepting the paper.